# SCAFFLSA: Taming Heterogeneity in Federated Linear Stochastic Approximation and TD Learning

**Paul Mangold**
CMAP, UMR 7641,
École polytechnique

**Sergey Samsonov**
HSE University,
Russia

**Safwan Labbi**
CMAP, UMR 7641,
École polytechnique

**Ilya Levin**
HSE University,
Russia

**Reda Alami**
Technology Innovation Institute,
9639 Masdar City, Abu Dhabi,
United Arab Emirates

**Alexey Naumov**
HSE University,
Steklov Mathematical Institute
of Russian Academy of Sciences

**Eric Moulines**
CMAP, UMR 7641,
École polytechnique
MZBUAI

## Abstract

In this paper, we analyze the sample and communication complexity of the federated linear stochastic approximation (FedLSA) algorithm. We explicitly quantify the effects of local training with agent heterogeneity. We show that the communication complexity of FedLSA scales polynomially with the inverse of the desired accuracy $\epsilon$. To overcome this, we propose SCAFFLSA, a new variant of FedLSA that uses control variates to correct for client drift, and establish its sample and communication complexities. We show that for statistically heterogeneous agents, its communication complexity scales *logarithmically* with the desired accuracy, similar to Scaffnew [37]. An important finding is that, compared to the existing results for Scaffnew, the sample complexity scales with the inverse of the number of agents, a property referred to as *linear speed-up*. Achieving this linear speed-up requires completely new theoretical arguments. We apply the proposed method to federated temporal difference learning with linear function approximation and analyze the corresponding complexity improvements.

## 1 Introduction

Heterogeneity has a major impact on communication complexity in federated learning (FL) [28, 36]. In FL, multiple agents use different local oracles to update a global model together. A central server then performs a *consensus step* to incrementally update the global model. Since communication with the server is costly, reducing the frequency of the consensus steps is a central challenge. At the same time, limiting communications induces *client drift* when agents are heterogeneous, biasing them towards their local solutions. This issue has mostly been discussed for FL with stochastic gradient methods [23, 51]. In this paper, we investigate the impact of heterogeneity in the field of federated linear stochastic approximation (federated LSA). The goal is to solve a system of linear equations where (i) the system matrix and the corresponding objective are only accessible via stochastic oracles, and (ii) these oracles are distributed over an ensemble of heterogeneous agents. This problem can be solved with the FedLSA method, which performs LSA locally with periodic consensus steps. This approach suffers from two major drawbacks: heterogeneity bias, and high variance of local oracles.

A popular means of overcoming heterogeneity problems is the method of control variables, which goes back to the line of research initiated by [23]. However, existing results on the complexity of these methods tend to neglect the linear decrease of the mean squared error (MSE) of the algorithm with the number of agents $N$ [37], or they require a lot of communication [23]. In this paper, we

38th Conference on Neural Information Processing Systems (NeurIPS 2024).

Table 1: Communication and sample complexity for finding a solution with MSE lower than $\epsilon^2$ for FedLSA, Scaffnew, and SCAFFLSA with i.i.d. samples (see Cor. 4.3 for results with Markovian samples). Our analysis is the first to show that FedLSA exhibits linear speed-up, as well as its variant that reduces bias using control variates.

| Algorithm | Communication $T$ | Local updates $H$ | Sample complexity $TH$ |
|---|---|---|---|
| FedLSA [12] | $\mathcal{O}\left(\frac{N^2}{a^2\epsilon^2}\log\frac{1}{\epsilon}\right)$ | $1$ | $\mathcal{O}\left(\frac{N^2}{a^2\epsilon^2}\log\frac{1}{\epsilon}\right)$ |
| FedLSA (Cor. 4.3) | $\mathcal{O}\left(\frac{1}{a^2\epsilon}\log\frac{1}{\epsilon}\right)$ | $\mathcal{O}\left(\frac{1}{N\epsilon}\right)$ | $\mathcal{O}\left(\frac{1}{Na^2\epsilon^2}\log\frac{1}{\epsilon}\right)$ |
| Scaffnew (Cor. F.3) | $\mathcal{O}\left(\frac{1}{q\epsilon}\log\frac{1}{\epsilon}\right)$ | $\mathcal{O}\left(\frac{1}{q\epsilon}\right)$ | $\mathcal{O}\left(\frac{1}{a^2\epsilon^2}\log\frac{1}{\epsilon}\right)$ |
| SCAFFLSA (Cor. 5.2) | $\mathcal{O}\left(\frac{1}{a^2}\log\frac{1}{\epsilon}\right)$ | $\mathcal{O}\left(\frac{1}{N\epsilon^2}\right)$ | $\mathcal{O}\left(\frac{1}{Na^2\epsilon^2}\log\frac{1}{\epsilon}\right)$ |

show that it is possible to reduce communication complexity using control variates while preserving the linear speed-up in terms of sample complexity. Our contributions are the following:

- We provide the sample and communication complexity of the FedLSA algorithm, inspired by the work of [51]. Our analysis highlights the relationship between the MSE of the FedLSA method and three key factors: the number of local updates, the step size, and the number of agents. We provide an exact analytical formulation of the algorithm's bias, which is confirmed in our numerical study. We also give results under Markovian noise sampling.
- We propose SCAFFLSA, a method that provably reduces communication while maintaining linear speed-up in the number of agents. This method uses control variates to allow for extended local training. We establish finite sample and communication complexity for SCAFFLSA. Our study is based on a new analysis technique, that carefully tracks the fluctuations of the parameters and ccommunicationsontrol variates. This allows to prove that SCAFFLSA *simultanously maintains linear speedup and reduced communication*. To our knowledge, this is the first time that these two phenomenons are proven to occur simultaneously in FL.
- We apply both these methods to TD learning with linear function approximation, where heterogeneous agents collaboratively estimate the value function of a common policy.

We provide a synthetic overview of this paper's theoretical results in Table 1 in the general federated LSA setting, and we instantiate these results for federated TD learning in Table 2 (Appendix E). We start by discussing related work in Section 2. We then introduce federated LSA in Section 3, and analyze it in Section 4. In Section 5 we introduce SCAFFLSA, a novel strategy to mitigate the bias. Finally, we illustrate our results numerically in Section 6. Since an important application of LSA is TD learning [47] with linear function approximation, we instantiate the results of Section 3-5 for federated TD learning.

**Notations.** For matrix $A$ we denote by $\|A\|$ its operator norm. Setting $N$ for the number of agents, we use the notation $\mathbb{E}_c[a_c] = N^{-1}\sum_{c=1}^N a_c$ for the average over different clients. For the matrix $A = A^\top \succeq 0, A \in \mathbb{R}^{d\times d}$ and $x \in \mathbb{R}^d$ we define the corresponding norm $\|x\|_A = \sqrt{x^\top A x}$. For sequences $a_n$ and $b_n$, we write $a_n \lesssim b_n$ if there exists a constant $c > 0$ such that $a_n \leq cb_n$ for $n \geq 0$.

## 2   Related Work

**Federated Learning.** With few exceptions (see e.g. [12]), most of the FL literature is devoted to federated stochastic gradient (SG) methods. A strong focus has been placed on the Federated Averaging (FedAvg) algorithm [36], which aims to reduce communication through local training, resulting in *local drift* when agents are heterogeneous [53]. Sample and communication complexity of FedAvg were investigated under a variety of conditions covering both homogeneous [31, 20] and heterogeneous agents [25, 27]. Different ways of measuring heterogeneity for FedAvg have then been proposed [51, 41]. In [44] it was also shown that FedAvg yields linear speedup in the number of agents when gradients are stochastic, a phenomenon that we prove is still present in FedLSA.

In order to correct the client drift of FedAvg, [23] proposed Scaffold, a method that tames heterogeneity using control variates. [17, 38] prove that Scaffold retrieves the rate of convergence of the gradient descent independently of heterogeneity, although without benefit from local training. It has

**Algorithm 1** FedLSA

---

**Input:** $\eta > 0, \theta_0 \in \mathbb{R}^d, T, N, H > 0$
**for** $t = 0$ to $T - 1$ **do**
 Initialize $\theta_{t,0} = \theta_t$
 **for** $c = 1$ to $N$ **do**
  **for** $h = 1$ to $H$ **do**
   Receive $Z_{t,h}^c$ and perform local update: $\theta_{t,h} = \theta_{t,h-1}^c - \eta(\mathbf{A}^c(Z_{t,h}^c)\theta_{t,h-1}^c - \mathbf{b}^c(Z_{t,h}^c))$
 Aggregate local updates $\theta_{t+1} = \frac{1}{N} \sum_{c=1}^{N} \theta_{t,H}^c$           (1)

---

been shown in [37] (with the analysis of ProxSkip, which generalizes Scaffold) that such methods *accelerate* training. However, unlike Scaffold, the analysis of [37] loses the linear speedup in the number of agents. Several other methods with accelerated rates have been proposed [35, 5, 6, 18, 21], albeit all of them lose the linear speedup. Contrary to these papers, we show that our approach to FedLSA with control variates *preserves both the acceleration and the linear speedup*.

**Federated TD learning.** Temporal difference (TD) learning has a long history in policy evaluation [47, 9], with the asymptotic analysis under linear function approximation (LFA) setting performed in [49, 48]. Several non-asymptotic MSE analyses have been carried out in [4, 8, 42, 32, 45]. Much attention has been paid to federated reinforcement learning [33, 43, 52] and federated TD learning with LFA. [26, 7, 34] provides an analysis under the strong homogeneity assumption. Federated TD was also investigated with heterogeneous agents, first without local training [12], then with local training but without linear acceleration [11, 22]. Recently, [50] proposed an analysis of federated TD with heterogeneous agents, local training, and linear speed-up in number of agents. However, [50] do not mitigate the local drift effects, and their conclusions are valid only in the low-heterogeneity setting. In high heterogeneity settings, their analysis exhibits a large bias. Additionally, their analysis requires the server to project aggregated iterates to a ball of unknown radius. In contrast, our analysis shows that FedLSA converges to the true solution without bias even without such projection.

## 3 Federated Linear Stochastic Approximation and TD learning

### 3.1 Federated Linear Stochastic Approximation

In federated linear stochastic approximation, $N$ agents collaboratively solve a system linear equation system with the following finite sum structure

$$\bar{\mathbf{A}}\theta_\star = \bar{\mathbf{b}}, \quad \text{where } \bar{\mathbf{A}} = \frac{1}{N}\sum_{c=1}^{N}\bar{\mathbf{A}}^c, \quad \bar{\mathbf{b}} = \frac{1}{N}\sum_{c=1}^{N}\bar{\mathbf{b}}^c,$$

where for $c \in [N]$, $\bar{\mathbf{A}}^c \in \mathbb{R}^{d \times d}$, $\bar{\mathbf{b}}^c \in \mathbb{R}^d$. We assume the solution $\theta_\star$ to be unique, and that each local system $\bar{\mathbf{A}}^c\theta_\star^c = \bar{\mathbf{b}}^c$ also has a unique solution $\theta_\star^c$. The values of $\bar{\mathbf{A}}^c$'s and $\bar{\mathbf{b}}^c$'s can be different, representing the different realities of the agents. In federated LSA, neither matrices $\bar{\mathbf{A}}^c$ nor vectors $\bar{\mathbf{b}}^c$ are observed directly. Instead, each agent $c \in [N]$ has access to its own observation sequence $(Z_k^c)_{k\in\mathbb{N}}$, that are independent from one agent to another. Agent $c$ obtains estimates $\{(\mathbf{A}^c(Z_k^c), \mathbf{b}^c(Z_k^c))\}_{k\in\mathbb{N}}$ of $\bar{\mathbf{A}}^c$ and $\bar{\mathbf{b}}^c$, where $\mathbf{A}^c : \mathsf{Z} \to \mathbb{R}^{d \times d}$ and $\mathbf{b}^c : \mathsf{Z} \to \mathbb{R}^d$ are two measurable functions. Naturally, we define the error of estimation of $\bar{\mathbf{A}}^c$ and $\bar{\mathbf{b}}^c$ as $\widetilde{\mathbf{b}}^c(z) = \mathbf{b}^c(z) - \bar{\mathbf{b}}^c$, $\widetilde{\mathbf{A}}^c(z) = \mathbf{A}^c(z) - \bar{\mathbf{A}}^c$. This allows to measure the noise at local and global solutions as

$$\varepsilon^c(z) = \widetilde{\mathbf{A}}^c(z)\theta_\star^c - \widetilde{\mathbf{b}}^c(z), \text{ and } \omega^c(z) = \widetilde{\mathbf{A}}^c(z)\theta_\star - \widetilde{\mathbf{b}}^c(z), \quad\quad (2)$$

together with the associated covariances,

$$\Sigma_{\widetilde{\mathbf{A}}}^c = \int_{\mathsf{Z}} \widetilde{\mathbf{A}}^c(z)\widetilde{\mathbf{A}}^c(z)^\top \mathrm{d}\pi_c(z), \ \Sigma_\varepsilon^c = \int_{\mathsf{Z}} \varepsilon^c(z)\varepsilon^c(z)^\top \mathrm{d}\pi_c(z), \ \Sigma_\omega^c = \int_{\mathsf{Z}} \omega^c(z)\omega^c(z)^\top \mathrm{d}\pi_c(z), \ (3)$$

that are finite whenever one of the following assumptions on the $\{Z_{t,h}^c\}_{t,h\geq 0}$ hold.

**A1.** *For each agent $c$, $(Z_k^c)_{k\in\mathbb{N}}$ are i.i.d. random variables with values in $(\mathsf{Z}, \mathcal{Z})$ and distribution $\pi_c$ satisfying $\mathbb{E}_{\pi_c}[\mathbf{A}^c(Z_k^c)] = \bar{\mathbf{A}}^c$ and $\mathbb{E}_{\pi_c}[\mathbf{b}(Z_k^c)] = \bar{\mathbf{b}}^c$, and we define $\mathrm{C}_{\mathbf{A}} = \sup_c \|\bar{\mathbf{A}}^c\|$.*

**A2.** *For each $c \in [N]$, $(Z_k^c)_{k\in\mathbb{N}}$ is a Markov chain with values in $(\mathsf{Z}, \mathcal{Z})$, with Markov kernel $\mathrm{P}_c$. The kernel $\mathrm{P}_c$ admits a unique invariant distribution $\pi_c$, $Z_0^c \sim \pi_c$, and $\mathrm{P}_c$ is uniformly geometrically ergodic, that is, there exist $\tau_{\mathrm{mix}}(c) \in \mathbb{N}$, such that for any $k \in \mathbb{N}$,*

$$\sup_{z,z'\in\mathsf{Z}} (1/2)\|\mathrm{P}_c^k(\cdot|z) - \mathrm{P}_c^k(\cdot|z')\|_{\mathsf{TV}} \le (1/4)^{\lfloor k/\tau_{\mathrm{mix}}(c)\rfloor} ,$$

*and for $c \in [N]$, we have $\mathbb{E}_{\pi_c}[\mathbf{A}^c(Z_1^c)] = \bar{\mathbf{A}}^c$ and $\mathbb{E}_{\pi_c}[\mathbf{b}(Z_1^c)] = \bar{\mathbf{b}}^c$, and we define*

$$\|\varepsilon\|_\infty = \max_{c\in[N]} \sup_{z\in\mathsf{Z}} \|\varepsilon^c(z)\| < \infty , \quad \mathrm{C}_{\mathbf{A}} = \max_{c\in[N]} \sup_{z\in\mathsf{Z}} \|\mathbf{A}^c(z)\| < \infty .$$

*Moreover, each of the matrices $-\bar{\mathbf{A}}^c$ is Hurwitz.*

In A2, random matrices $\mathbf{A}^c(z)$ and noise variables $\varepsilon^c(z)$ are almost surely bounded. This is necessary for working with the uniformly geometrically ergodic Markov kernels $\mathrm{P}_c$. For simplicity, we state most of our results using A1, which is classical in finite-time studies of LSA [46, 14]. Nonetheless, we show that our analysis of FedLSA can be extended to the Markovian setting under A2.

In a federated environment, agents can only communicate via a central server, which is generally costly. Hence, in FedLSA, agents' local updates are only aggregated after a given time. During the round $t \ge 0$, the agents start with a shared value $\theta_t$ and perform $H > 0$ local updates, for $h = 1$ to $H$, given by the recurrence

$$\theta_{t,h}^c = \theta_{t,h-1}^c - \eta(\mathbf{A}^c(Z_{t,h}^c)\theta_{t,h-1}^c - \mathbf{b}^c(Z_{t,h}^c)) , \tag{4}$$

with $\theta_{t,0}^c = \theta_t$, and where we use the alias $Z_{t,h}^c = Z_{Ht+h}$ to simplify notations. Agents then send $\theta_{t,H}$ to the server, that aggregates them as $\theta_t = N^{-1}\sum_{c=1}^N \theta_{t-1,H}^c$ and sends it back to all agents. We summarize this procedure in Algorithm 1. Our next assumption, which holds whenever $\bar{\mathbf{A}}^c$ is Hurwitz [19, 39, 15], ensures the stability of the local updates.

**A3.** *There exist $a > 0$, $\eta_\infty > 0$, such that $\eta_\infty a \le 1/2$, and for $\eta \in (0; \eta_\infty)$, $c \in [N]$, $u \in \mathbb{R}^d$, it holds for $Z_0^c \sim \pi_c$, that $\mathbb{E}^{1/2}[\|(\mathrm{I} - \eta\mathbf{A}^c(Z_0^c))u\|^2] \le (1 - \eta a)\|u\|$.*

### 3.2 Federated Temporal Difference Learning

A major application of FedLSA is federated TD learning with linear function approximation. Consider $N$ Markov Decision Processes $\{(\mathcal{S}, \mathcal{A}, \mathbb{P}_{\mathrm{MDP}}^c, r^c, \gamma)\}_{c\in[N]}$ with shared state space $\mathcal{S}$, action space $\mathcal{A}$, and discounting factor $\gamma \in (0, 1)$. Each agent $c \in [N]$ has its own transition kernel $\mathbb{P}_{\mathrm{MDP}}^c$, where $\mathbb{P}_{\mathrm{MDP}}^c(\cdot|s, a)$ specifies the transition probability from state $s$ upon taking action $a$ for this specific agent, as well as its own reward function $r^c : \mathcal{S} \times \mathcal{A} \to [0, 1]$, that we assume to be deterministic for simplicity. Agents' heterogeneity lies in the different transition kernels and reward functions, that are *specific to each agent*.

In federated TD learning, all agents use the same shared policy $\pi$, and aim to construct a single shared function, that simultaneously approximates all value functions, defined as, for $s \in \mathcal{S}$ and $c \in [N]$,

$$V^{c,\pi}(s) = \mathbb{E}\Big[\sum_{k=0}^\infty \gamma^k r^c(S_k^c, A_k^c)\Big] , \text{ with } S_0^c = s, A_k^c \sim \pi(\cdot|S_k^c), \text{ and } S_{k+1}^c \sim \mathbb{P}_{\mathrm{MDP}}^c(\cdot|S_k^c, A_k^c).$$

In the following, we aim to approximate $V^{c,\pi}(s)$ as a linear combination of features built using a mapping $\varphi : \mathcal{S} \to \mathbb{R}^d$. Formally, we look for $\theta \in \mathbb{R}^d$ such that the function $\mathcal{V}_\theta(s) = \varphi^\top(s)\theta$ properly estimate the true value. For $c \in [N]$, we denote $\mu^c$ the invariant distribution over $\mathcal{S}$ induced by the policy $\pi$ and transition kernel $\mathbb{P}_{\mathrm{MDP}}^c$ of agent $c$. Our goal is to find a parameter $\theta_\star^c$ which is defined as a unique solution to the projected Bellman equation, see [49], which defines the best linear approximation of $V^{c,\pi}$. This problem can be cast as a federated LSA problem [42, 50] by viewing the local optimum parameter $\theta_\star^c$ as the solution of the system $\bar{\mathbf{A}}^c\theta_\star^c = \bar{\mathbf{b}}^c$ , where

$$\bar{\mathbf{A}}^c = \mathbb{E}_{s\sim\mu^c, s'\sim P^{\pi,c}(\cdot|s)}[\phi(s)\{\phi(s)-\gamma\phi(s')\}^\top] , \quad \text{and} \quad \bar{\mathbf{b}}^c = \mathbb{E}_{s\sim\mu^c, a\sim\pi(\cdot|s)}[\phi(s)r^c(s,a)] . \tag{5}$$

The global optimal parameter is then defined as the solution $\theta_\star$ of the averaged system $(\frac{1}{N}\sum_{c=1}^N \bar{\mathbf{A}}^c)\theta_\star = \frac{1}{N}\sum_{c=1}^N \bar{\mathbf{b}}^c$. As it is the case for federated LSA, this parameter may give a better overall estimation of the value function. Indeed, the distribution $\mu^c$ of some agents may be

strongly biased towards some states, whereas obtaining an estimation that is more balanced across all states may be more relevant.

In practice, when computing value functions, the tuples $\{(S_k^c, A_k^c, S_{k+1}^c)\}_{k \in \mathbb{N}}$ are sampled along one of the two following rules.

**TD** 1. $(S_k^c, A_k^c, S_{k+1}^c)$ *are generated i.i.d. with* $S_k^c \sim \mu^c$, $A_k^c \sim \pi(\cdot|S_k^c)$, $S_{k+1}^c \sim \mathbb{P}_{MDP}^c(\cdot|S_k^c, A_k^c)$ .

**TD** 2. $(S_k^c, A_k^c, S_{k+1}^c)$ *are generated sequentially with* $A_k^c \sim \pi(\cdot|S_k^c)$, $S_{k+1}^c \sim \mathbb{P}_{MDP}^c(\cdot|S_k^c, A_k^c)$ .

The generative model assumption **TD** 1 is common in TD learning [8, 30, 42, 45]. It is possible to generalize all our results to the more general Assumption **TD** 2, sampling over a single trajectory and leveraging the Markovian noise dynamics. This would have a similar impact on our results on TD(0) as it has on the ones we will present for general **FedLSA** in Section 4. In our analysis, we require the following assumption on the feature design matrix $\Sigma_\varphi^c = \mathbb{E}_{\mu^c}[\varphi(S_0^c)\varphi(S_0^c)^\top] \in \mathbb{R}^{d \times d}$.

**TD** 3. *Matrices* $\Sigma_\varphi^c$ *are non-degenerate with the minimal eigenvalue* $\nu = \min_{c \in [N]} \lambda_{\min}(\Sigma_\varphi^c) > 0$. *Moreover, the feature mapping* $\varphi(\cdot)$ *satisfies* $\sup_{s \in \mathcal{S}} \|\varphi(s)\| \leq 1$.

This assumption ensures the uniqueness of the optimal parameter $\theta_\star^c$. Under **TD** 1 and **TD** 3 we check the LSA assumptions A1 and A3, and the following holds.

**Claim 3.1.** *Assume TD 1 and TD 3. Then the sequence of TD(0) updates satisfies A1 and A3 with*

$$\mathrm{C_A} = 1 + \gamma , \qquad \|\Sigma_{\widetilde{\mathbf{A}}}^c\| \leq 2(1+\gamma)^2 , \qquad \mathrm{Tr}(\Sigma_\varepsilon^c) \leq 2(1+\gamma)^2 \left(\|\theta_\star^c\|^2 + 1\right) ,$$

$$a = \tfrac{(1-\gamma)\nu}{2} , \qquad \eta_\infty = \tfrac{(1-\gamma)}{4} .$$

We prove this claim in Appendix E.1, and refer to [45, 42] for more details on the link between TD and linear stochastic approximation.

## 4 Refined Analysis of the **FedLSA** Algorithm

### 4.1 Stochastic expansion for **FedLSA**

We use the error expansion framework [1, 14] for LSA to analyze the MSE of the estimates $\theta_t$ generated by Algorithm 1. For this purpose, we rewrite local update (4) as $\theta_{t,h}^c - \theta_\star^c = (\mathrm{I} - \eta\mathbf{A}(Z_{t,h}^c))(\theta_{t,h-1}^c - \theta_\star^c) - \eta\varepsilon^c(Z_{t,h}^c)$, where $\varepsilon^c(z)$ is defined in (2). Running this recursion until the start of local training, we obtain

$$\theta_{t,H}^c - \theta_\star^c = \Gamma_{t,1:H}^{(c,\eta)}\{\theta_{t,0}^c - \theta_\star^c\} - \eta \sum_{h=1}^H \Gamma_{t,h+1:H}^{(c,\eta)}\varepsilon^c(Z_{t,h}^c) ,$$

where $\varepsilon^c(z)$ is as in (3), and we recall that $\theta_{t,0}^c = \theta_{t-1}, \forall c \in [N]$. We also introduced the notation

$$\Gamma_{t,m:n}^{(c,\eta)} = \prod_{h=m}^n (\mathrm{I} - \eta\mathbf{A}(Z_{t,h}^c)) , \quad 1 \leq m \leq n \leq H ,$$

with the convention $\Gamma_{t,m:n}^{(c,\eta)} = \mathrm{I}$ for $m > n$. Note that by A3, $\Gamma_{t,m:n}^{(c,\eta)}$ is exponentially stable. That is, for any $h \in \mathbb{N}$, we have $\mathbb{E}^{1/2}\big[\|\Gamma_{t,m:m+h}^{(c,\eta)}u\|^2\big] \leq (1-\eta a)^h\|u\|$. Using the fact $\theta_{t,0}^c = \theta_{t-1}$, and employing (1), we obtain that

$$\theta_t - \theta_\star = \bar{\Gamma}_{t,H}^{(\eta)}\{\theta_{t-1} - \theta_\star\} + \bar{\rho}_H + \bar{\tau}_{t,H} - \eta\bar{\varphi}_{t,H} , \qquad \text{with} \quad \bar{\Gamma}_{t,H}^{(\eta)} = N^{-1}\sum_{c=1}^N \Gamma_{t,1:H}^{(c,\eta)} , \quad (6)$$

where $\bar{\tau}_{t,H} = \frac{1}{N}\sum_{c=1}^N \{(\mathrm{I} - \eta\bar{\mathbf{A}}^c)^H - \Gamma_{t,1:H}^{(c,\eta)}\}\{\theta_\star^c - \theta_\star\}$, $\bar{\varphi}_{t,H} = \frac{1}{N}\sum_{c=1}^N\sum_{h=1}^H \Gamma_{t,h+1:H}^{(c,\eta)}\varepsilon^c(Z_{t,h}^c)$ are zero-mean fluctuation terms, and

$$\bar{\rho}_H = \frac{1}{N}\sum_{c=1}^N (\mathrm{I} - (\mathrm{I} - \eta\bar{\mathbf{A}}^c)^H)\{\theta_\star^c - \theta_\star\}$$

is the deterministic heterogeneity bias accumulated in one round of local training. Note that $\bar{\rho}_H$ vanishes when either (i) agents are homogeneous, or (ii) number of local updates is $H = 1$. To analyze **FedLSA**, we run the recurrence (6) to obtain the decomposition

$$\theta_t - \theta_\star = \tilde{\theta}_t^{(\mathrm{tr})} + \tilde{\theta}_t^{(\mathrm{bi,bi})} + \tilde{\theta}_t^{(\mathrm{fl})} . \qquad (7)$$

Here $\tilde{\theta}_t^{(\mathrm{tr})} = \prod_{s=1}^t \bar{\Gamma}_{s,H}^{(\eta)}\{\theta_0 - \theta_\star\}$ is a transient term that vanishes geometrically, $\tilde{\theta}_t^{(\mathrm{fl})}$ is a zero-mean fluctuation term, with detailed expression provided in Appendix A, and the term $\tilde{\theta}_t^{(\mathrm{bi,bi})}$ is

$$\tilde{\theta}_t^{(\mathrm{bi,bi})} = \sum_{s=1}^t (\bar{\Gamma}_H^{(\eta)})^{t-s}\bar{\rho}_H , \quad \text{where} \quad \bar{\Gamma}_H^{(\eta)} = \mathbb{E}[\bar{\Gamma}_{s,H}^{(\eta)}] ,$$

and accounts for the bias of **FedLSA** due to local training, that vanishes whenever $\bar{\rho}_H = 0$.

## 4.2 Convergence rate of FedLSA for i.i.d. observation model

First, we analyze the rate at which FedLSA converges to a biased solution $\theta_\star + \tilde{\theta}_t^{(\mathrm{bi,bi})}$. The following two quantities, which stem from the heterogeneity and stochasticity of the local estimators, play a central role in this rate

$$\bar{\sigma}_\varepsilon = \mathbb{E}_c\left[\mathrm{Tr}(\Sigma_\varepsilon^c)\right] , \quad \tilde{v}_{\mathrm{heter}} = \mathbb{E}_c\left[\|\Sigma_{\bar{\mathbf{A}}}^c\|\,\|\theta_\star^c - \theta_\star\|^2\right] .$$

Here $\bar{\sigma}_\varepsilon$ and $\tilde{v}_{\mathrm{heter}}$ correspond to the different sources of noise in the error decomposition (7). The term $\bar{\sigma}_\varepsilon$ is related to the variance of the *local* LSA iterate on each of the agents, while $\tilde{v}_{\mathrm{heter}}$ controls the bias fluctuation term. In the centralized setting (i.e. if $N = 1$), the $\tilde{v}_{\mathrm{heter}}$ term disappears, but not the $\bar{\sigma}_\varepsilon$ term. We now proceed to analyze the MSE of the iterates of FedLSA:

**Theorem 4.1.** *Assume A1 and A3. Then for any step size $\eta \in (0, \eta_\infty)$ it holds that*

$$\mathbb{E}^{1/2}\left[\|\theta_t - \tilde{\theta}_t^{(\mathrm{bi,bi})} - \theta_\star\|^2\right] \lesssim \sqrt{\frac{\eta\tilde{v}_{\mathrm{heter}}}{aN}} + \sqrt{\frac{\eta\bar{\sigma}_\varepsilon}{aN}} + \sqrt{\frac{\mathbb{E}_c[\|\Sigma_{\bar{\mathbf{A}}}^c\|]}{HN}}\frac{\|\bar{\rho}_H\|}{a} + (1 - \eta a)^{tH}\|\theta_0 - \theta_\star\| ,$$

*where the bias $\tilde{\theta}_t^{(\mathrm{bi,bi})}$ converges in expectation to $\tilde{\theta}_\infty^{(\mathrm{bi,bi})} = (\mathrm{I} - \bar{\Gamma}_H^{(\eta)})^{-1}\bar{\rho}_H$ at a geometric rate, and is uniformly bounded by $\mathbb{E}^{1/2}[\|\tilde{\theta}_t^{(\mathrm{bi,bi})}\|^2] \lesssim \frac{\eta H\mathbb{E}_c[\|\theta_\star^c - \theta_\star\|]}{a}$.*

The proof of Theorem 4.1 relies on bounding each term from (7). We provide a proof with explicit constants in Appendix A. Importantly, the fluctuation terms scale linearly with $N$. Moreover, in the centralized setting (that is, $N = 1$), the bias terms $\bar{\rho}_H$, $\tilde{\theta}_t^{(\mathrm{bi,bi})}$ and $\tilde{v}_{\mathrm{heter}}$ vanish in Theorem 4.1, yielding the last-iterate bound

$$\mathbb{E}^{1/2}\left[\|\theta_t - \theta_\star\|^2\right] \lesssim \sqrt{\frac{\eta\bar{\sigma}_\varepsilon}{a}} + (1 - \eta a)^{tH}\|\theta_0 - \theta_\star\| ,$$

which is known to be sharp in its dependence on $\eta$ for single-agent LSA (see Theorem 5 in [15]). Based on Claim 3.1, Theorem 4.1 translates for federated TD(0) as follows.

**Corollary 4.2.** *Assume TD 1 and TD 3. Then for any step size $\eta \in (0, \frac{1-\gamma}{4})$, the iterates of federated TD(0) satisfy, with $\chi(\theta_\star, \theta_\star^1, \ldots, \theta_\star^N) = \mathbb{E}_c[\|\theta_\star^c - \theta_\star\|^2] \vee (1 + \mathbb{E}_c[\|\theta_\star^c\|^2])$,*

$$\mathbb{E}^{1/2}\left[\|\theta_t - \tilde{\theta}_t^{(\mathrm{bi,bi})} - \theta_\star\|^2\right] \lesssim \sqrt{\frac{\eta\chi(\theta_\star, \theta_\star^1, \ldots, \theta_\star^N)}{(1-\gamma)\nu N}} + \sqrt{\frac{1}{HN}}\frac{\|\bar{\rho}_H\|}{(1-\gamma)\nu} + \left(1 - \frac{\eta(1-\gamma)\nu}{2}\right)^{tH}\|\theta_0 - \theta_\star\| .$$

The right-hand side of Corollary 4.2 scales linearly with $N$, allowing for linear speed-up. This is in line with recent results on federated TD(0), which shows linear speed-up either without local training [7] or up to a possibly large bias term [50] (see analysis of their Theorem 2). While Corollary 4.2 shows the algorithm's convergence to some fixed, biased value, one can set the parameters of FedLSA such that this bias is small. This allows to rewrite the result of Theorem 4.1 in order to get a sample complexity bound in the following form.

**Corollary 4.3.** *Assume A1 and A3. Let $H > 1$, and $0 < \epsilon < \frac{\left(\sqrt{\tilde{v}_{\mathrm{heter}} \vee \bar{\sigma}_\varepsilon}\mathbb{E}_c[\|\theta_\star^c - \theta_\star\|]\right)^{2/5}}{a} \vee \frac{\mathbb{E}_c[\|\theta_\star^c - \theta_\star\|]}{a\,C_{\mathbf{A}}}$. Set the step size $\eta = \mathcal{O}\left(\frac{aN\epsilon^2}{\tilde{v}_{\mathrm{heter}} \vee \bar{\sigma}_\varepsilon} \wedge \eta_\infty\right)$ and the number of local steps $H = \mathcal{O}\left(\frac{\tilde{v}_{\mathrm{heter}} \vee \bar{\sigma}_\varepsilon}{\mathbb{E}_c[\|\theta_\star^c - \theta_\star\|]}\frac{1}{N\epsilon}\right)$. Then, to achieve $\mathbb{E}\left[\|\theta_T - \theta_\star\|^2\right] < \epsilon^2$ the required number of communications for federated LSA is*

$$T = \mathcal{O}\left(\left(\frac{1}{a\eta_\infty} \vee \frac{\mathbb{E}_c[\|\theta_\star^c - \theta_\star\|]}{a^2\epsilon}\right)\log\frac{\|\theta_0 - \theta_\star\|}{\epsilon}\right) .$$

In Corollary 4.3, the number of oracle calls scales as $TH = \mathcal{O}\left(\frac{\tilde{v}_{\mathrm{heter}} \vee \bar{\sigma}_\varepsilon}{Na^2\epsilon^2}\log\frac{\|\theta_0 - \theta_\star\|}{\epsilon}\right)$, which shows that FedLSA has linear speed-up. Importantly, the number of communications $T$ required to achieve precision $\epsilon^2$ scales as $\epsilon^{-1}$. In the next section, we will show how this dependence on $\epsilon^{-1}$ can be reduced from polynomial to logarithmic. Now we state the communication bound of federated TD(0).

**Corollary 4.4.** *Assume TD 1 and TD 3. Then for any $0 < \epsilon < \frac{g_1(\theta_\star^c, \theta_\star)}{(1-\gamma)\nu}$ with $g_1 = \mathcal{O}((1 + \|\theta_\star\|)\mathbb{E}_c[\|\theta_\star^c - \theta_\star\|])$. Set $\eta = \mathcal{O}\left(\frac{(1-\gamma)\nu N\epsilon^2}{\mathbb{E}_c[\|\theta_\star^c\|^2] + 1}\right)$ and $H = \mathcal{O}\left(\frac{\mathbb{E}_c[\|\theta_\star^c\|^2] + 1}{N\epsilon\mathbb{E}_c[\|\theta_\star^c - \theta_\star\|^2]}\right)$. Then, to achieve $\mathbb{E}\left[\|\theta_T - \theta_\star\|^2\right] < \epsilon^2$, the required number of communications for federated TD(0) is*

$$T = \mathcal{O}\left(\left(\frac{1}{(1-\gamma)^2\nu} \vee \frac{\mathbb{E}_c[\|\theta_\star^c - \theta_\star\|]}{(1-\gamma)^2\nu^2\epsilon}\right)\log\frac{\|\theta_0 - \theta_\star\|}{\epsilon}\right) .$$

Corollary 4.4 is the first result to show that, even with local training and heterogeneous agents, federated TD(0) can converge to $\theta_\star$ with arbitrary precision. Importantly, this result preserves the *linear speed-up effect*, showing that federated learning indeed accelerates the training.

## 4.3 Convergence of FedLSA under Markovian observations model

The analysis of FedLSA can be generalized to the setting where observations $\{Z_k^c\}_{k\in\mathbb{N}}$ form a Markov chain with kernel $P_c$. To handle the Markovian nature of observations, we propose a variant of FedLSA that skips some observations (see the full procedure in Appendix B). This follows classical schemes for Markovian data in optimization [40], as adjusting the number of skipped observations (keeping about 1 observation out of $\tau_{\mathrm{mix}}(c)$) allows to control the correlation of successive observations. We may now state the counterpart of Corollary 4.3 for the Markovian setting.

**Corollary 4.5** (Corollary 4.3 adjusted to the Markov samples). *Assume A2 and A3 and let* $0 < \epsilon < \frac{\left(\sqrt{\tilde{v}_{heter}\vee\bar{\sigma}_\varepsilon}\mathbb{E}_c[\|\theta_\star^c-\theta_\star\|]\right)^{2/5}}{a} \vee \frac{\mathbb{E}_c[\|\theta_\star^c-\theta_\star\|]}{a\,\mathrm{C_A}}$. *Set the step size* $\eta = \mathcal{O}\left(\frac{aN\epsilon^2}{\tilde{v}_{heter}\vee\bar{\sigma}_\varepsilon} \wedge \eta_\infty \wedge \eta_\infty^{(\mathrm{M})}\right)$, *where we give the expression of* $\eta_\infty^{(\mathrm{M})}$ *is* (37). *Then, for the iterates of Algorithm 3, in order to achieve* $\mathbb{E}\left[\|\theta_T-\theta_\star\|^2\right] \le \epsilon^2$, *the required number of communication is*

$$T = \mathcal{O}\left(\left(\frac{1}{a\eta_\infty} \vee \frac{\mathbb{E}_c[\|\theta_\star^c-\theta_\star\|]}{a^2\epsilon}\right)\log\frac{\|\theta_0-\theta_\star\|}{\epsilon}\right)\,,$$

*where the number of local updates $H$ satisfies*

$$\frac{H}{\log H} = \mathcal{O}\left(\frac{\tilde{v}_{heter}\vee\bar{\sigma}_\varepsilon}{\mathbb{E}_c[\|\theta_\star^c-\theta_\star\|]}\frac{\max_c\tau_{\mathrm{mix}}(c)\log\left(NT^3(\|\theta_0-\theta_\star\|+2\mathbb{E}_c[\|\theta_\star^c-\theta_\star\|]+\eta\|\varepsilon\|_\infty)/\epsilon^2\right)}{N\epsilon}\right)\,.$$

The proof of Corollary 4.3 follows the idea outlined in [40], using Berbee's lemma [10]. We give all the details in Appendix B. This result is very similar to Corollary 4.3. Most crucially, it shows that the communication complexity is the same, regardless of the type of noise. The differences with Corollary 4.3 lie in (i) the number local updates $H$, that is scaled by $\tau_{\mathrm{mix}}$ (up to logarithmic factors), and (ii) the additional condition $\eta \le \eta_\infty^{(\mathrm{M})}$, that allows verifying the stability of random matrix products with Markovian dependence (see Lemma B.2 in the appendix).

*Remark* 4.6. Although, for clarity of exposition, we only state the counterpart of Corollary 4.3 in the Markovian result, all of our results can be extended to Markovian observations using the same ideas.

## 5 SCAFFLSA: Federated LSA with Bias Correction

### 5.1 Stochastic Controlled Averaging for Federated LSA

We now introduce the *Stochastic Controlled Averaging for Federated LSA* algorithm (SCAFFLSA), an improved version of FedLSA that mitigates client drift using control variates. This method is inspired by *Scaffnew* (see 37). In SCAFFLSA, each agent $c \in [N]$ keeps a local variable $\xi_t^c$, that remains constant during each communication round $t$. Agents perform local updates on the current estimates of the parameters $\hat{\theta}_{t,0}^c = \theta_t$ for $c \in [N]$, and for $h \in [H]$,

$$\hat{\theta}_{t,h}^c = \hat{\theta}_{t,h-1}^c - \eta(\mathbf{A}^c(Z_{t,h}^c)\hat{\theta}_{t,h-1}^c - \mathbf{b}^c(Z_{t,h}^c) - \xi_t^c)\,,$$

At the end of the round, (i) the agents communicate the current estimate to the central server, (ii) the central server averages local iterates, and (iii) agents update their local control variates; see Algorithm 2. By defining the *ideal* control variates at the global solution, given by $\xi_\star^c = \bar{\mathbf{A}}^c\theta_\star - \bar{\mathbf{b}}^c = \bar{\mathbf{A}}^c(\theta_\star - \theta_\star^c)$, we can rewrite the local update as

$$\hat{\theta}_{t,h}^c - \theta_\star = (\mathrm{I} - \eta\mathbf{A}^c(Z_{t,h}^c))(\hat{\theta}_{t,h-1}^c - \theta_\star) + \eta(\xi_t^c - \xi_\star^c) - \eta\omega^c(Z_{t,h}^c)\,, \tag{8}$$

where $\omega^c(z)$ is defined in (2). Under A1, it has finite covariance $\Sigma_\omega^c = \int_\mathsf{Z}\omega^c(z)\omega^c(z)^\top\mathrm{d}\pi_c(z)$.

Similarly to the analysis of FedLSA, we use (8) to describe the sequence of aggregated iterates and control variates as, for $t \ge 0$ and $c \in [N]$,

$$\begin{aligned}\theta_{t+1} - \theta_\star &= \bar{\Gamma}_{t,H}^{(\eta)}(\theta_t - \theta_\star) + \tfrac{\eta}{N}\textstyle\sum_{c=1}^N C_{t+1}^c(\xi_t^c - \xi_\star^c) - \eta\bar{\omega}_{t+1}\,,\\ \xi_{t+1}^c - \xi_\star^c &= \xi_t^c - \xi_\star^c + \tfrac{1}{\eta H}(\theta_{t+1} - \theta_{t,H})\,,\end{aligned} \tag{9}$$

where $C_{t+1}^c = \sum_{h=1}^H \Gamma_{t,h+1:H}^{(c,\eta)}$ and $\bar{\omega}_{t+1} = \frac{1}{N}\sum_{c=1}^N\sum_{h=1}^H \Gamma_{t,h+1:H}^{(c,\eta)}\omega^c(Z_{t,h}^c)$. We now state the convergence rate, as well as sample and communication complexity of Algorithm 2.

**Algorithm 2** SCAFFLSA: Stochastic Controlled FedLSA with deterministic communication

---

**Input:** $\eta > 0, \theta_0, \xi_0^c \in \mathbb{R}^d, T, N, H$
**for** $t = 1$ to $T$ **do**
   **for** $c = 1$ to $N$ **do**
      Set $\hat{\theta}_{t,0}^c = \theta_t$
      **for** $h = 1$ to $H$ **do**
         Receive $Z_{t,h}^c$ and perform local update $\hat{\theta}_{t,h}^c = \hat{\theta}_{t,h-1}^c - \eta(\mathbf{A}^c(Z_{t,h}^c)\hat{\theta}_{t,h-1}^c - \mathbf{b}^c(Z_{t,h}^c) - \xi_t^c)$
   Aggregate local iterates: $\theta_{t+1} = \frac{1}{N}\sum_{c=1}^N \hat{\theta}_{t,H}^c$
   Update local control variates: $\xi_{t+1}^c = \xi_t^c + \frac{1}{\eta H}(\theta_{t+1} - \hat{\theta}_{t,H}^c)$

---

**Theorem 5.1.** *Assume A1, A3. Let $\eta, H > 0$ such that $\eta \leq \eta_\infty$, and $H \leq {}^a/240\eta\left\{\mathrm{C}_{\mathbf{A}}^2 + \|\Sigma_{\mathbf{A}}^c\|\right\}$. Set $\xi_0^c = 0$ for all $c \in [N]$. Then we have*

$$\mathbb{E}[\|\theta_T - \theta_\star\|^2] \lesssim \frac{\eta}{Na}\|\Sigma_\omega\| + \left(1 - \frac{\eta a H}{2}\right)^T\left\{\|\theta_0 - \theta_\star\|^2 + \eta^2 H^2 \mathbb{E}_c[\|\bar{\mathbf{A}}^c(\theta_\star^c - \theta_\star)\|^2]\right\}.$$

**Corollary 5.2.** *Let $\epsilon > 0$. Set the step size $\eta = \mathcal{O}(\min(\eta_\infty, {}^{Na\epsilon}/\bar{\sigma}))$ and the number local updates to $H = \mathcal{O}\left(\max\left(\frac{a}{\eta(\mathrm{C}+\|\Sigma\cdot\|)}, \frac{\|\Sigma\|}{N\epsilon(\mathrm{C}+\|\Sigma\cdot\|)}\right)\right)$. Then, to achieve $\mathbb{E}[\|\theta_T - \theta_\star\|^2] \leq \epsilon^2$, the required number of communication for* SCAFFLSA *is*

$$T = \mathcal{O}\left(\frac{\mathrm{C}+\|\Sigma\|}{a}\log\left(\frac{\|\theta-\theta\|+\mathbb{E}[\|\bar{\mathbf{A}}(\theta-\theta)\|]a/\mathrm{C}}{\epsilon}\right)\right).$$

We provide detailed proof of these statements in Appendix C. They are based on a novel analysis, where we study virtual parameters $\check{\theta}_{t,h}^c$, that follow the same update as (8), without the last term $\eta\omega^c(Z_{t,h}^c)$. After each round, virtual parameters are aggregated, and virtual control variate updated as

$$\check{\theta}_{t+1} - \theta_\star = \frac{1}{N}\sum_{c=1}^N \check{\theta}_{t,H}^c, \text{ and } \check{\xi}_{t+1}^c - \xi_\star^c = \check{\xi}_t^c - \xi_\star^c + \frac{1}{\eta H}(\check{\theta}_{t+1} - \check{\theta}_{t,H}).$$

This allows to decompose

$$\theta_t - \theta_\star = \check{\theta}_t - \theta_\star + \widetilde{\theta}_t, \text{ and } \xi_t^c - \xi_\star^c = \check{\xi}_t^c - \xi_\star^c + \widetilde{\xi}_t^c,$$

where $\check{\theta}_t - \theta_\star$ and $\check{\xi}_t^c - \xi_\star^c$ are transient terms, and $\widetilde{\theta}_t = \theta_t - \check{\theta}_t$ and $\widetilde{\xi}_t^c = \xi_t^c - \check{\xi}_t^c$ capture the fluctuations of the parameters and control variates.

We stress that our analysis shows that, in comparison with FedLSA, the SCAFFLSA algorithm reduces communication complexity while preserving the *linear speed-up in the number of agents*. This is in stark contrast with existing analyses of control-variate methods in heterogeneous federated learning, that either have large communication cost, or lose the linear speed-up [24, 37, 21]. To obtain this result, we conduct a very careful analysis of the propagation of variances and covariances of $\widetilde{\theta}_t$ and $\widetilde{\xi}_t^c$ between successive communication rounds. We describe this in full detail in Appendix C.2.

In Corollary 5.2, we show that the total number of communications depends only logarithmically on the precision $\epsilon$. This is in stark contrast with Algorithm 1, where the necessity of controlling the bias' magnitude prevents from scaling $H$ with ${}^1/\epsilon$. Additionally, this shows that the number of required local updates reduces as the number of agents grows. Thus, in the high precision regime (i.e. small $\epsilon$ and $\eta$), using control variates reduces communication complexity compared to FedLSA.

### 5.2 Application to Federated TD(0)

Applying SCAFFLSA to TD learning, we obtain SCAFFTD(0) (see Algorithm 5 in Appendix E). The analysis of SCAFFLSA directly translates to SCAFFTD(0), resulting in the following communication complexity bound.

**Corollary 5.3.** *Assume TD 1 and TD 3 and let $0 < \epsilon \leq \sqrt{8\mathbb{E}_c[1 + \|\theta_\star^c\|^2]/((1-\gamma)\nu)}$. Set the step size $\eta = \mathcal{O}(\frac{(1-\gamma)\nu N\epsilon}{\|\theta\|+1})$ and the number local updates to $H = \mathcal{O}(\frac{\|\theta\|+1}{N\epsilon})$. Then, to achieve $\mathbb{E}[\|\theta_T - \theta_\star\|^2] \leq \epsilon^2$, the required number of communication for* SCAFFTD(0) *is*

$$T = \mathcal{O}\left(\frac{1}{(1-\gamma)\nu}\log\left(\frac{\|\theta-\theta\|+(1-\gamma)\nu\mathbb{E}[\|\theta-\theta\|]}{\epsilon}\right)\right).$$

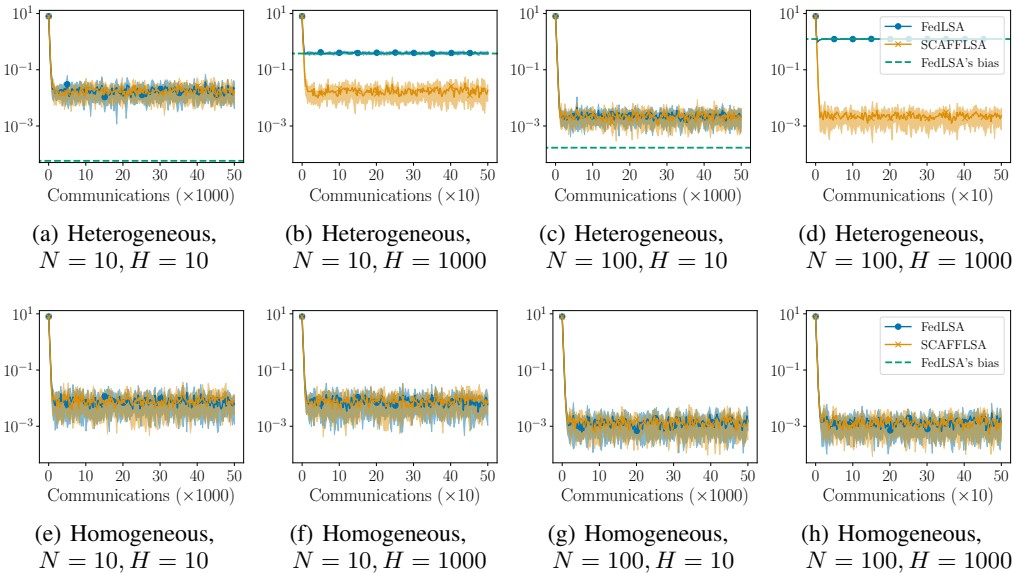

Figure 1: MSE as a function of the number of communication rounds for FedLSA and SCAFFLSA applied to federated TD(0) in homogeneous and heterogeneous settings, for different number of agents and number of local steps. Green dashed line is FedLSA's bias, as predicted by Theorem 4.1. For each algorithm, we report the average MSE and variance over 5 runs.

Corollary 5.3 confirms that, when applied to TD(0), SCAFFLSA's communication complexity depends only logarithmically on heterogeneity and on the desired precision. In contrast with existing methods for federated TD(0) [11, 22, 50], it converges even with many local steps, whose number diminishes linearly with the number of agents $N$, producing the linear speed-up effect.

*Remark* 5.4. In Appendix F, we extend the analysis of Scaffnew [37] to the LSA setting. Their analysis does not exploit the fact that agents' estimators are not correlated, and thus lose the linear speed-up. In contrast, our novel analysis technique carefully tracks correlations between parameters and control variates throughout the run of the algorithm.

## 6 Numerical Experiments

In this section, we demonstrate the performance of FedLSA and SCAFFLSA under varying levels of heterogeneity. We consider the Garnet problem [2, 16], with $n = 30$ states embedded in $d = 8$ dimensions, $a = 2$ actions, and each state is linked to $b = 2$ others in the transition kernel. We aim to estimate the value function of the policy which chooses actions uniformly at random, in homogeneous and heterogeneous setups. In all experiments, we initialize the algorithms in a neighborhood of the solution, allowing to observe both transient and stationary regimes. We provide all details regarding the experimental setup in Appendix G. Our code is available either as supplementary material or online on GitHub: `https://github.com/pmangold/scafflsa`.

**SCAFFLSA properly handles heterogeneity.** This heterogeneous scenario is composed of two different Garnet environments, that are each held by half of the agents, with small perturbations. Such a setting may arise in cases where each agent's environment reflects only a part of the world. For instance, if half of the individuals live in the city, while the other half live in the countryside: both have different observations, but learning a shared value function gives a better representation of the overall reality. In Figures 1(a) to 1(d), we plot the MSE with $N \in \{10, 100\}$, $H \in \{10, 1000\}$ and $\eta = 0.1$, with the same total number of updates $TH = 500,000$. As predicted by our theory, FedLSA stalls when the number of local updates increases, and its bias (green dashed line in Figures 1(a) to 1(d)) is in line with the value predicted by our theory (see Theorem 4.1). For completeness, we plot the error of FedLSA in estimating $\theta_\star + \tilde{\theta}_\infty^{(\text{bi,bi})}$ in Appendix G. On the opposite, SCAFFLSA's bias-correction mechanism allows to eliminate all bias, improving the MSE until noise dominates.

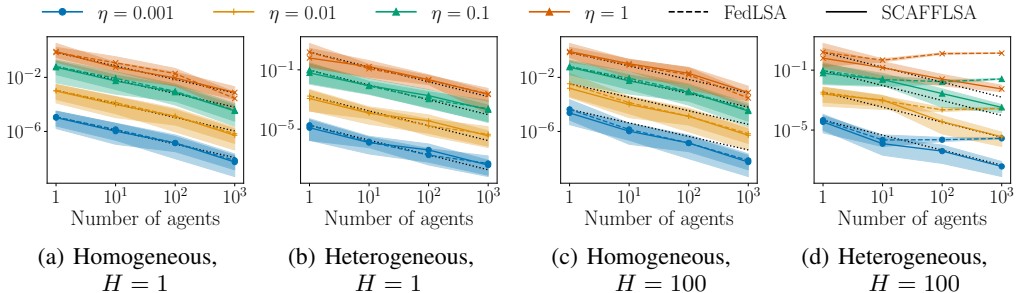

Figure 2: MSE, averaged over 10 runs, for last iterates of FedLSA (dashed lines) and SCAFFLSA (solid lines) in the stationary regime, as a function of the number of agents, in different federated TD(0) problems. The black dotted line decreases in $1/N$, serving as a visual guide for linear speed-up.

**Both algorithms behave alike in homogeneous settings.** In the homogeneous setting, we create *one* instance of a Garnet environment. Then, each agent receives a slightly perturbed variant of this environment. This illustrates a situation where all agents solve the same exact problem, but may have small divergences in their measures of states and rewards. We plot the MSE in Figures 1(e) to 1(h) with $N \in \{10, 100\}$ agents, $\eta = 0.1$, and $H \in \{10, 1000\}$, with the same total number of updates $TH = 500,000$. In this case, as predicted in Corollary 4.3, the number of local steps $H$ has little influence on the final MSE. Since agents are homogeneous, control variates have virtually no effect, and SCAFFLSA is on par with FedLSA. The MSE is dominated by the noise term, which diminishes with the step size (see additional experiments in Appendix G with smaller $\eta = 0.01$).

**Both algorithms enjoy linear speed-up!** In Figure 2, we plot the MSE obtained once algorithms reach the stationary regime, as a function of the number of agents $N = 1$ to 1000, for step sizes $\eta \in \{0.001, 0.01, 0.1, 1\}$ and $H \in \{1, 100\}$, in both homogeneous and heterogeneous settings. Whenever (i) agents are homogeneous, or (ii) the number of local steps is small, both FedLSA and SCAFFLSA can achieve similar precision with a step size that increases with the number of agents. This allows to use larger step sizes, so as to reach a given precision level faster, resulting in the so-called *linear speed-up*. However, when agents are heterogeneous and the number of local updates increases, FedLSA loses the speed-up due to large bias. Remarkably, and as explained by our theory (see Corollary 5.2), SCAFFLSA maintains this speed-up even in heterogeneous settings.

## 7    Conclusion

In this paper, we studied the role of heterogeneity in federated linear stochastic approximation. We proposed a new analysis of FedLSA, where we formally characterize FedLSA's bias. This allows to show that, with proper hyperparameter setting, FedLSA (i) can converge to arbitrary precision even with local training, and (ii) enjoys linear speed-up in the number of agents. We then proposed a novel algorithm, SCAFFLSA, that uses control variates to allow for extended local training. We analyzed this method using on a novel analysis technique, and formally proved that control variates reduce communication complexity of the algorithm. Importantly, our analysis shows that SCAFFLSA preserves the linear speed-up, which is the first time that a federated algorithm provably accelerates while preserving this linear speed-up. Finally, we instantiated our results for federated TD learning, and conducted an empirical study that demonstrates the soundness of our theory in this setting.

## Acknowledgement

The work of P. Mangold and S. Labbi has been supported by Technology Innovation Institute (TII), project Fed2Learn. The work of E. Moulines has been partly funded by the European Union (ERC-2022-SYG-OCEAN-101071601). Views and opinions expressed are however those of the author(s) only and do not necessarily reflect those of the European Union or the European Research Council Executive Agency. Neither the European Union nor the granting authority can be held responsible for them. The work of I. Levin, A. Naumov and S. Samsonov was prepared within the framework of the

HSE University Basic Research Program. This research was supported in part through computational resources of HPC facilities at HSE University [29].

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

# A   Analysis of Federated Linear Stochastic Approximation

For the analysis we need to define two filtration: $\mathcal{F}_{s,h}^+ := \sigma(Z_{t,k}^c, t \geq s, k \geq h, 1 \leq c \leq N)$, corresponding to the future events, and $\mathcal{F}_{s,h}^- := \sigma(Z_{t,k}^c, t \leq s, k \leq h, 1 \leq c \leq N)$, corresponding to the preceding events. Recall that the local LSA updates are written as

$$\theta_{t,h}^c - \theta_\star^c = (I - \eta \mathbf{A}(Z_{t,h}^c))(\theta_{t,h-1}^c - \theta_\star^c) - \eta \varepsilon^c(Z_{t,h}^c) \ .$$

Performing $H$ local steps and taking average, we end up with the decomposition

$$\theta_t - \theta_\star = \bar{\Gamma}_{t,H}^{(\eta)}\{\theta_{t-1} - \theta_\star\} + \bar{\rho}_H + \bar{\tau}_{t,H} + \eta \bar{\varphi}_{t,H} \ , \tag{10}$$

where we have defined

$$\bar{\Gamma}_{t,H}^{(\eta)} = \frac{1}{N} \sum_{c=1}^N \Gamma_{t,1:H}^{(c,\eta)} \ , \tag{11}$$

$$\bar{\rho}_H = \frac{1}{N} \sum_{c=1}^N (I - (I - \eta \bar{\mathbf{A}}^c)^H)\{\theta_\star^c - \theta_\star\} \ ,$$

$$\bar{\tau}_{t,H} = \frac{1}{N} \sum_{c=1}^N \{(I - \eta \bar{\mathbf{A}}^c)^H - \Gamma_{t,1:H}^{(c,\eta)}\}\{\theta_\star^c - \theta_\star\} \ ,$$

$$\bar{\varphi}_{t,H} = -\frac{1}{N} \sum_{c=1}^N \sum_{h=1}^H \Gamma_{t,h+1:H}^{(c,\eta)} \varepsilon^c(Z_{t,h}^c) \ .$$

The *transient* term $\bar{\Gamma}_{t,H}^{(\eta)}(\theta_{t-1} - \theta_\star)$, responsible for the rate of forgetting the previous iteration error $\theta_{t-1} - \theta_\star$, and the *fluctuation* term $\eta \bar{\varphi}_{t,H}$, reflecting the oscillations of the iterates around $\theta_\star$, are similar to the ones from the standard LSA error decomposition [14]. The two additional terms in (10) reflect the *heterogeneity bias*. This bias is composed of two parts: the true bias $\bar{\rho}_H$, which is non-random, and its fluctuations $\bar{\tau}_{t,H}$. To analyze the complexity and communication complexity of FedLSA, we run the recurrence (6) to obtain

$$\theta_t - \theta_\star = \tilde{\theta}_t^{(\mathrm{tr})} + \tilde{\theta}_t^{(\mathrm{bi,bi})} + \tilde{\theta}_t^{(\mathrm{fl,bi})} + \tilde{\theta}_t^{(\mathrm{fl})} \ , \tag{12}$$

where we have defined

$$\tilde{\theta}_t^{(\mathrm{tr})} = \prod_{s=1}^t \bar{\Gamma}_{s,H}^{(\eta)}\{\theta_0 - \theta_\star\} \ , \tag{13}$$

$$\tilde{\theta}_t^{(\mathrm{bi,bi})} = \sum_{s=1}^t \left(\bar{\Gamma}_H^{(\eta)}\right)^{t-s} \bar{\rho}_H \ ,$$

$$\tilde{\theta}_t^{(\mathrm{fl,bi})} = \sum_{s=1}^t \prod_{i=s+1}^t \bar{\Gamma}_{i,H}^{(\eta)} \bar{\tau}_{s,H} + \Delta_{H,s,t}^{(\eta)} \bar{\rho}_H \ ,$$

$$\tilde{\theta}_t^{(\mathrm{fl})} = \eta \sum_{s=1}^t \prod_{i=s+1}^t \bar{\Gamma}_{i,H}^{(\eta)} \bar{\varphi}_{s,H} \ ,$$

with the notations $\bar{\Gamma}_H^{(\eta)} = \mathbb{E}[\bar{\Gamma}_{s,H}^{(\eta)}] = \frac{1}{N} \sum_{c=1}^N (I - \eta \bar{\mathbf{A}}^c)^H$ and $\Delta_{H,s,t}^{(\eta)} = \{\prod_{i=s+1}^t \bar{\Gamma}_{i,H}^{(\eta)}\} - (\bar{\Gamma}_H^{(\eta)})^{t-s}$. The first term, $\tilde{\theta}_t^{(\mathrm{tr})}$ gives the rate at which the initial error is forgotten. The terms $\tilde{\theta}_t^{(\mathrm{bi,bi})}$ and $\tilde{\theta}_t^{(\mathrm{fl,bi})}$ represent the bias and fluctuation due to statistical heterogeneity across agents. Note that in the special case where agents are homogeneous (i.e. $\bar{\mathbf{A}}^c = \bar{\mathbf{A}}$ for all $c \in [N]$), these two terms vanish. Finally, the term $\tilde{\theta}_t^{(\mathrm{fl})}$ depicts the fluctuations of $\theta_t$ around the solution $\theta_\star$. Now we need to upper bound each of the terms in decomposition (12). This is done in a sequence of lemmas below: $\tilde{\theta}_t^{(\mathrm{fl})}$ is bounded in Lemma A.1, $\tilde{\theta}_t^{(\mathrm{fl,bi})}$ in Lemma A.2, $\tilde{\theta}_t^{(\mathrm{tr})}$ in Lemma A.4, and $\tilde{\theta}_t^{(\mathrm{bi,bi})}$ in Lemma A.5. Then we combine the bounds in order to state a version of Theorem 4.1 with explicit constants in Theorem A.6.

**Lemma A.1.** *Assume A1 and A3. Then, for any step size $\eta \in (0, \eta_\infty)$ it holds*

$$\mathbb{E}\big[\|\tilde{\theta}_t^{(\mathrm{fl})}\|^2\big] \leq \frac{\eta \bar{\sigma}_\varepsilon}{aN(1 - \mathrm{e}^{-2})} \ .$$

*Proof.* We start from the decomposition (13). With the definition of $\tilde{\theta}_t^{(\mathsf{fl})}$ and $\mathbb{E}^{\mathcal{F}}\big[\{\prod_{i=s+1}^{t}\bar{\Gamma}_{i,H}^{(\eta)}\}\bar{\varphi}_{s,H}\big] = 0$, we obtain that

$$\mathbb{E}\big[\|\tilde{\theta}_t^{(\mathsf{fl})}\|^2\big] = \eta^2 \sum_{s=1}^{t}\mathbb{E}\big[\|\{\prod_{i=s+1}^{t}\bar{\Gamma}_{i,H}^{(\eta)}\}\bar{\varphi}_{s,H}\|^2\big] \ .$$

Now, using the assumption A3 and Minkowski's inequality, we obtain that

$$\mathbb{E}^{1/2}\big[\|\{\prod_{i=s+1}^{t}\bar{\Gamma}_{i,H}^{(\eta)}\}\bar{\varphi}_{s,H}\|^2\big] \leq \frac{1}{N}\sum_{c=1}^{N}\mathbb{E}^{1/2}\|\big[\bar{\Gamma}_{t,H}^{(c,\eta)}\{\prod_{i=s+1}^{t-1}\bar{\Gamma}_{i,H}^{(\eta)}\}\bar{\varphi}_{s,H}\|^2\big]$$

$$\overset{(a)}{\leq} (1-\eta a)^H\mathbb{E}^{1/2}\big[\|\{\prod_{i=s+1}^{t-1}\bar{\Gamma}_{i,H}^{(\eta)}\}\bar{\varphi}_{s,H}\|^2\big] \ . \tag{14}$$

In (a) applied A3 conditionally on $\mathcal{F}_{t-1,H}^{-}$. Hence, by induction we get from the previous formulas that

$$\mathbb{E}\big[\|\tilde{\theta}_t^{(\mathsf{fl})}\|^2\big] \leq \eta^2 \sum_{s=1}^{t}(1-\eta a)^H\mathbb{E}\big[\|\bar{\varphi}_{s,H}\|^2\big] \ . \tag{15}$$

Now we proceed with bounding $\mathbb{E}\big[\|\bar{\varphi}_{s,H}\|^2\big]$. Indeed, since the clients are independent, we get using (11) that

$$\mathbb{E}\big[\|\bar{\varphi}_{s,H}\|^2\big] = \frac{1}{N^2}\sum_{c=1}^{N}\mathbb{E}\big[\|\sum_{h=1}^{H}\Gamma_{s,h+1:H}^{(c,\eta)}\varepsilon^c(Z_{s,h}^c)\|^2\big]$$

$$= \frac{1}{N^2}\sum_{c=1}^{N}\Big[\sum_{h=1}^{H}\mathbb{E}\big[\|\Gamma_{s,h+1:H}^{(c,\eta)}\varepsilon^c(Z_{s,h}^c)\|^2\big]\Big]$$

$$\leq \frac{1}{N^2}\sum_{c=1}^{N}\sum_{h=1}^{H}(1-\eta a)^{2(H-h)}\mathbb{E}\big[\|\varepsilon^c(Z_{s,h}^c)\|^2\big] \ .$$

Therefore, using (3) and the following inequality,

$$\sum_{h=0}^{H-1}(1-\eta a)^{2h} \leq H \wedge \frac{1}{\eta a}, \quad \text{for all } \eta \geq 0, \text{ such that } \eta a \leq 1,$$

we get

$$\mathbb{E}\big[\|\bar{\varphi}_{s,H}\|^2\big] \leq \tfrac{1}{N}\left(H \wedge \tfrac{1}{\eta a}\right)\bar{\sigma}_{\varepsilon} \ .$$

Plugging this inequality in (15), we get

$$\mathbb{E}\big[\|\tilde{\theta}_t^{(\mathsf{fl})}\|^2\big] \leq \frac{\mathbb{E}_c[\mathrm{Tr}(\Sigma_{\varepsilon}^c)]}{N}\left(\eta^2 H \wedge \frac{\eta}{a}\right)\sum_{s=1}^{t}\big[(1-\eta a)^{2H(t-s)}\big]$$

$$\leq \frac{\bar{\sigma}_{\varepsilon}}{N}\left(\eta^2 H \wedge \frac{\eta}{a}\right)\frac{1}{1-(1-\eta a)^{2H}}$$

$$\leq \frac{\eta\bar{\sigma}_{\varepsilon}}{aN}\left(\eta a H \wedge 1\right)\frac{1}{1-\mathrm{e}^{-2\eta a H}} \ ,$$

where we used additionally

$$\mathrm{e}^{-2x} \leq 1 - x \leq \mathrm{e}^{-x} \ , \tag{16}$$

which is valid for $x \in [0; 1/2]$. Now it remains to notice that

$$\frac{x \wedge 1}{1-\mathrm{e}^{-2x}} \leq \frac{1}{1-\mathrm{e}^{-2}}$$

for any $x > 0$. $\qquad\square$

We proceed with analyzing the fluctuation of the true bias component of the error $\theta_t$ defined in (13). The first step towards this is to obtain the respective bound for $\bar{\tau}_{s,H}$, $s \in \{1, \ldots, T\}$, where $\bar{\tau}_{s,H}$ is defined in (11). Now we provide an upper bound for $\tilde{\theta}_t^{(\mathsf{fl},\mathsf{bi})}$:

**Lemma A.2.** *Assume A1 and A3. Then, for any step size $\eta \in (0, \eta_\infty)$ it holds*

$$\mathbb{E}^{1/2}\big[\|\tilde{\theta}_t^{(\mathrm{fl,bi})}\|^2\big] \leq \sqrt{\frac{2\eta\tilde{v}_{heter}}{Na}} + \frac{2\sqrt{\mathbb{E}_c\|\Sigma_{\widetilde{\mathbf{A}}}^c\|}\|\bar{\rho}_H\|}{aH^{1/2}N^{1/2}} \ .$$

*Proof.* Recall that $\tilde{\theta}_t^{(\mathrm{fl,bi})}$ is given (see (13)) by

$$\tilde{\theta}_t^{(\mathrm{fl,bi})} = \underbrace{\sum_{s=1}^t \prod_{i=s+1}^t \bar{\Gamma}_{i,H}^{(\eta)}\bar{\tau}_{s,H}}_{T_1} + \underbrace{\left(\sum_{s=1}^t \{\prod_{i=s+1}^t \bar{\Gamma}_{i,H}^{(\eta)}\} - (\bar{\Gamma}_H^{(\eta)})^{t-s}\right)\bar{\rho}_H}_{T_2} \ , \tag{17}$$

where $\bar{\tau}_{s,H}$ and $\bar{\rho}_H$ are defined in (11). We begin with bounding $T_1$. In order to do it we first need to bound $\bar{\tau}_{s,H}$. Since the different agents are independent, we have

$$\mathbb{E}[\|\bar{\tau}_{s,H}\|^2] = \frac{1}{N^2}\sum_{c=1}^N \mathbb{E}[\|((I - \eta\bar{\mathbf{A}}^c)^H - \Gamma_{s,1:H}^{(c,\eta)})\{\theta_\star^c - \theta_\star\}\|^2] \ . \tag{18}$$

Applying Lemma D.1 and the fact that $\big\{(I - \eta\bar{\mathbf{A}}^c)^{h-1}\widetilde{\mathbf{A}}^c(Z_{s,h}^c)\Gamma_{s,(h+1):H}^{(c,\eta)}(\theta_\star^c - \theta_\star)\big\}_{h=1}^H$ is a martingale-difference w.r.t. $\mathcal{F}_{s,h}^-$, we get that

$$\mathbb{E}[\|((I - \eta\bar{\mathbf{A}}^c)^H - \Gamma_{s,1:H}^{(c,\eta)})\{\theta_\star^c - \theta_\star\}\|^2]$$

$$= \eta^2\mathbb{E}[\|\sum_{h=1}^H (I - \eta\bar{\mathbf{A}}^c)^{h-1}\widetilde{\mathbf{A}}^c(Z_{s,h}^c)\Gamma_{s,(h+1):H}^{(c,\eta)}\{\theta_\star^c - \theta_\star\}\|^2]$$

$$= \eta^2\sum_{h=1}^H \mathbb{E}[\|(I - \eta\bar{\mathbf{A}}^c)^{h-1}\widetilde{\mathbf{A}}^c(Z_{s,h}^c)\Gamma_{s,(h+1):H}^{(c,\eta)}\{\theta_\star^c - \theta_\star\}\|^2]$$

$$\leq \eta^2\sum_{h=1}^H (1-\eta a)^{2(h-1)}\{\theta_\star^c - \theta_\star\}^\top \mathbb{E}[(\Gamma_{s,(h+1):H}^{(c,\eta)})^\top(\widetilde{\mathbf{A}}^c(Z_{s,h}^c))^\top \widetilde{\mathbf{A}}^c(Z_{s,h}^c)\Gamma_{s,(h+1):H}^{(c,\eta)}]\{\theta_\star^c - \theta_\star\} \ .$$

Using the tower property conditionally on $\mathcal{F}_{s,h+1}^+$, we get

$$\mathbb{E}[(\Gamma_{s,(h+1):H}^{(c,\eta)})^\top(\widetilde{\mathbf{A}}^c(Z_{s,h}^c))^\top \widetilde{\mathbf{A}}^c(Z_{s,h}^c)\Gamma_{s,(h+1):H}^{(c,\eta)}] = \mathbb{E}[(\Gamma_{s,(h+1):H}^{(c,\eta)})^\top\Sigma_{\widetilde{\mathbf{A}}}^c\Gamma_{s,(h+1):H}^{(c,\eta)}] \ ,$$

where $\Sigma_{\widetilde{\mathbf{A}}}^c$ is the noise covariance matrix defined in (3). Since for any vector $u \in \mathbb{R}^d$ we have $\|u\|_\Sigma \leq \|\Sigma_{\widetilde{\mathbf{A}}}^c\|^{1/2}\|u\|$, we get

$$\mathbb{E}[\|((I - \eta\bar{\mathbf{A}}^c)^H - \Gamma_{s,1:H}^{(c,\eta)})\{\theta_\star^c - \theta_\star\}\|^2] \leq \eta^2\sum_{h=1}^H (1-\eta a)^{2(h-1)}\mathbb{E}\big[\|\Gamma_{s,(h+1):H}^{(c,\eta)}\{\theta_\star^c - \theta_\star\}\|_\Sigma^2\big]$$

$$\leq \eta^2\|\Sigma_{\widetilde{\mathbf{A}}}^c\|\sum_{h=1}^H (1-\eta a)^{2(h-1)}\mathbb{E}\big[\|\Gamma_{s,(h+1):H}^{(c,\eta)}\{\theta_\star^c - \theta_\star\}\|^2\big] \tag{19}$$

$$\leq H\eta^2(1-\eta a)^{2(H-1)}\|\Sigma_{\widetilde{\mathbf{A}}}^c\|\|\theta_\star^c - \theta_\star\|^2 \ .$$

Combining the above bounds in (18) yields that

$$\mathbb{E}\big[\|\bar{\tau}_{s,H}\|^2\big] \leq \frac{H\eta^2(1-\eta a)^{2(H-1)}\sum_{c=1}^N \|\Sigma_{\widetilde{\mathbf{A}}}^c\|\|\theta_\star^c - \theta_\star\|^2}{N^2} \ . \tag{20}$$

Thus, proceeding as in (14) together with (20), we get

$$
\begin{aligned}
\mathbb{E}[\|T_1\|^2] &= \sum_{s=1}^{t} \mathbb{E}[\|\prod_{i=s+1}^{t} \bar{\Gamma}_{i,H}^{(\eta)} \bar{\tau}_{s,H}\|^2] \\
&\leq \sum_{s=1}^{t} \frac{H\eta^2(1-\eta a)^{2(H-1)} \sum_{c=1}^{N} \|\Sigma_{\widetilde{\mathbf{A}}}^{c}\| \|\theta_{\star}^{c} - \theta_{\star}\|^2}{N^2} (1-\eta a)^{2H(t-s)} \\
&\leq \frac{H\eta^2(1-\eta a)^{2(H-1)}}{(1-(1-\eta a)^{2H})N} \mathbb{E}_c[\|\Sigma_{\widetilde{\mathbf{A}}}^{c}\| \|\theta_{\star}^{c} - \theta_{\star}\|^2] \\
&\leq \frac{\eta}{aN(1-\eta a)^2} \frac{Ha\eta e^{-2Ha\eta}}{1-e^{-2Ha\eta}} \mathbb{E}_c[\|\Sigma_{\widetilde{\mathbf{A}}}^{c}\| \|\theta_{\star}^{c} - \theta_{\star}\|^2] \\
&\leq \frac{2\eta}{Na} \mathbb{E}_c[\|\Sigma_{\widetilde{\mathbf{A}}}^{c}\| \|\theta_{\star}^{c} - \theta_{\star}\|^2] \,.
\end{aligned}
$$

In the bound above we used (16) together with the bound

$$
\frac{xe^{-2x}}{1-e^{-2x}} \leq \frac{1}{2} \,, x \geq 0 \,.
$$

Now we bound the second part of $\tilde{\theta}_t^{(\mathsf{fl,bi})}$ in (17), that is, $T_2$. To begin with, we start with applying Lemma D.1 and we get for any $s \in \{1, \ldots, t\}$ and $i \in \{s+1, \ldots, t\}$, that

$$
\{\prod_{i=s+1}^{t} \bar{\Gamma}_{i,H}^{(\eta)}\} - (\bar{\Gamma}_{H}^{(\eta)})^{t-s})\bar{\rho}_H = \sum_{i=s+1}^{t} \{\prod_{r=i+1}^{t} \bar{\Gamma}_{r,H}^{(\eta)}\}(\bar{\Gamma}_{i,H}^{(\eta)} - \bar{\Gamma}_{H}^{(\eta)})(\bar{\Gamma}_{H}^{(\eta)})^{i-s-1}\bar{\rho}_H \,.
$$

Note that,

$$
\mathbb{E}^{\mathcal{F}}[\{\prod_{r=i+1}^{t} \bar{\Gamma}_{r,H}^{(\eta)}\}(\bar{\Gamma}_{i,H}^{(\eta)} - \bar{\Gamma}_{H}^{(\eta)})(\bar{\Gamma}_{H}^{(\eta)})^{i-s-1}\bar{\rho}_H] = 0 \tag{21}
$$

Proceeding as in (19), we get using independence between agents for any $u \in \mathbb{R}^d$,

$$
\begin{aligned}
\mathbb{E}[\|(\bar{\Gamma}_{i,H}^{(\eta)} - \bar{\Gamma}_{H}^{(\eta)})u\|^2] &= \frac{1}{N^2} \mathbb{E}[\|\sum_{c=1}^{N}(\Gamma_{s,1:H}^{(c,\eta)} - (I-\eta\bar{\mathbf{A}}^c)^H)u\|^2] \\
&= \frac{1}{N^2} \sum_{c=1}^{N} \mathbb{E}[\|(\Gamma_{s,1:H}^{(c,\eta)} - (I-\eta\bar{\mathbf{A}}^c)^H)u\|^2] \\
&\leq \frac{H\eta^2(1-\eta a)^{2(H-1)}}{N} \left(\frac{1}{N}\sum_{c=1}^{N}\|\Sigma_{\widetilde{\mathbf{A}}}^{c}\|\right) \|u\|^2 \,.
\end{aligned}
$$

Hence, using (21), we get

$$
\mathbb{E}[\|(\{\prod_{i=s+1}^{t} \bar{\Gamma}_{i,H}^{(\eta)}\} - (\bar{\Gamma}_{H}^{(\eta)})^{t-s})\bar{\rho}_H\|^2] = \frac{H\eta^2(1-\eta a)^{2H(t-s)-2}\mathbb{E}_c\|\Sigma_{\widetilde{\mathbf{A}}}^{c}\|}{N} \|\bar{\rho}_H\|^2 \,.
$$

Combining the above estimates in (17), and using Minkowski's inequality, we get

$$
\begin{aligned}
\mathbb{E}^{1/2}[\|T_2\|^2] &\leq \frac{H^{1/2}\eta}{(1-\eta a)N^{1/2}} \sqrt{\mathbb{E}_c\|\Sigma_{\widetilde{\mathbf{A}}}^{c}\|} \|\bar{\rho}_H\| \sum_{s=1}^{t-1}(1-\eta a)^{H(t-s)} \\
&\leq \frac{2}{aH^{1/2}N^{1/2}} \frac{Ha\eta e^{-Ha\eta}}{1-e^{-Ha\eta}} \sqrt{\mathbb{E}_c\|\Sigma_{\widetilde{\mathbf{A}}}^{c}\|} \|\bar{\rho}_H\| \\
&\leq \frac{2}{aH^{1/2}N^{1/2}} \sqrt{\mathbb{E}_c\|\Sigma_{\widetilde{\mathbf{A}}}^{c}\|} \|\bar{\rho}_H\| \,,
\end{aligned}
$$

where we used that $\eta a \leq 1/2$ and

$$
\frac{xe^{-x}}{1-e^{-x}} \leq 1 \,, \quad x \geq 0 \,.
$$

and the statement follows. $\qquad\square$

**Lemma A.3.** *Recall that* $\bar{\rho}_H = \frac{1}{N}\sum_{c=1}^{N}(I - (I - \eta\bar{\mathbf{A}}^c)^H)\{\theta_\star^c - \theta_\star\}$, *it satisfies*

$$\|\bar{\rho}_H\| \leq \frac{\eta^2 H^2}{N}\sum_{c=1}^{N}\exp(\eta H\|\bar{\mathbf{A}}^c\|)\|\theta_\star^c - \theta_\star\| \tag{22}$$

*Proof.* Using the identity,

$$1 - (1 - u)^H = Hu - u^2\sum_{k=0}^{H-2}(-1)^k\binom{H}{k+2}u^k \tag{23}$$

and the inequality $\binom{H}{k+2} \leq \binom{H-2}{k}H^2$, we get that

$$\left|\sum_{k=0}^{H-2}(-1)^k\binom{H}{k+2}u^k\right| \leq \frac{H^2}{2}\sum_{k=0}^{H-2}\binom{H-2}{k}|u|^k \leq \frac{H^2}{2}\exp((H-2)|u|) \tag{24}$$

Using (23) with $u = \eta\bar{\mathbf{A}}^c$ for all $c$, we get

$$\bar{\rho}_H = \frac{1}{N}\sum_{c=1}^{N}H\eta\bar{\mathbf{A}}^c - \eta^2(\bar{\mathbf{A}}^c)^2\sum_{k=0}^{H-2}(-1)^k\binom{H}{k+2}(\eta\bar{\mathbf{A}}^c)^k\ ,$$

by definition of $\theta_\star^c$ and $\theta_\star$, we have that $\sum_{c=1}^{N}\bar{\mathbf{A}}^c(\theta_\star^c - \theta_\star) = \sum_{c=1}^{N}\bar{\mathbf{A}}^c\theta_\star^c - (\sum_{c=1}^{N}\bar{\mathbf{A}}^c)\theta_\star = \sum_{c=1}^{N}\bar{\mathbf{b}}^c - \sum_{c=1}^{N}\bar{\mathbf{b}}^c = 0$. Using this and (24), we finally get (22). $\qquad\square$

**Lemma A.4.** *Assume A1 and A3. Then for any step size $\eta \in (0, \eta_\infty)$ we have*

$$\mathbb{E}^{1/2}[\|\tilde{\theta}_t^{(\mathsf{tr})}\|^2] \leq (1 - \eta a)^{tH}\|\theta_0 - \theta_\star\|$$

*Proof.* Proceeding as in (14) for any $u \in \mathbb{R}^d$ we have

$$\mathbb{E}^{1/2}[\|\prod_{s=1}^{t}\bar{\Gamma}_{s,H}^{(\eta)}u\|^2] \leq (1 - \eta a)^{tH}\|u\|$$

Using this result for $u = \theta_0 - \theta_\star$ we get the statement. $\qquad\square$

**Lemma A.5.** *Assume A1 and A3. Then for any $\eta \in (0, \eta_\infty)$ we have*

$$\|\tilde{\theta}_t^{(\mathsf{bi,bi})} - (I - \bar{\Gamma}_H^{(\eta)})^{-1}\bar{\rho}_H\| \leq (1 - \eta a)^{tH}\|(I - \bar{\Gamma}_H^{(\eta)})^{-1}\|\|\bar{\rho}_H\|$$

*Proof.* Using A3 and Minkowski's inequalitty, we get

$$\begin{aligned}
\|\tilde{\theta}_t^{(\mathsf{bi,bi})} - (I - \bar{\Gamma}_H^{(\eta)})^{-1}\bar{\rho}_H\| &= \|(I - \bar{\Gamma}_H^{(\eta)})^{-1}(\bar{\Gamma}_H^{(\eta)})^t\bar{\rho}_H\| \\
&\leq \|(I - \bar{\Gamma}_H^{(\eta)})^{-1}\|\|(\bar{\Gamma}_H^{(\eta)})^t\bar{\rho}_H\| \\
&\leq \|(I - \bar{\Gamma}_H^{(\eta)})^{-1}\|\frac{1}{N}\sum_{c=1}^{N}\|(I - \eta\bar{\mathbf{A}}^c)^H(\bar{\Gamma}_H^{(\eta)})^{t-1}\bar{\rho}_H\| \\
&\leq (1 - \eta a)^H\|(I - \bar{\Gamma}_H^{(\eta)})^{-1}\|\|(\bar{\Gamma}_H^{(\eta)})^{t-1}\bar{\rho}_H\|
\end{aligned}$$

and the statement follows. $\qquad\square$

**Theorem A.6.** *Assume A1 and A3. Then for any step size $\eta \in (0, \eta_\infty)$ it holds that*

$$\begin{aligned}
\mathbb{E}^{1/2}\big[\|\theta_t - \tilde{\theta}_t^{(\mathsf{bi,bi})} - \theta_\star\|^2\big] &\leq \sqrt{\frac{\eta\bar{\sigma}_\varepsilon}{aN(1 - \mathrm{e}^{-2})}} + \sqrt{\frac{2\eta\tilde{v}_{heter}}{Na}} \\
&\quad + \frac{2\sqrt{\mathbb{E}_c\|\Sigma_{\tilde{\mathbf{A}}}^c\|}\|\bar{\rho}_H\|}{aH^{1/2}N^{1/2}} + (1 - \eta a)^{tH}\|\theta_0 - \theta_\star\|\ ,
\end{aligned} \tag{25}$$

*where the bias $\tilde{\theta}_t^{(\mathsf{bi,bi})}$ converges to $(I - \bar{\Gamma}_H^{(\eta)})^{-1}\bar{\rho}_H$ at a rate*

$$\|\tilde{\theta}_t^{(\mathsf{bi,bi})} - (I - \bar{\Gamma}_H^{(\eta)})^{-1}\bar{\rho}_H\| \leq (1 - \eta a)^{tH}\|(I - \bar{\Gamma}_H^{(\eta)})^{-1}\|\|\bar{\rho}_H\|\ .$$

*Proof.* Proof follows by combining the results Lemma A.1-Lemma A.5 above. □

In the lemma below we provide a simplified sample complexity bound of Corollary 4.3 corresponding to the synchronous setting, that is, with number of local training steps $H = 1$. There, the bias term disappears, and above results directly give a simplified sample complexity bound.

**Corollary A.7.** *Assume A1 and A3. Let $H = 1$, then for any $0 < \epsilon < 1$, in order to achieve $\mathbb{E}\big[\|\theta_T - \theta_\star\|^2\big] \leq \epsilon^2$ the required number of communications is*

$$T = \mathcal{O}\left(\frac{\tilde{v}_{heter} \vee \bar{\sigma}_\varepsilon}{Na^2\epsilon^2} \log \frac{\|\theta_0 - \theta_\star\|}{\epsilon}\right)$$

*number of communications, setting the step size*

$$\eta_0 = \frac{aN\epsilon^2}{\tilde{v}_{heter} \vee \bar{\sigma}_\varepsilon} \ . \tag{26}$$

*Proof.* Bounding the first two terms in decomposition (25) we get that the step size should satisfy

$$\eta \leq \frac{aN\epsilon^2}{\tilde{v}_{heter} \vee \bar{\sigma}_\varepsilon} \ .$$

From the last term we have

$$t \geq \frac{1}{a\eta} \log \frac{\|\theta_0 - \theta_\star\|}{\epsilon} \geq \frac{\tilde{v}_{heter} \vee \bar{\sigma}_\varepsilon}{Na^2\epsilon^2} \log \frac{\|\theta_0 - \theta_\star\|}{\epsilon}$$

□

**Corollary A.8.** *Assume A1 and A3. For any*

$$0 \leq \epsilon \leq \frac{\mathbf{C_A}^{-1}\mathbb{E}_c\|\theta_\star^c - \theta_\star\|}{a} \vee \left(\frac{\sqrt{\tilde{v}_{heter} \vee \bar{\sigma}_\varepsilon}\mathbb{E}_c\|\theta_\star^c - \theta_\star\|}{a}\right)^{2/5}$$

*in order to achieve $\mathbb{E}\big[\|\theta_T - \theta_\star\|^2\big] < \epsilon^2$ the required number of communications is*

$$T = \mathcal{O}\left(\frac{\mathbb{E}_c\|\theta_\star^c - \theta_\star\|}{a^2\epsilon} \log \frac{\|\theta_0 - \theta_\star\|}{\epsilon}\right) \ , \tag{27}$$

*setting the step size*

$$\eta = \mathcal{O}\left(\frac{aN\epsilon^2}{\tilde{v}_{heter} \vee \bar{\sigma}_\varepsilon}\right)$$

*and number of local iterations*

$$H = \mathcal{O}\left(\frac{\tilde{v}_{heter} \vee \bar{\sigma}_\varepsilon}{N\epsilon\mathbb{E}_c\|\theta_\star^c - \theta_\star\|}\right)$$

*Proof.* We aim to bound separately all the terms in the r.h.s. of Theorem 4.1. Note that it requires to set $\eta \in (0; \eta_0)$ with $\eta_0$ given in (26) in order to fulfill the bounds

$$\sqrt{\frac{\eta\tilde{v}_{heter}}{aN}} \lesssim \varepsilon \ , \quad \sqrt{\frac{\eta\bar{\sigma}_\varepsilon}{aN}} \lesssim \varepsilon \ .$$

Now, we should bound the bias term

$$\mathbb{E}^{1/2}[\|\tilde{\theta}_t^{(\mathsf{bi,bi})}\|^2] \leq (1 + (1 - \eta a)^{tH})\|(I - \bar{\Gamma}_H^{(\eta)})^{-1}\bar{\rho}_H\| \leq 2\|(I - \bar{\Gamma}_H^{(\eta)})^{-1}\bar{\rho}_H\| \ .$$

Thus, using the Neuman series, we can bound the norm of the term above as

$$\|(I - \bar{\Gamma}_H^{(\eta)})^{-1}\bar{\rho}_H\| = \|\sum_{k=0}^{\infty}(\bar{\Gamma}_H^{(\eta)})^k\bar{\rho}_H\| \leq \sum_{k=0}^{\infty}(1 - \eta a)^{Hk}\|\bar{\rho}_H\| \leq \frac{\|\bar{\rho}_H\|}{1 - (1 - \eta a)^H} \ .$$

Hence, using the bound of Lemma A.3, we get

$$\mathbb{E}^{1/2}[\|\tilde{\theta}_t^{(\mathsf{bi},\mathsf{bi})}\|^2] \leq \frac{2\|\bar{\rho}_H\|}{1-(1-\eta a)^H} \leq \frac{\eta a H}{1-(1-\eta a)^H} \frac{\eta H \mathbb{E}_c[\exp(\eta H\|\bar{\mathbf{A}}^c\|)]\|\theta_\star^c - \theta_\star\|]}{a}$$

$$\leq \frac{2\eta H \mathbb{E}_c[\exp(\eta H\|\bar{\mathbf{A}}^c\|)]\|\theta_\star^c - \theta_\star\|]}{a} \lesssim \frac{\eta H \mathbb{E}_c[\|\theta_\star^c - \theta_\star\|]}{a} ,$$

where we used the fact that the step size $\eta$ is chosen in order to satisfy $\eta H\, \mathrm{C}_{\mathbf{A}} \leq 1$. Thus in order to fulfill $\mathbb{E}^{1/2}[\|\tilde{\theta}_t^{(\mathsf{bi},\mathsf{bi})}\|^2] \lesssim \varepsilon$ we need to choose $\eta$ and $H$ such that

$$\eta H \mathbb{E}_c[\|\theta_\star^c - \theta_\star\|] \leq \varepsilon a .$$

It remains to bound the term $\frac{\sqrt{\mathbb{E}\|\Sigma\|\|\bar{\rho}\|}}{aHN}$. Using the bound of Lemma A.3, we get

$$\frac{\sqrt{\mathbb{E}_c\|\Sigma_{\bar{\mathbf{A}}}^c\|}\|\bar{\rho}_H\|}{aH^{1/2}N^{1/2}} \leq \sqrt{\frac{\eta}{N}} \times \frac{\sqrt{\mathbb{E}_c\|\Sigma_{\bar{\mathbf{A}}}^c\|}(\eta H)^{3/2}}{a} \lesssim \varepsilon^{5/2}\sqrt{\frac{1}{\tilde{v}_{\mathsf{heter}} \vee \bar{\sigma}_\varepsilon}} \frac{a}{\mathbb{E}_c[\|\theta_\star^c - \theta_\star\|]} .$$

Hence, it remains to combine the bounds above in order to get the sample complexity result (27). □

**Corollary A.9.** *Assume **TD** 1 and **TD** 3. Then for any*

$$0 \leq \epsilon \leq \frac{2\left(\sqrt{2}(1+\gamma)\sqrt{\mathbb{E}\|\theta-\theta\|\vee(1+\mathbb{E}[\|\theta\|])}\mathbb{E}[\|\theta-\theta\|]\right)}{(1-\gamma)\nu} \vee \frac{2\mathbb{E}[\|\theta-\theta\|]}{(1-\gamma)\nu(1+\gamma)} ,$$

*in order to achieve $\mathbb{E}\big[\|\theta_T - \theta_\star\|^2\big] < \epsilon^2$ the required number of communications for federated TD(0) algorithm is*

$$T = \mathcal{O}\left(\left(\frac{1}{(1-\gamma)\nu} \vee \frac{\mathbb{E}[\|\theta-\theta\|]}{(1-\gamma)\nu\epsilon}\right) \log \frac{\|\theta-\theta\|}{\epsilon}\right) .$$

# B  Markovian sampling schemes for **FedLSA**

Note that under A2 each of the matrices $\bar{\mathbf{A}}^c$, $c \in [N]$ is Hurwitz. This guarantees the existence and uniqueness of a positive definite matrix $Q_c$ which is a solution of the *Lyapunov equation*

$$\{\bar{\mathbf{A}}^c\}^\top Q_c + Q_c \bar{\mathbf{A}}^c = \mathrm{I} .$$

We further introduce the associated quantities, that will be used throughout the proof.

$$a_c = \|Q_c\|^{-1}/2 , \quad \tilde{\eta}_{\infty,c} = (1/2)\|\bar{\mathbf{A}}^c\|_Q^{-2}\|Q_c\|^{-1} \wedge \|Q_c\| , \quad \tilde{a} = \min_{c\in[N]} a_c , \quad \tilde{\eta}_\infty = \min_{c\in[N]} \tilde{\eta}_{\infty,c} ,$$

$$\kappa_{Q,c} = \lambda_{\mathsf{max}}(Q_c)/\lambda_{\mathsf{min}}(Q_c) , \quad b_{Q,c} = 2\sqrt{\kappa_{Q,c}}\,\mathrm{C}_{\mathbf{A}} , \quad \kappa_Q = \max_{c\in[N]} \kappa_{Q,c} , \quad b_Q = \max_{c\in[N]} b_{Q,c} .$$

In our statement of A2 we also required that each of the chains $(Z_k^c)_{k\in\mathbb{N}}$ starts from its invariant distribution $\pi_c$. This requirement can be removed, and extension to the setting of arbitrary initial distribution can be done based on the maximal exact coupling argument [13, Lemma 19.3.6 and Theorem 19 3.9]. However, to better highlight the main ingredients of the proof, we prefer to keep stationary assumption.

*Proof of Corollary 4.5.* Assume that the total number of local iterations, that is, $TH$, satisfies

$$TH = 2qm + k , \quad 0 \leq k < 2q , \tag{29}$$

where $q \in \mathbb{N}$ is a parameter that will be determined later. With Lemma B.4 we construct for each $c \in [N]$ a sequence of random variables $\{\tilde{Z}_{2jq}^{\star,c}\}_{j=1,\dots,m}$, which are i.i.d. with the same distribution $\pi_c$. Moreover, Lemma B.4 together with union bound imply

$$\mathbb{P}(\exists j \in [m], c \in [N] : \tilde{Z}_{2jq}^{\star,c} \neq Z_{2jq}^c) \leq mN(1/4)^{\lfloor q/\tau \rfloor} .$$

The bound (29) implies that $m \leq TH/(2q)$. Thus, for any $\delta \in (0,1)$, in order to guarantee that

$$\mathbb{P}(\exists j \in [m], c \in [N] : \tilde{Z}_{2jq}^{\star,c} \neq Z_{2jq}^c) \leq \delta$$

**Algorithm 3** FedLSA with Markovian data

---

**Input:** $\eta > 0$, $\theta_0 \in \mathbb{R}^d$, $T, N, H > 0$, time window $q \in \mathbb{N}$.
**for** $t = 0$ to $T - 1$ **do**
    Initialize $\theta_{t,0} = \theta_t$
    **for** $c = 1$ to $N$ **do**
        **for** $h = 1$ to $H$ **do**
            Receive $Z_{t,h}^c$, then check the condition:
            **if** $h = qj, j \in \mathbb{N}$ **then**
                Compute local update

$$\theta_{t,j}^c = \theta_{t,j-1}^c - \eta(\mathbf{A}^c(Z_{t,qj}^c) - \mathbf{b}^c(Z_{t,qj}^c))$$

            **else**
                Skip current update

Average: $\theta_{t+1} = \frac{1}{N} \sum_{c=1}^{N} \theta_{t,H}^c$                                   (28)

---

it is enough to ensure that

$$mN(1/4)^{\lfloor q/\tau \rfloor} \le \frac{2NHT(1/4)^{q/\tau}}{q} \le \delta .$$ (30)

Inequality (30) holds for fixed $\delta \in (0, 1)$, if we choose

$$q = \left\lceil \frac{\tau_{\mathrm{mix}} \log(2NHT/\delta)}{\log 4} \right\rceil .$$ (31)

Thus, setting the block size $q$ as in (31), we get that for total number iterations $TH$ satisfying (29), with probability at least $1 - \delta$ the results of Algorithm 3 are indistinguishable from the result of its counterpart Algorithm 1 applied with number of local steps $H/q$. We will denote the iterates of the latter algorithm applied with number of local steps $h \in \mathbb{N}$ as $\theta_T^{(\mathrm{ind}),h}$ in order to make explicit the dependence of global parameter upon the number of local iterates. We further denote the event, where $\theta_T^{(\mathrm{ind}),H/q} = \theta_T$, by $\mathsf{A}_\delta$. Thus, setting

$$\frac{H}{q} = \mathcal{O}\left( \frac{\tilde{v}_{\mathrm{heter}} \vee \bar{\sigma}_\varepsilon}{\mathbb{E}_c[\|\theta_\star^c - \theta_\star\|]} \frac{1}{N\epsilon} \right) ,$$ (32)

similarly to the way the number of local updates is set in Corollary 4.3, we obtain that

$$\mathbb{E}[\|\theta_T - \theta_\star\|^2] = \mathbb{E}[\|\theta_T - \theta_\star\|^2 \mathbf{1}_{\mathsf{A}}] + \mathbb{E}[\|\theta_T - \theta_\star\|^2 \mathbf{1}_{\overline{\mathsf{A}}}]$$ (33)
$$= \mathbb{E}[\|\theta_T^{(\mathrm{ind}),H/q} - \theta_\star\|^2 \mathbf{1}_{\mathsf{A}}] + \mathbb{E}[\|\theta_T - \theta_\star\|^2 \mathbf{1}_{\overline{\mathsf{A}}}]$$
$$\le \epsilon^2 + \sqrt{\delta} \mathbb{E}^{1/2}[\|\theta_T - \theta_\star\|^4] ,$$

where in the last inequality we relied on the special choice of $H/q$ from (32) together with Holder's inequality. Now it remains to bound $\mathbb{E}[\|\theta_T - \theta_\star\|^4]$ and tune the parameter $\delta$ appropriately. Note that within this bound we can not rely on the estimates based on independent observations $\{\tilde{Z}_{2jq}^{\star,c}\}_{j=1,\ldots,m}$. At the same time, note that the skeleton $Z_{2jq}^c$, $j \ge 0$ for any $c \in [N]$ is a Markov chain with the Markov kernel $\mathsf{P}_c^q$ and mixing time $\tau_{\mathrm{mix}} = 1$. This allows us to write a simple upper bound on $\mathbb{E}[\|\theta_T - \theta_\star\|^4]$ based on the stability result for product of random matrices provided in [14]. Indeed, applying the result of Lemma B.1, we get

$$\mathbb{E}^{1/2}[\|\theta_T - \theta_\star\|^4] \le \left( \|\theta_0 - \theta_\star\| + \frac{2T}{N} \sum_{c=1}^{N} \|\theta_\star^c - \theta_\star\| + \eta TH\|\varepsilon\|_\infty \right)^2 ,$$

and the corresponding bound (33) can be rewritten as

$$\mathbb{E}[\|\theta_T - \theta_\star\|^2] \le \epsilon^2 + \sqrt{\delta} \left( \|\theta_0 - \theta_\star\| + \frac{2T}{N} \sum_{c=1}^{N} \|\theta_0 - \theta_\star\| + \eta TH\|\varepsilon\|_\infty \right)^2 .$$

Thus, setting

$$\delta = \frac{\epsilon^4}{H^4 T^4 \left( \|\theta_0 - \theta_\star\| + \frac{2}{N} \sum_{c=1}^{N} \|\theta_\star^c - \theta_\star\| + \eta\|\varepsilon\|_\infty \right)^2} \;,$$

we obtain that the corresponding bound for block size $q$ scales as

$$q = \left\lceil \frac{\tau_{\mathrm{mix}} \log\left( 2NHT/\delta \right)}{\log 4} \right\rceil \lesssim \left\lceil \tau_{\mathrm{mix}} \log H \log\left( NT^5 \Delta_{corr}/\epsilon^2 \right) \right\rceil \;,$$

where we write $\lesssim$ for inequality up to an absolute constant and set

$$\Delta_{corr} = \left( \|\theta_0 - \theta_\star\| + \frac{2}{N} \sum_{c=1}^{N} \|\theta_\star^c - \theta_\star\| + \eta\|\varepsilon\|_\infty \right)^2 \;.$$

Combination of the above bounds yields that

$$\mathbb{E}[\|\theta_T - \theta_\star\|^2] \leq 2\epsilon^2 \;,$$

and the proof is completed. $\qquad\qquad\square$

**Lemma B.1.** *Assume A2 and A3. Then, for the iterates $\theta_t$ of Algorithm 3 run with parameters $\eta, H, q$ satisfying the relation*

$$\frac{\eta H}{q} \geq \frac{12}{\tilde{a}} \left( 2 + \frac{\log d}{2} + \frac{\log \kappa_Q}{2} \right) \;, \tag{34}$$

*it holds for any probability distribution $\xi$ on $(\mathsf{Z}, \mathcal{Z})$ and any $t \in \mathbb{N}$, that*

$$\mathbb{E}_\xi^{1/4}[\|\theta_t - \theta_\star\|^4] \leq \|\theta_0 - \theta_\star\| + \frac{2t}{N} \sum_{c=1}^{N} \|\theta_\star^c - \theta_\star\| + \eta t H \|\varepsilon\|_\infty \;.$$

*Proof.* First we write a counterpart of the error decomposition (12) - (13) for the LSA error of the subsampled iterates of Algorithm 3. Namely, we write that

$$\theta_t - \theta_\star = \bar{\Gamma}_{t,H}^{(\eta,q)} \{\theta_{t-1} - \theta_\star\} + \varkappa_{t,H} + \eta\bar{\varphi}_{t,H} \;, \tag{35}$$

where we have defined

$$\Gamma_{t,m:n}^{(c,\eta,q)} = \prod_{h=m}^{n} \left( \mathrm{I} - \eta\mathbf{A}(Z_{t,qh}^c) \right) \;, \quad 1 \leq m \leq n \leq H \;, \tag{36}$$

$$\bar{\Gamma}_{t,H}^{(\eta,q)} = \frac{1}{N} \sum_{c=1}^{N} \Gamma_{t,1:H}^{(c,\eta,q)} \;,$$

$$\varkappa_{t,H} = \frac{1}{N} \sum_{c=1}^{N} (\mathrm{I} - \Gamma_{t,1:H}^{(c,\eta,q)}) \{\theta_\star^c - \theta_\star\} \;,$$

$$\bar{\varphi}_{t,H} = -\frac{1}{N} \sum_{c=1}^{N} \sum_{h=1}^{H} \Gamma_{t,h+1:H}^{(c,\eta,q)} \varepsilon^c(Z_{t,qh}^c) \;.$$

For notation simplicity we have removed the dependence of $\varkappa_{t,H}$ on the subsampling parameter $q \in \mathbb{N}$. Thus, applying the result of [14, Proposition 7] (see also Lemma B.2) together with Minkowski's inequality, we obtain from the previous bound that for any distribution $\xi$ on $(\mathsf{Z}, \mathcal{Z})$,

$$\mathbb{E}_\xi^{1/4}[\|\Gamma_{t,1:H}^{(c,\eta,q)}\|^4] \leq \sqrt{\kappa_{Q,c}} \mathrm{e}^2 d^{1/2} \mathrm{e}^{-\eta\tilde{a}H/(12q)} \leq 1 \;,$$

provided that the ratio $\eta H/q$ satisfies the relation (34). This bound yields that

$$\mathbb{E}_\xi^{1/4}[\|\bar{\Gamma}_{t,H}^{(\eta,q)}\|^4] \leq 1 \;,$$

$$\mathbb{E}_\xi^{1/4}[\|\varkappa_{t,H}\|^4] \leq \frac{2}{N} \sum_{c=1}^{N} \|\theta_\star^c - \theta_\star\| \;,$$

$$\mathbb{E}_\xi^{1/4}[\|\bar{\varphi}_{t,H}\|^4] \leq H\|\varepsilon\|_\infty \;.$$

Hence, we obtain by running the recurrence (35), that

$$\mathbb{E}_\xi^{1/4}[\|\theta_t - \theta_\star\|^4] \leq \|\theta_0 - \theta_\star\| + \frac{2t}{N} \sum_{c=1}^{N} \|\theta_\star^c - \theta_\star\| + \eta t H \|\varepsilon\|_\infty \;,$$

and the statement follows. $\qquad\qquad\square$

**Stability results on product of random matrices.** The results of this paragraph provides the stability bound for the product of random matrices $\Gamma_{t,m:n}^{(c,\eta,q)}$ defined in (36). Define the quantities

$$\eta_\infty^{(\mathrm{M})} = \left[\tilde{\eta}_\infty \wedge \kappa_Q^{-1/2} \, \mathrm{C}_{\mathbf{A}}^{-1} \wedge \tilde{a}/(6\mathrm{e}\kappa_Q \, \mathrm{C}_{\mathbf{A}})\right] \times \lceil 8\kappa_Q^{1/2} \, \mathrm{C}_{\mathbf{A}} \, /\tilde{a}\rceil^{-1} \wedge \mathrm{c}_{\mathbf{A}}^{(\mathrm{M})} /2 \,, \qquad (37)$$

$$\mathrm{C}_{\boldsymbol{\Gamma}} = 4(\kappa_Q^{1/2} \, \mathrm{C}_{\mathbf{A}} + \tilde{a}/6)^2 \times \lceil 8\kappa_Q^{1/2} \, \mathrm{C}_{\mathbf{A}} \, /\tilde{a}\rceil \,, \quad \mathrm{c}_{\mathbf{A}}^{(\mathrm{M})} = \tilde{a}/\{12\,\mathrm{C}_{\boldsymbol{\Gamma}}\} \,.$$

Then the following result holds:

**Lemma B.2** (Proposition 7 from [14], simplified)**.** *Assume A2 and A3. Then, for any $c \in [N]$, $t \in \mathbb{N}$, step size $\eta \in \left(0, \eta_\infty^{(\mathrm{M})}\right]$, any $n \in \mathbb{N}$, $q \geq \tau_{\mathrm{mix}}$, and probability distribution $\xi$ on $(\mathsf{Z}, \mathcal{Z})$, it holds*

$$\mathbb{E}_\xi^{1/4}\left[\|\Gamma_{t,m:n}^{(c,\eta,q)}\|^4\right] \leq \sqrt{\kappa_{Q,c}}\mathrm{e}^2 d^{1/2}\mathrm{e}^{-\tilde{a}\eta(n-m)/12} \,.$$

*Proof.* It is enough to note that, since $q \geq \tau_{\mathrm{mix}}$, and we consider $q$-skeleton of each Markov kernels $\mathrm{P}_c$, each of the subsampled kernels $\mathrm{P}_c^q$ will have a mixing time 1. $\qquad\square$

**Berbee's lemma construction.** We outline some preliminaries associated with the Berbee's coupling lemma [3] construction. We recall first a definition of the $\beta$-mixing coefficient. Consider a probability space $(\Omega, \mathcal{F}, \mathbb{P})$ equipped with $\sigma$-fields $\mathfrak{F}$ and $\mathfrak{G}$ such that $\mathfrak{F} \subseteq \mathcal{F}, \mathfrak{G} \subseteq \mathcal{F}$. Then the $\beta$-mixing coefficient of $\mathfrak{F}$ and $\mathfrak{G}$ is defined as

$$\beta(\mathfrak{F}, \mathfrak{G}) = (1/2)\sup \sum_{i\in\mathsf{I}}\sum_{j\in\mathsf{J}} |\mathbb{P}(\mathsf{A}_i \cap \mathsf{B}_j) - \mathbb{P}(\mathsf{A}_i)\mathbb{P}(\mathsf{B}_j)| \,,$$

and the supremum is taken over all pairs of partitions $\{\mathsf{A}_i\}_{i\in\mathsf{I}} \in \mathfrak{F}^{\mathsf{I}}$ and $\{\mathsf{B}_j\}_{j\in\mathsf{J}} \in \mathfrak{G}^{\mathsf{J}}$ of $\tilde{\mathsf{Z}}_{\mathbb{N}}$ with finite $\mathsf{I}$ and $\mathsf{J}$.

Now let $(\mathsf{Z}, \mathsf{d}_{\mathsf{Z}})$ be a Polish space endowed with its Borel $\sigma$-field, denoted by $\mathcal{Z}$, and let $(\mathsf{Z}^{\mathbb{N}}, \mathcal{Z}^{\otimes\mathbb{N}})$ be the corresponding canonical space. Consider a Markov kernel $\mathrm{P}$ on $\mathsf{Z} \times \mathcal{Z}$ and denote by $\mathbb{P}_\xi$ and $\mathbb{E}_\xi$ the corresponding probability distribution and expectation with initial distribution $\xi$. Without loss of generality, we assume that $(Z_k)_{k\in\mathbb{N}}$ is the associated canonical process. By construction, for any $\mathsf{A} \in \mathcal{Z}$, $\mathbb{P}_\xi\left(Z_k \in \mathsf{A} \,|\, Z_{k-1}\right) = \mathrm{P}(Z_{k-1}, \mathsf{A})$, $\mathbb{P}_\xi$-a.s. In the case $\xi = \delta_z$, $z \in \mathsf{Z}$, $\mathbb{P}_\xi$ and $\mathbb{E}_\xi$ are denoted by $\mathbb{P}_z$ and $\mathbb{E}_z$, respectively. We now make an assumption about the mixing properties of $\mathrm{P}$:

*UGE* 1. The Markov kernel $\mathrm{P}$ admits $\pi$ as an invariant distribution and is uniformly geometrically ergodic, that is, there exists $\tau_{\mathrm{mix}} \in \mathbb{N}$ such that for all $k \in \mathbb{N}$,

$$\Delta(\mathrm{P}^k) = \sup_{z,z'\in\mathsf{Z}}(1/2)\|\mathrm{P}^k(z, \cdot) - \mathrm{P}^k(z', \cdot)\|_{\mathsf{TV}} \leq (1/4)^{\lfloor k/\tau\rfloor} \,.$$

For $q \in \mathbb{N}$, $k \in \mathbb{N}$, and the Markov chain $\{Z_n\}_{n\in\mathbb{N}}$ satisfying the uniform geometric ergodicity constraint **UGE** 1, we define the $\sigma$-algebras $\mathcal{F}_k = \sigma(Z_\ell, \ell \leq k)$ and $\mathcal{F}_{k+q}^+ = \sigma(Z_\ell, \ell \geq k + q)$. In such a scenario, using [13, Theorem 3.3], the respective $\beta$-mixing coefficient of $\mathcal{F}_k$ and $\mathcal{F}_{k+q}^+$ is bounded by

$$\beta(q) \equiv \beta(\mathcal{F}_k, \mathcal{F}_{k+q}^+) \leq \Delta(\mathrm{P}^k) = (1/4)^{\lfloor q/\tau\rfloor} \,.$$

We rely on the following useful version of Berbee's coupling lemma [3], which is due to [10, Lemma 4.1]:

**Theorem B.3** (Lemma 4.1 in [10])**.** *Let $X$ and $Y$ be two random variables taking their values in Borel spaces $\mathcal{X}$ and $\mathcal{Y}$, respectively, and let $U$ be a random variable with uniform distribution on $[0; 1]$ that is independent of $(X, Y)$. There exists a random variable $Y^\star = f(X, Y, U)$ where $f$ is a measurable function from $\mathcal{X} \times \mathcal{Y} \times [0, 1]$ to $\mathcal{Y}$, such that:*

    *1. $Y^\star$ is independent of $X$ and has the same distribution as $Y$;*

    *2. $\mathbb{P}(Y^\star \neq Y) = \beta(\sigma(X), \sigma(Y))$.*

Let us now consider the extended measurable space $\tilde{Z}_{\mathbb{N}} = Z^{\mathbb{N}} \times [0,1]$, equipped with the $\sigma$-field $\tilde{\mathcal{Z}}_{\mathbb{N}} = \mathcal{Z}^{\otimes \mathbb{N}} \otimes \mathcal{B}([0,1])$. For each probability measure $\xi$ on $(Z, \mathcal{Z})$, we consider the probability measure $\tilde{\mathbb{P}}_\xi = \mathbb{P}_\xi \otimes \mathbf{Unif}([0,1])$ and denote by $\tilde{\mathbb{E}}_\xi$ the corresponding expected value. Finally, we denote by $(\tilde{Z}_k)_{k \in \mathbb{N}}$ the canonical process $\tilde{Z}_k \colon ((z_i)_{i \in \mathbb{N}}, u) \in \tilde{Z}_{\mathbb{N}} \mapsto z_k$ and $U \colon ((z_i)_{i \in \mathbb{N}}, u) \in \tilde{Z}_{\mathbb{N}} \mapsto u$. Under $\tilde{\mathbb{P}}_\xi$, $\{\tilde{Z}_k\}_{k \in \mathbb{N}}$ is by construction a Markov chain with initial distribution $\xi$ and Markov kernel P independent of $U$. Moreover, the distribution of $U$ under $\tilde{\mathbb{P}}_\xi$ is uniform over $[0,1]$. Using the above construction, we obtain a useful blocking lemma, which is also stated in [10].

**Lemma B.4.** *Assume UGE 1, let $q \in \mathbb{N}$ and $\xi$ be a probability measure on $(Z, \mathcal{Z})$. Then, there exists a random process $(\tilde{Z}_k^\star)_{k \in \mathbb{N}}$ defined on $(\tilde{Z}_{\mathbb{N}}, \tilde{\mathcal{Z}}_{\mathbb{N}}, \tilde{\mathbb{P}}_\xi)$ such that for any $k \in \mathbb{N}$, it holds:*

1. *For any $i$, vector $V_i^\star = (\tilde{Z}_{iq+1}^\star, \ldots, \tilde{Z}_{iq+q}^\star)$ has the same distribution as $V_i = (Z_{iq+1}, \ldots, Z_{iq+q})$ under $\tilde{\mathbb{P}}_\xi$;*

2. *The sequences $(V_{2i}^\star)_{i \geq 0}$ and $(V_{2i+1}^\star)_{i \geq 0}$ are i.i.d. ;*

3. *For any $i$, $\tilde{\mathbb{P}}_\xi(V_i \neq V_i^\star) \leq \beta(q)$;*

*Proof.* The proof follows from Theorem B.3 and the relations between **UGE** 1 and $\beta$-mixing coefficient, see e.g. [13, Theorem 3.3]. $\qquad\square$

## C Federated Linear Stochastic Approximation with Control Variates

### C.1 Technical Lemmas

**Lemma C.1.** *Assume A 1 and A3. Recall $C_{\eta,H}^{(t,c)} = \sum_{h=1}^H \Gamma_{t,h+1:H}^{(c,\eta)}$ Then it holds that*

$$\mathbb{E}\left[\left\|I - \frac{1}{H}C_{\eta,H}^{(t,c)}\right\|^2\right] \leq \frac{\eta^2 H^2}{4}\left\{C_{\mathbf{A}}^2 + \|\Sigma_{\tilde{\mathbf{A}}}^c\|\right\} .$$

*Proof.* We rewrite $I - C_{\eta,H}^{(t,c)}$ using Lemma D.1 as

$$I - \frac{1}{H}C_{\eta,H}^{(t,c)} = \frac{1}{H}\sum_{h=1}^H\left\{I - \Gamma_{t,h+1:H}^{(c,\eta)}\right\} = \frac{\eta}{H}\sum_{h=1}^H\sum_{\ell=h+1}^H \mathbf{A}^c(Z_{t,\ell}^c)\Gamma_{t,\ell+1:H}^{(c,\eta)} ,$$

which can then be decomposed as

$$I - \frac{1}{H}C_{\eta,H}^{(t,c)} = \frac{\eta}{H}\sum_{h=1}^H\sum_{\ell=h+1}^H \bar{\mathbf{A}}^c\Gamma_{t,\ell+1:H}^{(c,\eta)} + \frac{\eta}{H}\sum_{h=1}^H\sum_{\ell=h+1}^H\left\{\mathbf{A}^c(Z_{t,\ell}^c) - \bar{\mathbf{A}}^c\right\}\Gamma_{t,\ell+1:H}^{(c,\eta)} .$$

Minkowski's inequality and A3 give $\mathbb{E}^{1/2}\left[\|\frac{\eta}{H}\sum_{h=1}^H\sum_{\ell=h+1}^H\bar{\mathbf{A}}^c\Gamma_{t,\ell+1:H}^{(c,\eta)}\|^2\right] \leq \frac{\eta H}{2}\|\bar{\mathbf{A}}^c\|$. The second term has a reverse martingale structure, and we thus have

$$\mathbb{E}\left[\left\|I - \frac{1}{H}C_{\eta,H}^{(t,c)}\right\|^2\right] \leq \frac{\eta^2 H^2}{4}\left\{C_{\mathbf{A}}^2 + \|\Sigma_{\tilde{\mathbf{A}}}^c\|\right\} ,$$

which is the result of the lemma. $\qquad\square$

**Lemma C.2.** *Assume A 1 and A3. Recall $\tilde{C}_{t+1}^c = \sum_{h=1}^H\left\{\Gamma_{t,h+1:H}^{(c,\eta)} - (I - \bar{\mathbf{A}}^c)^{H-h}\right\}$. Then we have*

$$\mathbb{E}\left[\|\tilde{C}_{t+1}^c\|^2\right] \leq \eta^2 H^4\left\{C_{\mathbf{A}}^2 + \|\Sigma_{\tilde{\mathbf{A}}}^c\|\right\} .$$

*Proof.* We start by recalling the definition of $\tilde{C}_{t+1}^c$, that is

$$\tilde{C}_{t+1}^c = C_{t+1}^c - \frac{1}{N}\sum_{c=1}^N \mathbb{E}[C_{t+1}^{\tilde{c}}] = \frac{1}{N}\sum_{\tilde{c}=1}^N\sum_{h=1}^H\left\{\Gamma_{t,h+1:H}^{(c,\eta)} - (I - \bar{\mathbf{A}}^c)^{H-h}\right\} .$$

Using Lemma D.1, we have

$$\widetilde{C}_{t+1}^c = \frac{\eta}{N} \sum_{\tilde{c}=1}^N \sum_{h=1}^H \sum_{\ell=h}^H \Gamma_{t,h+1:\ell}^{(c,\eta)} \left\{ \mathbf{A}^c(Z_{t,\ell}^c) - \bar{\mathbf{A}}^{\tilde{c}} \right\} (\mathrm{I} - \bar{\mathbf{A}}^c)^{H-\ell-1} .$$

By Minkowski's inequality and Assumption A3, we obtain

$$\mathbb{E}^{1/2} \left[ \|\widetilde{C}_{t+1}^c\|^2 \right] = \frac{\eta}{N} \sum_{\tilde{c}=1}^N \sum_{h=1}^H \sum_{\ell=h}^H \mathbb{E}^{1/2} \left[ \|\mathbf{A}^c(Z_{t,\ell}^c) - \bar{\mathbf{A}}^{\tilde{c}}\|^2 \right] .$$

Now, we notice that

$$\mathbb{E} \left[ \|\mathbf{A}^c(Z_{t,\ell}^c) - \bar{\mathbf{A}}^{\tilde{c}}\|^2 \right] = \mathbb{E} \left[ \|\mathbf{A}^c(Z_{t,\ell}^c) - \bar{\mathbf{A}}^c\|^2 \right] + \|\bar{\mathbf{A}}^c - \bar{\mathbf{A}}^{\tilde{c}}\|^2 \leq \mathrm{C}_{\mathbf{A}}^2 + \|\Sigma_{\bar{\mathbf{A}}}^c\| ,$$

and the result of the lemma follows. $\square$

**Lemma C.3.** *Assume A1 and A3. Recall* $C_{\eta,H}^{(t,c)} = \sum_{h=1}^H \Gamma_{t,h+1:H}^{(c,\eta)}$ *then*

$$\mathbb{E}[\|\widetilde{\Gamma}_{t+1}^c\|] \leq 2\eta H \left\{ \mathrm{C}_{\mathbf{A}}^2 + \|\Sigma_{\bar{\mathbf{A}}}^c\| \right\} .$$

*Proof.* Denote $\mathbf{A}_h^c = \mathbf{A}^c(Z_{t+1,h}^c)$,

$$\widetilde{\Gamma}_{t+1}^c = \Gamma_{t+1} - \Gamma_{t+1}^c$$
$$= \frac{1}{N} \sum_{c=1}^N \left\{ \prod_{h=1}^H (\mathrm{I} - \eta\mathbf{A}_h^c) - \prod_{h=1}^H (\mathrm{I} - \eta\mathbf{A}_h^c) \right\} .$$

Using Lemma D.1, we can rewrite

$$\widetilde{\Gamma}_{t+1}^c = \frac{\eta}{N} \sum_{c=1}^N \sum_{k=1}^H \left\{ \prod_{h=1}^{k-1} (\mathrm{I} - \eta\mathbf{A}_h^c) \right\} \{\mathbf{A}_k^c - \mathbf{A}_k^c\} \left\{ \prod_{h=k+1}^H (\mathrm{I} - \eta\mathbf{A}_h^c) \right\} .$$

Using triangle inequality and the fact that $\mathbf{A}_h^c$'s are independent from each other, we have

$$\mathbb{E}[\|\widetilde{\Gamma}_{t+1}^c\|] = \frac{\eta}{N} \sum_{c=1}^N \sum_{k=1}^H \mathbb{E}\left[ \Big\| \prod_{h=1}^{k-1} (\mathrm{I} - \eta\mathbf{A}_h^c) \Big\| \right] \mathbb{E}\left[ \Big\| \mathbf{A}_k^c - \mathbf{A}_k^c \Big\| \right] \mathbb{E}\left[ \Big\| \prod_{h=k+1}^H (\mathrm{I} - \eta\mathbf{A}_h^c) \Big\| \right] .$$

By triangle inequality, and using the definition of $\mathrm{C}_{\mathbf{A}}$ and $\|\Sigma_{\bar{\mathbf{A}}}^c\|$, we have $\mathbb{E}[\|\mathbf{A}_k^c - \mathbf{A}_k^c\|] \leq \mathbb{E}[\|\mathbf{A}_k^c - \bar{\mathbf{A}}^c\| + \|\bar{\mathbf{A}}^c - \bar{\mathbf{A}}^c\| + \|\mathbf{A}_k^c - \bar{\mathbf{A}}^c\|] \leq 2\,\mathrm{C}_{\mathbf{A}} + 2\|\Sigma_{\bar{\mathbf{A}}}^c\|$. Therefore, we obtain

$$\mathbb{E}[\|\widetilde{\Gamma}_{t+1}^c\|] \leq 2\eta \sum_{k=1}^H (1 - \eta a)^{H-1} \left( \mathrm{C}_{\mathbf{A}} + \|\Sigma_{\bar{\mathbf{A}}}^c\| \right) ,$$

and the result follows. $\square$

## C.2  Proof

The linear structure of SCAFFLSA's updates allow to decompose the updates between a transient term, and a fluctuation term. To materialize this, we define the following *virtual* parameters

$$\check{\theta}_0 = \theta_0 , \quad \check{\theta}_{0,0}^c = \check{\theta}_0 , \text{ and } \check{\xi}_0^c = \xi_0^c , \quad \text{for all } c \in \{1, \dots, N\} .$$

These parameters are updated similarly to $\theta_t$'s and $\xi_t^c$'s, although without the last fluctuation term. For the virtual parameter $\check{\theta}$, the update is similar to (8), as follows

$$\check{\theta}_{t,h}^c - \theta_\star = (\mathrm{I} - \eta\mathbf{A}^c(Z_{t,h}^c))(\check{\theta}_{t,h-1}^c - \theta_\star) + \eta(\xi_t^c - \xi_\star^c) ,$$

which gives, after $H$ local updates,

$$\check{\theta}_{t,H}^c - \theta_\star = \Gamma_{t+1}^c(\check{\theta}_t^c - \theta_\star) + \eta C_{t+1}^c(\xi_t^c - \xi_\star^c) ,$$

where we recall $\Gamma_{t+1}^c = \prod_{h=1}^H (\mathrm{I} - \eta \mathbf{A}(Z_{t,h}^c))$ and $C_{t+1}^c = \sum_{h=1}^H \Gamma_{t,h+1:H}^{(c,\eta)}$. The virtual parameters obtained after $H$ local updates are then aggregated as

$$\check\theta_{t+1} = \frac{1}{N} \sum_{c=1}^N \check\theta_{t,H}^c \ .$$

This is then used to define the virtual control variates, similarly to (9),

$$\check\xi_{t+1}^c = \check\xi_t^c + \frac{1}{\eta H}(\check\theta_{t+1} - \check\theta_{t,H}^c) \ .$$

These updates can be summarized over one block, which gives

$$\check\theta_{t+1} - \check\theta_\star = \Gamma_{t+1}(\check\theta_t - \theta_\star) + \frac{\eta}{N} \sum_{c=1}^N C_{t+1}^c(\check\xi_t^c - \xi_\star^c) \ ,$$

$$\check\xi_{t+1}^c - \xi_\star^c = \frac{1}{\eta H}(\Gamma_{t+1} - \Gamma_{t+1}^c)(\check\theta_t - \theta_\star) + \big(\mathrm{I} - \frac{1}{H}C_{t+1}^c\big)(\check\xi_t^c - \xi_\star^c) + \frac{1}{HN} \sum_{\tilde c=1}^N C_{t+1}^{\tilde c}(\check\xi_t^{\tilde c} - \xi_\star^{\tilde c}) \ .$$

The analysis of SCAFFLSA can then be decomposed into (i) analysis of the "transient" virtual iterates $\check\theta_t$'s and $\check\xi_t^c$'s, and (ii) analysis of the fluctuations $\theta_t - \check\theta_t$ and $\xi_t^c - \check\xi_t^c$.

**Analysis of the Transient Term.** First, we analyze the convergence of the virtual variables $\check\theta_t$ and $\check\xi_t^c$ for $t \geq 0$ and $c \in \{1,\dots,N\}$. Consider the Lyapunov function,

$$\psi_t = \|\check\theta_t - \theta_\star\|^2 + \frac{\eta^2 H^2}{N} \sum_{c=1}^N \|\check\xi_t^c - \xi_\star^c\|^2 \ ,$$

which is naturally defined as the error in $\theta_\star$ estimation using the virtual iterates, on communication rounds, and the average error on the virtual control variates.

**Theorem C.4.** *Assume A1 and A3. Let $\eta, H$ such that $\eta a H \leq 1$, and $H \leq \frac{a}{2\eta\{\mathrm{C}+\|\Sigma\|\}}$, and set $\xi_0^c = 0$ for all $c \in [N]$. Then, the sequence $(\psi_t)_{t\in\mathbb{N}}$ satisfies, for all $t \geq 0$,*

$$\mathbb{E}[\psi_t] \leq \left(1 - \frac{\eta a H}{4}\right)^t \mathbb{E}[\psi_0] \ ,$$

*where $\psi_0 = \|\theta_0 - \theta_\star\|^2 + \frac{\eta H}{N} \sum_{c=1}^N \|\bar{\mathbf{A}}^c(\theta_\star^c - \theta_\star)\|^2$.*

*Proof.* **Expression of the Lyapunov function.** Since the sum virtual control variates is $\sum_{t=1}^N \check\xi_t^c = \sum_{t=1}^N \check\xi_\star^{\,c} = 0$, we have $\check\theta_{t+1} = \frac{1}{N}\sum_{c=1}^N \check\theta_{t,H}^c = \frac{1}{N}\sum_{c=1}^N \check\theta_{t,H}^c - \eta H(\xi_t^c - \xi_\star^c)$. Applying Lemma D.3, we obtain

$$\|\check\theta_{t+1} - \theta_\star\|^2 = \|\frac{1}{N}\sum_{c=1}^N \check\theta_{t,H}^c - \theta_\star - \eta H(\check\xi_t^c - \xi_\star^c)\|^2$$

$$= \frac{1}{N}\sum_{c=1}^N \|\check\theta_{t,H}^c - \theta_\star - \eta H(\check\xi_t^c - \xi_\star^c)\|^2 - \frac{1}{N}\sum_{c=1}^N \|\check\theta_{t+1} - \check\theta_{t,H}^c + \eta H(\check\xi_t^c - \xi_\star^c)\|^2$$

$$= \frac{1}{N}\sum_{c=1}^N \|\check\theta_{t,H}^c - \theta_\star - \eta H(\check\xi_t^c - \xi_\star^c)\|^2 - \frac{\eta^2 H^2}{N}\sum_{c=1}^N \|\check\xi_{t+1}^c - \xi_\star^c\|^2 \ ,$$

since $\check\xi_{t+1}^c = \check\xi_t^c + \frac{1}{\eta H}(\check\theta_{t+1} - \check\theta_{t,H}^c)$. Adding $\frac{\eta H}{N}\sum_{c=1}^N \|\check\xi_{t+1}^c - \xi_\star^c\|^2$ on both sides, we obtain

$$\psi_{t+1} = \frac{1}{N}\sum_{c=1}^N \|\check\theta_{t,H}^c - \theta_\star - \eta H(\check\xi_t^c - \xi_\star^c)\|^2 \ , \tag{38}$$

where we defined $C^c_{\eta,H} = \sum_{h=1}^{H} \Gamma^{(c,\eta)}_{t,h+1:H}$. In the following, we will use the filtration of all events up to step $t$, $\mathcal{F}_t := \sigma(Z^c_{s,h}, 0 \le s \le t, 0 \le h \le H, 1 \le c \le N)$.

Using Young's inequality, and Assumption A3, we can bound

$$
\begin{aligned}
\mathbb{E}[\|\check\theta^c_{t,H} - \theta_\star - \eta H(\check\xi^c_t - \xi^c_\star)\|^2] &= \|\Gamma^{(c,\eta)}_{t,1:H}(\check\theta_t - \theta_\star) - \eta H(\mathrm{I} - \tfrac{1}{H}C^c_{\eta,H})(\check\xi^c_t - \xi^c_\star)\|^2 \\
&\le \mathbb{E}[(1+\alpha_0)\|\Gamma^{(c,\eta)}_{t,1:H}(\check\theta_t - \theta_\star)\|^2] + (1+\alpha_0^{-1})\eta^2 H^2 \mathbb{E}[\|(\mathrm{I} - \tfrac{1}{H}C^c_{\eta,H})(\check\xi^c_t - \xi^c_\star)\|^2] \\
&\le (1+\alpha_0)(1-\eta a)^{2H} \mathbb{E}[\|\check\theta_t - \theta_\star\|^2] + (1+\alpha_0^{-1})\eta^2 H^2 \mathbb{E}[\|(\mathrm{I} - \tfrac{1}{H}C^c_{\eta,H})(\check\xi^c_t - \xi^c_\star)\|^2] .
\end{aligned}
$$

Using Lemma C.1, we have

$$
\mathbb{E}[\|(\mathrm{I} - \tfrac{1}{H}C^c_{\eta,H})(\check\xi^c_t - \xi^c_\star)\|^2] \le \frac{\eta^2 H^2}{4}\left\{\mathrm{C}^2_\mathbf{A} + \|\Sigma^c_{\widetilde{\mathbf{A}}}\|\right\} \mathbb{E}[\|\check\xi^c_t - \xi^c_\star\|^2] .
$$

We thus obtain, for $H$ such that $\eta a H \le 1$, and after setting $\alpha_0 = \frac{\eta a H}{2}$ and using the facts that $(1-\eta a H)(1+\alpha_0) \le 1 - \frac{\eta a H}{2}$ and $1 + \alpha_0^{-1} \le 2\alpha_0^{-1}$,

$$
\begin{aligned}
\mathbb{E}[\|\check\theta^c_{t,H} &- \theta_\star - \eta H(\check\xi^c_t - \xi^c_\star)\|^2] \\
&\le \left(1 - \frac{\eta a H}{2}\right)\mathbb{E}[\|\check\theta_t - \theta_\star\|^2] + \frac{1}{\eta a H}\left\{\mathrm{C}^2_\mathbf{A} + \|\Sigma^c_{\widetilde{\mathbf{A}}}\|\right\} \eta^4 H^4 \mathbb{E}[\|\check\xi^c_t - \xi^c_\star\|^2] \\
&= \left(1 - \frac{\eta a H}{2}\right)\mathbb{E}[\|\check\theta_t - \theta_\star\|^2] + \frac{\eta H}{a}\left\{\mathrm{C}^2_\mathbf{A} + \|\Sigma^c_\varepsilon\|\right\} \eta^2 H^2 \mathbb{E}[\|\check\xi^c_t - \xi^c_\star\|^2] .
\end{aligned}
$$

Then, since $\frac{\eta H}{a}\left\{\mathrm{C}^2_\mathbf{A} + \|\Sigma^c_\varepsilon\|\right\} \le \frac{1}{2}$, we obtain

$$
\mathbb{E}[\|\check\theta^c_{t,H} - \theta_\star - \eta H(\check\xi^c_t - \xi^c_\star)\|^2] \le \left(1 - \frac{\eta a H}{2}\right)\mathbb{E}\left[\|\check\theta_t - \theta_\star\|^2 + \eta^2 H^2\|\check\xi^c_t - \xi^c_\star\|^2\right], \quad (39)
$$

and the result follows by plugging (39) back in (38). $\qquad\square$

**Analysis of the Fluctuations.** To study the fluctuations, we define the following quantities,

$$
\widetilde\theta_t = \theta_t - \check\theta_t , \quad \text{and} \quad \widetilde\xi^c_t = \xi^c_t - \check\xi^c_t , \quad \text{for } t \ge 0 , \text{ and } c \in \{1,\dots,N\} .
$$

Our analysis is based on a careful study of the recurrence between variances and covariances of parameters and control variates. We thus start by deriving recurrence properties on these quantities. From the update of $\theta_t$, we have,

$$
\begin{aligned}
\theta_{t+1} - \theta_\star &= \Gamma_{t+1}(\theta_t - \theta_\star) + \frac{\eta}{N}\sum_{\tilde c=1}^{N} \widetilde C^{\tilde c}_{t+1}(\xi^{\tilde c}_t - \xi^{\tilde c}_\star) - \eta\bar\varepsilon_{t+1} \\
&= \check\theta_{t+1} - \theta_\star + \Gamma_{t+1}\widetilde\theta_t + \frac{\eta}{N}\sum_{\tilde c=1}^{N} \widetilde C^{\tilde c}_{t+1}\widetilde\xi^{\tilde c}_t - \eta\bar\varepsilon_{t+1} ,
\end{aligned}
$$

which can be rewritten as a recursive update of the fluctuations

$$
\widetilde\theta_{t+1} = \Gamma_{t+1}\widetilde\theta_t + \frac{\eta}{N}\sum_{\tilde c=1}^{N} \widetilde C^{\tilde c}_{t+1}\widetilde\xi^{\tilde c}_t - \eta\bar\varepsilon_{t+1} . \quad (40)
$$

Similarly, we have, for the fluctuations of the control variates

$$
\widetilde\xi^c_{t+1} = \frac{1}{\eta H}(\Gamma_{t+1} - \Gamma^c_{t+1})\widetilde\theta_t + \left(\mathrm{I} - \frac{1}{H}C^c_{t+1}\right)\widetilde\xi^c_t + \frac{1}{NH}\sum_{\tilde c=1}^{N} \widetilde C^{\tilde c}_{t+1}\widetilde\xi^{\tilde c}_t - \frac{1}{H}(\bar\varepsilon_{t+1} - \varepsilon^c_{t+1}) .
$$

Remark that, for all $t \ge 0$, $\widetilde\theta_t$ and $\widetilde\xi^c_t$'s are sums of (random) linear operations computed on zero-mean vectors, that are independent from these linear operations. Thus, for all $t \ge 0$ and all $c \in \{1,\dots,N\}$ we have

$$
\mathbb{E}[\widetilde\theta_t] = 0 \quad , \quad \mathbb{E}[\widetilde\xi^c_t] = 0 . \quad (41)
$$

We now aim at recursively finding a sequence of upper bounds $\{b_t^{(\theta,\theta)}, b^{(\theta,\xi)}, b^=, b^{\neq}\}_{t \geq 0}$ such that, for all $t \geq 0$, $c, c' \in \{1, \ldots, N\}$ such that $c \neq c'$,

$$\left\| \mathbb{E}\left[ (\widetilde{\theta}_t)(\widetilde{\theta}_t)^\top \right] \right\| \leq b_t^{(\theta,\theta)} \ ,$$

$$\left\| \mathbb{E}\left[ (\widetilde{\theta}_t)(\widetilde{\xi}_t^c)^\top \right] \right\| \leq b_t^{(\theta,\xi)} \text{ and } \left\| \mathbb{E}\left[ (\widetilde{\xi}_t^c)(\widetilde{\theta}_t)^\top \right] \right\| \leq b_t^{(\theta,\xi)} \ ,$$

$$\left\| \mathbb{E}\left[ (\widetilde{\xi}_t^c)(\widetilde{\xi}_t^c)^\top \right] \right\| \leq b_t^{\neq} \ ,$$

$$\left\| \mathbb{E}\left[ (\widetilde{\xi}_t^c)(\widetilde{\xi}_t^c)^\top \right] \right\| \leq b_t^= \ .$$

*(Initialization.)* For $t = 0$, nothing is random so the fluctuations are zero, and $b_0^{(\theta,\theta)} = b_0^{(\theta,\xi)} = b^= = b^{\neq} = 0$. We also study the first iteration of SCAFFLSA. In the following lemma, we give upper bounds on the variances and covariances of the parameters obtained after one iteration.

**Lemma C.5.** *Assume A1 and A3, then the first iterate of SCAFFLSA satisfy the following inequalities*

$$b_1^{(\theta,\theta)} = \frac{\eta^2 H}{N} \|\Sigma_\omega\| \ , \quad b_1^= = \frac{N-1}{NH} \|\Sigma_\omega^c\| \ , \quad b_1^{(\theta,\xi)} = \frac{2\eta}{N} \|\Sigma_\omega\| \ , \quad b_1^{\neq} = \frac{3}{NH} \|\Sigma_\omega\| \ .$$

*Proof. (Value of $b_1^{(\theta,\theta)}$.)* From the definition of $\widetilde{\theta}_1$, we have $\widetilde{\theta}_1 = \frac{\eta}{N} \sum_{c=1}^N \sum_{h=1}^H \Gamma_{1,h+1:H}^{(c,\eta)} \omega^c(Z_{1,h}^c)$. By independence of the agents, and since $\mathbb{E}[\Gamma_{1,h+1:H}^{(c,\eta)} \omega(Z_{1,h}^c)] = 0$ for all $c \in \{1, \ldots, N\}$, and for all $h \in \{0, \ldots, H-1\}$,

$$\mathbb{E}[(\widetilde{\theta}_1)(\widetilde{\theta}_1)^\top] = \frac{\eta^2}{N^2} \sum_{c=1}^N \sum_{h=1}^H \mathbb{E}\left[ \Gamma_{1,h+1:H}^{(c,\eta)} \omega^c(Z_{1,h}^c) \omega^c(Z_{1,h}^c)^\top (\Gamma_{1,h+1:H}^{(c,\eta)})^\top \right]$$

$$= \frac{\eta^2}{N^2} \sum_{c=1}^N \sum_{h=1}^H \mathbb{E}\left[ \Gamma_{1,h+1:H}^{(c,\eta)} \Sigma_\omega^c (\Gamma_{1,h+1:H}^{(c,\eta)})^\top \right] \ ,$$

where the second equality comes from the fact that, for all $h \in \{1, \ldots, H-1\}$, the matrix $\Gamma_{1,h+1:H}^{(c,\eta)}$ and the vector $\omega^c(Z_{1,h}^c)$ are independent. Triangle inequality, Jensen's inequality, and definition of the operator norm then give

$$\left\| \mathbb{E}[(\widetilde{\theta}_1)(\widetilde{\theta}_1)^\top] \right\| \leq \frac{\eta^2}{N^2} \sum_{c=1}^N \sum_{h=1}^H \left\| \mathbb{E}\left[ \Gamma_{1,h+1:H}^{(c,\eta)} \Sigma_\omega^c (\Gamma_{1,h+1:H}^{(c,\eta)})^\top \right] \right\|$$

$$\leq \frac{\eta^2}{N^2} \sum_{c=1}^N \sum_{h=1}^H \mathbb{E}\left[ \left\| \Gamma_{1,h+1:H}^{(c,\eta)} \Sigma_\omega^c (\Gamma_{1,h+1:H}^{(c,\eta)})^\top \right\| \right]$$

$$\leq \frac{\eta^2}{N^2} \sum_{c=1}^N \sum_{h=1}^H \mathbb{E}\left[ \left\| \Gamma_{1,h+1:H}^{(c,\eta)} \right\|^2 \right] \|\Sigma_\omega^c\| \ .$$

Assumption A3 ensures that $\mathbb{E}\left[ \left\| \Gamma_{1,h+1:H}^{(c,\eta)} \right\|^2 \right] \leq 1$, and we have

$$\left\| \mathbb{E}[(\widetilde{\theta}_1)(\widetilde{\theta}_1)^\top] \right\| \leq \frac{\eta^2 H}{N^2} \sum_{c=1}^N \|\Sigma_\omega^c\| \leq \frac{\eta^2 H}{N} \|\Sigma_\omega\| \ .$$

*(Value of $b_1^=$.)* Let $c \in \{1, \ldots, N\}$. The definition of $\widetilde{\xi}_1^c$ gives the following expression for the fluctuation $\widetilde{\xi}_1^c = \frac{1}{NH} \sum_{\widetilde{c}=1}^N \sum_{h=1}^H \left\{ \Gamma_{1,h+1:H}^{(\widetilde{c},\eta)} \omega^{\widetilde{c}}(Z_{1,h}^{\widetilde{c}}) \right\} - \frac{1}{H} \sum_{h=1}^H \Gamma_{1,h+1:H}^{(c,\eta)} \omega^c(Z_{1,h}^c)$. Therefore,

we have

$$\mathbb{E}[(\widetilde{\xi}_1^c)(\widetilde{\xi}_1^c)^\top] = \mathbb{E}\left[\left(\frac{1}{NH}\sum_{\tilde{c}=1}^N\sum_{h=1}^H\left\{\Gamma_{1,h+1:H}^{(\tilde{c},\eta)}\omega^{\tilde{c}}(Z_{1,h}^{\tilde{c}})\right\} - \frac{1}{H}\sum_{h=1}^H\Gamma_{1,h+1:H}^{(c,\eta)}\omega^c(Z_{1,h}^c)\right)\right.$$
$$\left.\times\left(\frac{1}{NH}\sum_{\tilde{c}=1}^N\sum_{h=1}^H\left\{(\omega^{\tilde{c}}(Z_{1,h}^{\tilde{c}}))^\top(\Gamma_{1,h+1:H}^{(\tilde{c},\eta)})^\top\right\} - \frac{1}{H}\sum_{h=1}^H(\omega^c(Z_{1,h}^c))^\top(\Gamma_{1,h+1:H}^{(c,\eta)})^\top\right)\right].$$

With similar arguments as above, we have

$$\mathbb{E}[(\widetilde{\xi}_1^c)(\widetilde{\xi}_1^c)^\top] \leq \frac{1}{N^2H^2}\sum_{\tilde{c}=1}^N\mathbb{E}\left[\Gamma_{1,h+1:H}^{(\tilde{c},\eta)}\Sigma_\omega^{\tilde{c}}(\Gamma_{1,h+1:H}^{(\tilde{c},\eta)})^\top\right] + \frac{N-2}{NH^2}\mathbb{E}\left[\Gamma_{1,h+1:H}^{(c,\eta)}\Sigma_\omega^c(\Gamma_{1,h+1:H}^{(c,\eta)})^\top\right].$$

Assuming $N \geq 2$, triangle inequality gives

$$\left\|\mathbb{E}[(\widetilde{\xi}_1^c)(\widetilde{\xi}_1^c)^\top]\right\| \leq \frac{1}{NH}\|\Sigma_\omega\| + \frac{N-2}{NH}\|\Sigma_\omega\| = \frac{N-1}{NH}\|\Sigma_\omega\|.$$

*(Value of $b_1^{(\theta,\xi)}$.)* For the covariance of $\widetilde{\xi}_1^c$ and $\widetilde{\theta}_1$, we have

$$\mathbb{E}\left[(\widetilde{\theta}_1)(\widetilde{\xi}_1^c)^\top\right]$$
$$= \mathbb{E}\left[\left(\frac{\eta}{N}\sum_{\tilde{c}=1}^N\sum_{h=1}^H\Gamma_{1,h+1:H}^{(\tilde{c},\eta)}\omega^{\tilde{c}}(Z_{1,h}^{\tilde{c}})\right)\right.$$
$$\left.\times\left(\frac{1}{NH}\sum_{\tilde{c}=1}^N\sum_{h=1}^H(\omega^{\tilde{c}}(Z_{1,h}^{\tilde{c}}))^\top(\Gamma_{1,h+1:H}^{(\tilde{c},\eta)})^\top - \frac{1}{H}\sum_{h=1}^H(\omega^c(Z_{1,h}^c))^\top(\Gamma_{1,h+1:H}^{(c,\eta)})^\top\right)\right]$$
$$= \mathbb{E}\left[\frac{\eta}{N^2H}\sum_{\tilde{c}=1}^N\sum_{h=1}^H\Gamma_{1,h+1:H}^{(\tilde{c},\eta)}\Sigma_\omega^{\tilde{c}}(\Gamma_{1,h+1:H}^{(\tilde{c},\eta)})^\top\right] - \mathbb{E}\left[\frac{\eta}{NH}\sum_{h=1}^H\Gamma_{1,h+1:H}^{(c,\eta)}\Sigma_\omega^c(\Gamma_{1,h+1:H}^{(c,\eta)})^\top\right].$$

As a result, we have

$$\left\|\mathbb{E}\left[\langle\widetilde{\theta}_1\,,\,\widetilde{\xi}_1^c\rangle\right]\right\| \leq \frac{2\eta}{N}\|\Sigma_\omega\|.$$

*(Value of $b^{\neq}$.)* Similarly to above, for $c \neq c'$, we have

$$\mathbb{E}[(\widetilde{\xi}_1^c)(\widetilde{\xi}_1^c)^\top] = \mathbb{E}\left[\left(\frac{1}{NH}\sum_{\tilde{c}=1}^N\sum_{h=1}^H\left\{\Gamma_{1,h+1:H}^{(\tilde{c},\eta)}\omega^{\tilde{c}}(Z_{1,h}^{\tilde{c}})\right\} - \frac{1}{H}\sum_{h=1}^H\Gamma_{1,h+1:H}^{(c,\eta)}\omega^c(Z_{1,h}^c)\right)\right.$$
$$\left.\times\left(\frac{1}{NH}\sum_{\tilde{c}=1}^N\sum_{h=1}^H\left\{(\omega^{\tilde{c}}(Z_{1,h}^{\tilde{c}}))^\top(\Gamma_{1,h+1:H}^{(\tilde{c},\eta)})^\top\right\} - \frac{1}{H}\sum_{h=1}^H(\omega^c(Z_{1,h}^c))^\top(\Gamma_{1,h+1:H}^{(c,\eta)})^\top\right)\right]$$
$$= \frac{1}{N^2H^2}\sum_{\tilde{c}=1}^N\mathbb{E}\left[\Gamma_{1,h+1:H}^{(\tilde{c},\eta)}\Sigma_\omega^{\tilde{c}}(\Gamma_{1,h+1:H}^{(\tilde{c},\eta)})^\top\right]$$
$$- \frac{1}{NH^2}\mathbb{E}\left[\Gamma_{1,h+1:H}^{(c,\eta)}\Sigma_\omega^c(\Gamma_{1,h+1:H}^{(c,\eta)})^\top\right] - \frac{1}{NH^2}\mathbb{E}\left[\Gamma_{1,h+1:H}^{(c,\eta)}\Sigma_\omega^c(\Gamma_{1,h+1:H}^{(c,\eta)})^\top\right].$$

Which results in the bound

$$\left\|\mathbb{E}[(\widetilde{\xi}_1^c)(\widetilde{\xi}_1^c)^\top]\right\| \leq \frac{3}{NH}\|\Sigma_\omega\|.$$

$\square$

**Lemma C.6.** *Let $\nu > 0$, and assume that $\eta H \left\{ C_{\mathbf{A}} + \|\Sigma_{\widetilde{\mathbf{A}}}^c\|^{1/2} \right\} \le \nu$ and $\eta^2 H^2 \left\{ C_{\mathbf{A}}^2 + \|\Sigma_{\widetilde{\mathbf{A}}}^c\| \right\} \le \nu$. Then the following inequalities hold for any $t \ge 0$,*

$$b_{t+1}^{(\theta,\theta)} \le (1 - \eta a)^{2H} b_t^{(\theta,\theta)} + \nu \eta H b_t^{(\theta,\xi)} + 2\nu \frac{\eta H}{N} b_t^{=} + \nu \eta^2 H^2 b_t^{\neq} + \frac{3\eta^2 H}{N} \|\Sigma_\omega\| \ ,$$

$$\eta H b_{t+1}^{(\theta,\xi)} \le 2\nu b_t^{(\theta,\theta)} + 3\nu \eta H b_t^{(\theta,\xi)} + \frac{2\nu}{N} \eta^2 H^2 b_t^{=} + 2\nu \eta^2 H^2 b_t^{\neq} + \frac{2\eta^2 H}{N} \|\Sigma_\omega\| \ ,$$

$$\eta^2 H^2 b_{t+1}^{=} \le 2\nu b_t^{(\theta,\theta)} + 3\nu \eta H b_t^{(\theta,\xi)} + 4\nu \eta^2 H^2 b_t^{=} + 3\nu \eta^2 H^2 b_t^{\neq} + \eta^2 H \|\Sigma_\omega\| \ ,$$

$$\eta^2 H^2 b_{t+1}^{\neq} \le 2\nu b_t^{(\theta,\theta)} + 3\nu \eta H b_t^{(\theta,\xi)} + \frac{3\nu}{N} \eta^2 H^2 b_t^{=} + 4\nu \eta^2 H^2 b_t^{\neq} + \frac{3\eta^2 H}{N} \|\Sigma_\omega\| \ ,$$

*Proof.* (Value of $b_{t+1}^{(\theta,\theta)}$.) Replacing $\widetilde{\theta}_{t+1}$ by its expression from (40), then expanding the expression, we have

$$(\widetilde{\theta}_{t+1})(\widetilde{\theta}_{t+1})^\top = \left( \Gamma_{t+1}\widetilde{\theta}_t + \frac{\eta}{N}\sum_{\tilde{c}=1}^N \widetilde{C}_{t+1}^{\tilde{c}}\widetilde{\xi}_t^{\tilde{c}} - \eta\bar{\varepsilon}_{t+1} \right)\left( \Gamma_{t+1}\widetilde{\theta}_t + \frac{\eta}{N}\sum_{\tilde{c}=1}^N \widetilde{C}_{t+1}^{\tilde{c}}\widetilde{\xi}_t^{\tilde{c}} - \eta\bar{\varepsilon}_{t+1} \right)^\top$$

$$= \Gamma_{t+1}\widetilde{\theta}_t\widetilde{\theta}_t^\top \Gamma_{t+1}^\top + \frac{\eta}{N}\sum_{\tilde{c}=1}^N \Gamma_{t+1}\widetilde{\theta}_t(\widetilde{\xi}_t^{\tilde{c}})^\top (\widetilde{C}_{t+1}^{\tilde{c}})^\top + \frac{\eta}{N}\sum_{\tilde{c}=1}^N \widetilde{C}_{t+1}^{\tilde{c}}\widetilde{\xi}_t^{\tilde{c}}(\widetilde{\theta}_t)^\top \Gamma_{t+1}^\top$$

$$+ \frac{\eta^2}{N^2}\sum_{\tilde{c}=1}^N\sum_{\tilde{c}=1}^N \widetilde{C}_{t+1}^{\tilde{c}}\widetilde{\xi}_t^{\tilde{c}}(\widetilde{\xi}_t^{\tilde{c}})^\top (\widetilde{C}_{t+1}^{\tilde{c}})^\top - \eta\bar{\varepsilon}_{t+1}\widetilde{\theta}_t^\top \Gamma_{t+1}^\top - \frac{\eta}{N}\sum_{\tilde{c}=1}^N \bar{\varepsilon}_{t+1}(\widetilde{\xi}_t^{\tilde{c}})^\top (\widetilde{C}_{t+1}^{\tilde{c}})^\top$$

$$- \eta\Gamma_{t+1}\widetilde{\theta}_t(\bar{\varepsilon}_{t+1})^\top - \frac{\eta}{N}\sum_{\tilde{c}=1}^N (\widetilde{C}_{t+1}^{\tilde{c}})(\widetilde{\xi}_t^{\tilde{c}})(\bar{\varepsilon}_{t+1})^\top + \eta^2(\bar{\varepsilon}_{t+1})(\bar{\varepsilon}_{t+1})^\top \ .$$

From the triangle inequality and Jensen's inequality, we have

$$\|\mathbb{E}[(\widetilde{\theta}_{t+1})(\widetilde{\theta}_{t+1})^\top]\| \le \mathbb{E}[\|\Gamma_{t+1}\|^2]\|\mathbb{E}[\widetilde{\theta}_t\widetilde{\theta}_t^\top]\| + \frac{2\eta}{N}\sum_{\tilde{c}=1}^N \mathbb{E}^{1/2}[\|\widetilde{C}_{t+1}^{\tilde{c}}\|^2]\|\mathbb{E}[\widetilde{\theta}_t\widetilde{\xi}_t^{\tilde{c}\top}]\|$$

$$+ \frac{\eta^2}{N^2}\sum_{\tilde{c}=1}^N \mathbb{E}[\|\widetilde{C}_{t+1}^{\tilde{c}}\|^2]\|\mathbb{E}[\widetilde{\xi}_t^{\tilde{c}}\widetilde{\xi}_t^{\tilde{c}\top}]\| + \frac{\eta^2}{N^2}\sum_{\tilde{c}=1}^N\sum_{\substack{\tilde{c}=1\\\tilde{c}\neq\tilde{c}}}^N \mathbb{E}^{1/2}[\|\widetilde{C}_{t+1}^{\tilde{c}}\|^2]\mathbb{E}^{1/2}[\|\widetilde{C}_{t+1}^{\tilde{c}}\|^2]\|\mathbb{E}[\widetilde{\xi}_t^{\tilde{c}}\widetilde{\xi}_t^{\tilde{c}\top}]\|$$

$$+ \|\mathbb{E}[\eta\Gamma_{t+1}^{(\mathrm{fl})}\widetilde{\theta}_t\bar{\varepsilon}_{t+1}^\top + \eta\bar{\varepsilon}_{t+1}\widetilde{\theta}_t^\top \Gamma_{t+1}^{(\mathrm{fl})\top}]\| + \frac{1}{N}\sum_{\tilde{c}=1}^N \|\mathbb{E}[\eta\widetilde{C}_{t+1}^{\tilde{c}}\widetilde{\xi}_t^{\tilde{c}}\bar{\varepsilon}_{t+1}^\top + \eta\bar{\varepsilon}_{t+1}\widetilde{\xi}_t^{\tilde{c}\top}\widetilde{C}_{t+1}^{\tilde{c}\top}]\|$$

$$+ \eta^2\|\mathbb{E}[\bar{\varepsilon}_{t+1}\bar{\varepsilon}_{t+1}^\top]\| \ .$$

Now, we have from (41) that $\mathbb{E}[\widetilde{\theta}_t] = \mathbb{E}[\widetilde{\xi}_t^c] = 0$. Thus, we have, for all $c \in \{1, \ldots, N\}$,

$$\|\mathbb{E}[\eta\Gamma_{t+1}^{(\mathrm{fl})}\widetilde{\theta}_t\bar{\varepsilon}_{t+1}^\top + \eta\bar{\varepsilon}_{t+1}\widetilde{\theta}_t^\top \Gamma_{t+1}^{(\mathrm{fl})\top}]\| = \|\mathbb{E}[\eta\Gamma_{t+1}^{(\mathrm{fl})}\mathbb{E}[\widetilde{\theta}_t]\bar{\varepsilon}_{t+1}^\top + \eta\bar{\varepsilon}_{t+1}\mathbb{E}[\widetilde{\theta}_t^\top]\Gamma_{t+1}^{(\mathrm{fl})\top}]\| = 0 \ ,$$

$$\|\mathbb{E}[\eta\widetilde{C}_{t+1}^{\tilde{c}}\widetilde{\xi}_t^{\tilde{c}}\bar{\varepsilon}_{t+1}^\top + \eta\bar{\varepsilon}_{t+1}\widetilde{\xi}_t^{\tilde{c}\top}\widetilde{C}_{t+1}^{\tilde{c}\top}]\| = \|\mathbb{E}[\eta\widetilde{C}_{t+1}^{\tilde{c}}\mathbb{E}[\widetilde{\xi}_t^{\tilde{c}}]\bar{\varepsilon}_{t+1}^\top + \eta\bar{\varepsilon}_{t+1}\mathbb{E}[\widetilde{\xi}_t^{\tilde{c}\top}]\widetilde{C}_{t+1}^{\tilde{c}\top}]\| = 0 \ .$$

Which results in the following inequality

$$\|\mathbb{E}[(\widetilde{\theta}_{t+1})(\widetilde{\theta}_{t+1})^\top]\| \le \mathbb{E}[\|\Gamma_{t+1}\|^2]\|\mathbb{E}[\widetilde{\theta}_t\widetilde{\theta}_t^\top]\| + \frac{2\eta}{N}\sum_{\tilde{c}=1}^N \mathbb{E}^{1/2}[\|\widetilde{C}_{t+1}^{\tilde{c}}\|^2]\|\mathbb{E}[\widetilde{\theta}_t\widetilde{\xi}_t^{\tilde{c}\top}]\|$$

$$+ \frac{\eta^2}{N^2}\sum_{\tilde{c}=1}^N \mathbb{E}[\|\widetilde{C}_{t+1}^{\tilde{c}}\|^2]\|\mathbb{E}[\widetilde{\xi}_t^{\tilde{c}}\widetilde{\xi}_t^{\tilde{c}\top}]\| + \frac{\eta^2}{N^2}\sum_{\tilde{c}=1}^N\sum_{\substack{\tilde{c}=1\\\tilde{c}\neq\tilde{c}}}^N \mathbb{E}^{1/2}[\|\widetilde{C}_{t+1}^{\tilde{c}}\|^2]\mathbb{E}^{1/2}[\|\widetilde{C}_{t+1}^{\tilde{c}}\|^2]\|\mathbb{E}[\widetilde{\xi}_t^{\tilde{c}}\widetilde{\xi}_t^{\tilde{c}\top}]\|$$

$$+ 3\eta^2\|\mathbb{E}[\bar{\varepsilon}_{t+1}\bar{\varepsilon}_{t+1}^\top]\| \ .$$

Using Lemma C.2, we obtain

$$\|\mathbb{E}[(\widetilde{\theta}_{t+1})(\widetilde{\theta}_{t+1})^\top]\| \le (1-\eta a)^{2H}\|\mathbb{E}[\widetilde{\theta}_t\widetilde{\theta}_t^\top]\|$$

$$+\frac{\eta}{N}\sum_{\tilde{c}=1}^{N}\eta H^2\left\{\mathrm{C_A}+\|\Sigma_{\widetilde{\mathbf{A}}}^c\|^{1/2}\right\}\|\mathbb{E}[\widetilde{\theta}_t\widetilde{\xi}_t^{\tilde{c}\,\top}]\| + \frac{2\eta^2}{N^2}\sum_{\tilde{c}=1}^{N}\eta^2 H^4\left\{\mathrm{C_A^2}+\|\Sigma_{\widetilde{\mathbf{A}}}^c\|\right\}\|\mathbb{E}[\widetilde{\xi}_t^{\tilde{c}}\widetilde{\xi}_t^{\tilde{c}\,\top}]\|$$

$$+\frac{\eta^2}{N^2}\sum_{\tilde{c}=1}^{N}\sum_{\substack{\bar{c}=1\\\bar{c}\neq\tilde{c}}}^{N}\eta^2 H^4\left\{\mathrm{C_A^2}+\|\Sigma_{\widetilde{\mathbf{A}}}^c\|\right\}\|\mathbb{E}[\widetilde{\xi}_t^{\tilde{c}}\widetilde{\xi}_t^{\bar{c}\,\top}]\| + 3\eta^2\|\mathbb{E}[\bar{\varepsilon}_{t+1}\bar{\varepsilon}_{t+1}^\top]\| \ .$$

Assuming $\eta H\left\{\mathrm{C_A}+\|\Sigma_{\widetilde{\mathbf{A}}}^c\|^{1/2}\right\}\le\nu$ and $\eta^2 H^2\left\{\mathrm{C_A^2}+\|\Sigma_{\widetilde{\mathbf{A}}}^c\|\right\}\le\nu$, we obtain

$$\|\mathbb{E}[(\widetilde{\theta}_{t+1})(\widetilde{\theta}_{t+1})^\top]\| \le (1-\eta a)^{2H}\|\mathbb{E}[\widetilde{\theta}_t\widetilde{\theta}_t^\top]\|+\nu\frac{\eta H}{N}\sum_{\tilde{c}=1}^{N}\|\mathbb{E}[\widetilde{\theta}_t\widetilde{\xi}_t^{\tilde{c}\,\top}]\|+2\nu\frac{\eta^2 H^2}{N^2}\sum_{\tilde{c}=1}^{N}\|\mathbb{E}[\widetilde{\xi}_t^{\tilde{c}}\widetilde{\xi}_t^{\tilde{c}\,\top}]\|$$

$$+\nu\frac{\eta^2 H^2}{N^2}\sum_{\tilde{c}=1}^{N}\sum_{\substack{\bar{c}=1\\\bar{c}\neq\tilde{c}}}^{N}\|\mathbb{E}[\widetilde{\xi}_t^{\tilde{c}}\widetilde{\xi}_t^{\bar{c}\,\top}]\| + \frac{3\eta^2 H}{N}\|\Sigma_\omega\| \ .$$

This gives our first inequality that links our upper bounds,

$$b_{t+1}^{(\theta,\theta)} \le (1-\eta a)^{2H}\,b_t^{(\theta,\theta)} + \nu\eta H b_t^{(\theta,\xi)} + 2\nu\frac{\eta H}{N}b_t^{=} + \nu\eta^2 H^2 b_t^{\neq} + \frac{3\eta^2 H}{N}\|\Sigma_\omega\| \ .$$

*(Value of $b_{t+1}^{(\theta,\xi)}$.)* As for $b_{t+1}^{(\theta,\theta)}$, we bound, for $c\in\{1,\dots,N\}$,

$$\widetilde{\theta}_{t+1}\widetilde{\xi}_{t+1}^{c}{}^\top$$

$$= \left(\Gamma_{t+1}\widetilde{\theta}_t + \frac{\eta}{N}\sum_{\tilde{c}=1}^{N}\widetilde{C}_{t+1}^{\tilde{c}}\widetilde{\xi}_t^{\tilde{c}} - \eta\bar{\varepsilon}_{t+1}\right)$$

$$\times\left(\frac{1}{\eta H}\widetilde{\Gamma}_{t+1}^{c}\widetilde{\theta}_t + \left(\mathrm{I}-\frac{1}{H}C_{t+1}^c\right)\widetilde{\xi}_t^c + \frac{1}{NH}\sum_{\tilde{c}=1}^{N}\widetilde{C}_{t+1}^{\tilde{c}}\widetilde{\xi}_t^{\tilde{c}} - \frac{1}{H}\widetilde{\varepsilon}_{t+1}^c\right)^\top$$

$$= \frac{1}{\eta H}\Gamma_{t+1}\widetilde{\theta}_t\widetilde{\theta}_t^\top\widetilde{\Gamma}_{t+1}^{c}{}^\top + \frac{1}{NH}\sum_{\tilde{c}=1}^{N}\widetilde{C}_{t+1}^{\tilde{c}}\widetilde{\xi}_t^{\tilde{c}}\widetilde{\theta}_t^\top\widetilde{\Gamma}_{t+1}^{c}{}^\top - \frac{1}{H}\bar{\varepsilon}_{t+1}\widetilde{\theta}_t^\top\widetilde{\Gamma}_{t+1}^{c}{}^\top$$

$$+\Gamma_{t+1}\widetilde{\theta}_t\widetilde{\xi}_t^{c}{}^\top\left(\mathrm{I}-\frac{1}{H}C_{t+1}^c\right)^\top + \frac{\eta}{N}\sum_{\tilde{c}=1}^{N}\widetilde{C}_{t+1}^{\tilde{c}}\widetilde{\xi}_t^{\tilde{c}}\widetilde{\xi}_t^{c}{}^\top\left(\mathrm{I}-\frac{1}{H}C_{t+1}^c\right)^\top - \eta\bar{\varepsilon}_{t+1}\widetilde{\xi}_t^{c}{}^\top\left(\mathrm{I}-\frac{1}{H}C_{t+1}^c\right)^\top$$

$$+\frac{1}{NH}\sum_{\tilde{c}=1}^{N}\Gamma_{t+1}\widetilde{\theta}_t\widetilde{\xi}_t^{\tilde{c}}{}^\top\widetilde{C}_{t+1}^{\tilde{c}}{}^\top + \frac{\eta}{N^2 H}\sum_{\tilde{c}=1}^{N}\sum_{\bar{c}=1}^{N}\widetilde{C}_{t+1}^{\tilde{c}}\widetilde{\xi}_t^{\tilde{c}}\widetilde{\xi}_t^{\bar{c}}{}^\top\widetilde{C}_{t+1}^{\bar{c}}{}^\top - \frac{\eta}{NH}\bar{\varepsilon}_{t+1}\sum_{\tilde{c}=1}^{N}\widetilde{\xi}_t^{\tilde{c}}{}^\top\widetilde{C}_{t+1}^{\tilde{c}}{}^\top$$

$$-\frac{1}{H}\Gamma_{t+1}\widetilde{\theta}_t\widetilde{\varepsilon}_{t+1}^{c}{}^\top - \frac{\eta}{NH}\sum_{\tilde{c}=1}^{N}\widetilde{C}_{t+1}^{\tilde{c}}\widetilde{\xi}_t^{\tilde{c}}\widetilde{\varepsilon}_{t+1}^{c}{}^\top + \frac{\eta}{H}\bar{\varepsilon}_{t+1}\widetilde{\varepsilon}_{t+1}^{c}{}^\top \ .$$

Now we proceed as above by taking the expectation, then the norm, and using the triangle inequality. Note that by (41), we have $\mathbb{E}[\bar{\varepsilon}_{t+1}\widetilde{\theta}_t^\top\widetilde{\Gamma}_{t+1}^{c}{}^\top] = 0$, $\mathbb{E}[\bar{\varepsilon}_{t+1}\widetilde{\xi}_t^{c}{}^\top\left(\mathrm{I}-\frac{1}{H}C_{t+1}^c\right)^\top] = 0$, $\mathbb{E}[\bar{\varepsilon}_{t+1}\sum_{\tilde{c}=1}^{N}\widetilde{\xi}_t^{\tilde{c}}{}^\top\widetilde{C}_{t+1}^{\tilde{c}}{}^\top] = 0$, $\mathbb{E}[\Gamma_{t+1}\widetilde{\theta}_t\widetilde{\varepsilon}_{t+1}^{c}{}^\top] = 0$, and $\mathbb{E}[\widetilde{C}_{t+1}^{\tilde{c}}\widetilde{\xi}_t^{\tilde{c}}\widetilde{\varepsilon}_{t+1}^{c}{}^\top] = 0$. After using

Jensen's inequality, we obtain

$$\|\mathbb{E}[\widetilde{\theta}_{t+1}\widetilde{\xi}_{t+1}^{c}{}^{\top}]\| \leq \frac{1}{\eta H}\mathbb{E}^{1/2}[\|\widetilde{\Gamma}_{t+1}^{c}\|^{2}]\|\mathbb{E}[\widetilde{\theta}_t\widetilde{\theta}_t^{\top}]\| + \frac{1}{NH}\sum_{\tilde{c}=1}^{N}\mathbb{E}^{1/2}[\|\widetilde{C}_{t+1}^{\tilde{c}}\|^{2}]\|\mathbb{E}[\widetilde{\xi}_t^{\tilde{c}}\widetilde{\theta}_t^{\top}]\|$$

$$+ \mathbb{E}^{1/2}\left[\left\|\mathbf{I} - \frac{1}{H}C_{t+1}^{c}\right\|^{2}\right]\|\mathbb{E}[\widetilde{\theta}_t\widetilde{\xi}_t^{c\top}]\| + \frac{\eta}{N}\sum_{\tilde{c}=1}^{N}\mathbb{E}\left[\left\|\widetilde{C}_{t+1}^{\tilde{c}}\right\|\left\|\left(\mathbf{I} - \frac{1}{H}C_{t+1}^{c}\right)\right\|\right]\|\mathbb{E}[\widetilde{\xi}_t^{\tilde{c}}\widetilde{\xi}_t^{c\top}]\|$$

$$+ \frac{1}{NH}\sum_{\tilde{c}=1}^{N}\mathbb{E}^{1/2}\left[\|\widetilde{C}_{t+1}^{\tilde{c}}\|^{2}\right]\|\mathbb{E}[\widetilde{\theta}_t\widetilde{\xi}_t^{\tilde{c}\top}]\| + \frac{\eta}{N^2H}\sum_{\tilde{c}=1}^{N}\sum_{\tilde{c}=1}^{N}\mathbb{E}\left[\|\widetilde{C}_{t+1}^{\tilde{c}}\|\|\widetilde{C}_{t+1}^{\tilde{c}}{}^{\top}\|\right]\|\mathbb{E}[\widetilde{\xi}_t^{\tilde{c}}\widetilde{\xi}_t^{\tilde{c}\top}]\|$$

$$+ \left\|\mathbb{E}\left[\frac{\eta}{H}\bar{\varepsilon}_{t+1}\widetilde{\varepsilon}_{t+1}^{c}{}^{\top}\right]\right\|.$$

Using Lemma C.1, Lemma C.2, and Lemma C.3, we obtain

$$\|\mathbb{E}[\widetilde{\theta}_{t+1}\widetilde{\xi}_{t+1}^{c}{}^{\top}]\| \leq 2\left\{C_{\mathbf{A}} + \|\Sigma_{\widetilde{\mathbf{A}}}^{c}\|^{1/2}\right\}\|\mathbb{E}[\widetilde{\theta}_t\widetilde{\theta}_t^{\top}]\| + \frac{1}{NH}\sum_{\tilde{c}=1}^{N}\eta H^2\left\{C_{\mathbf{A}} + \|\Sigma_{\widetilde{\mathbf{A}}}^{c}\|^{1/2}\right\}\|\mathbb{E}[\widetilde{\xi}_t^{\tilde{c}}\widetilde{\theta}_t^{\top}]\|$$

$$+ \frac{\eta H}{2}\left\{C_{\mathbf{A}} + \|\Sigma_{\widetilde{\mathbf{A}}}^{c}\|^{1/2}\right\}\|\mathbb{E}[\widetilde{\theta}_t\widetilde{\xi}_t^{c\top}]\| + \frac{\eta}{N}\sum_{\tilde{c}=1}^{N}\eta H^2\left\{C_{\mathbf{A}} + \|\Sigma_{\widetilde{\mathbf{A}}}^{c}\|^{1/2}\right\}\|\mathbb{E}[\widetilde{\xi}_t^{\tilde{c}}\widetilde{\xi}_t^{c\top}]\|$$

$$+ \frac{1}{NH}\sum_{\tilde{c}=1}^{N}\eta H^2\left\{C_{\mathbf{A}} + \|\Sigma_{\widetilde{\mathbf{A}}}^{c}\|^{1/2}\right\}\|\mathbb{E}[\widetilde{\theta}_t\widetilde{\xi}_t^{\tilde{c}\top}]\|$$

$$+ \frac{\eta}{N^2H}\sum_{\tilde{c}=1}^{N}\sum_{\tilde{c}=1}^{N}\eta^2 H^4\left\{C_{\mathbf{A}}^2 + \|\Sigma_{\widetilde{\mathbf{A}}}^{c}\|\right\}\|\mathbb{E}[\widetilde{\xi}_t^{\tilde{c}}\widetilde{\xi}_t^{\tilde{c}\top}]\| + \left\|\mathbb{E}\left[\frac{\eta}{H}\bar{\varepsilon}_{t+1}\widetilde{\varepsilon}_{t+1}^{c}{}^{\top}\right]\right\|,$$

where we used the two following inequalities

$$\mathbb{E}\left[\left\|\widetilde{C}_{t+1}^{\tilde{c}}\right\|\left\|\left(\mathbf{I} - \frac{1}{H}C_{t+1}^{c}\right)\right\|\right] \leq \mathbb{E}^{1/2}\left[\left\|\widetilde{C}_{t+1}^{\tilde{c}}\right\|^{2}\right] \leq \eta H^2\left\{C_{\mathbf{A}} + \|\Sigma_{\widetilde{\mathbf{A}}}^{c}\|^{1/2}\right\},$$

$$\mathbb{E}\left[\|\widetilde{C}_{t+1}^{\tilde{c}}\|\|\widetilde{C}_{t+1}^{\tilde{c}}{}^{\top}\|\right] \leq \mathbb{E}\left[\frac{1}{2}\|\widetilde{C}_{t+1}^{\tilde{c}}\|^2 + \frac{1}{2}\|\widetilde{C}_{t+1}^{\tilde{c}}{}^{\top}\|^2\right] \leq \eta^2 H^4\left\{C_{\mathbf{A}}^2 + \|\Sigma_{\widetilde{\mathbf{A}}}^{c}\|\right\}$$

This leads to the following inequality

$$\eta H b_{t+1}^{(\theta,\xi)} \leq 2\eta H\left\{C_{\mathbf{A}} + \|\Sigma_{\widetilde{\mathbf{A}}}^{c}\|^{1/2}\right\}b_t^{(\theta,\theta)} + 3\eta^2 H^2\left\{C_{\mathbf{A}} + \|\Sigma_{\widetilde{\mathbf{A}}}^{c}\|^{1/2}\right\}b_t^{(\theta,\xi)}$$

$$+ \eta^2 H^2\left(\eta H\left\{C_{\mathbf{A}} + \|\Sigma_{\widetilde{\mathbf{A}}}^{c}\|^{1/2}\right\} + \eta^2 H^2\left\{C_{\mathbf{A}}^2 + \|\Sigma_{\widetilde{\mathbf{A}}}^{c}\|\right\}\right)\left\{\frac{1}{N}b_t^{=} + \left(1 - \frac{1}{N}\right)b_t^{\neq}\right\}$$

$$+ \frac{2\eta^2 H}{N}\|\Sigma_\omega\|,$$

where we used $\left\|\mathbb{E}\left[\frac{\eta}{H}\bar{\varepsilon}_{t+1}\widetilde{\varepsilon}_{t+1}^{c}{}^{\top}\right]\right\| \leq b_1^{(\theta,\xi)} = \frac{2\eta}{N}\|\Sigma_\omega\|$. Assuming $\eta H\left\{C_{\mathbf{A}} + \|\Sigma_{\widetilde{\mathbf{A}}}^{c}\|^{1/2}\right\} \leq \nu$ and $\eta^2 H^2\left\{C_{\mathbf{A}}^2 + \|\Sigma_{\widetilde{\mathbf{A}}}^{c}\|\right\} \leq \nu$, we obtain the following bound

$$\eta H b_{t+1}^{(\theta,\xi)} \leq 2\nu b_t^{(\theta,\theta)} + 3\nu\eta H b_t^{(\theta,\xi)} + 2\nu\eta^2 H^2\left\{\frac{1}{N}b_t^{=} + \left(1 - \frac{1}{N}\right)b_t^{\neq}\right\} + \frac{2\eta^2 H}{N}\|\Sigma_\omega\|.$$

*(Value of $b_{t+1}^{=}$ and $b_{t+1}^{\neq}$.)* As above, we start by expanding the matrix product,

$$\widetilde{\xi}_{t+1}^{c}\widetilde{\xi}_{t+1}^{c}{}^{\top} = \left(\frac{1}{\eta H}\widetilde{\Gamma}_{t+1}^{c}\widetilde{\theta}_t + \left(\mathbf{I} - \frac{1}{H}C_{t+1}^{c}\right)\widetilde{\xi}_t^{c} + \frac{1}{NH}\sum_{\tilde{c}=1}^{N}\widetilde{C}_{t+1}^{\tilde{c}}\widetilde{\xi}_t^{\tilde{c}} - \frac{1}{H}\widetilde{\varepsilon}_{t+1}^{c}\right)$$

$$\left(\frac{1}{\eta H}\widetilde{\Gamma}_{t+1}^{c}\widetilde{\theta}_t + \left(\mathbf{I} - \frac{1}{H}C_{t+1}^{c}\right)\widetilde{\xi}_t^{c} + \frac{1}{NH}\sum_{\tilde{c}=1}^{N}\widetilde{C}_{t+1}^{\tilde{c}}\widetilde{\xi}_t^{\tilde{c}} - \frac{1}{H}\widetilde{\varepsilon}_{t+1}^{c}\right)^{\top}$$

$$
= \frac{1}{\eta^2 H^2} \widetilde{\Gamma}_{t+1}^c \widetilde{\theta}_t \widetilde{\theta}_t^\top \widetilde{\Gamma}_{t+1}^c{}^\top + \frac{1}{\eta H} \Big( I - \frac{1}{H} C_{t+1}^c \Big) \widetilde{\xi}_t^c \widetilde{\theta}_t^\top \widetilde{\Gamma}_{t+1}^c{}^\top
$$

$$
+ \frac{1}{\eta N H^2} \sum_{\tilde{c}=1}^N \widetilde{C}_{t+1}^{\tilde{c}} \widetilde{\xi}_t^{\tilde{c}} \widetilde{\theta}_t^\top \widetilde{\Gamma}_{t+1}^c{}^\top - \frac{1}{\eta H^2} \widetilde{\varepsilon}_{t+1}^c \widetilde{\theta}_t^\top \widetilde{\Gamma}_{t+1}^c{}^\top
$$

$$
+ \frac{1}{\eta H} \widetilde{\Gamma}_{t+1}^c \widetilde{\theta}_t \widetilde{\xi}_t^c{}^\top \Big( I - \frac{1}{H} C_{t+1}^c \Big)^\top + \Big( I - \frac{1}{H} C_{t+1}^c \Big) \widetilde{\xi}_t^c \widetilde{\xi}_t^c{}^\top \Big( I - \frac{1}{H} C_{t+1}^c \Big)^\top
$$

$$
+ \frac{1}{N H} \sum_{\tilde{c}=1}^N \widetilde{C}_{t+1}^{\tilde{c}} \widetilde{\xi}_t^{\tilde{c}} \widetilde{\xi}_t^c{}^\top \Big( I - \frac{1}{H} C_{t+1}^c \Big)^\top - \frac{1}{H} \widetilde{\varepsilon}_{t+1}^c \widetilde{\xi}_t^c{}^\top \Big( I - \frac{1}{H} C_{t+1}^c \Big)^\top
$$

$$
+ \frac{1}{\eta N H^2} \widetilde{\Gamma}_{t+1}^c \widetilde{\theta}_t \widetilde{\xi}_t^{\tilde{c}}{}^\top \sum_{\tilde{c}=1}^N \widetilde{C}_{t+1}^{\tilde{c}}{}^\top + \frac{1}{N H} \Big( I - \frac{1}{H} C_{t+1}^c \Big) \widetilde{\xi}_t^c \widetilde{\xi}_t^{\tilde{c}}{}^\top \sum_{\tilde{c}=1}^N \widetilde{C}_{t+1}^{\tilde{c}}{}^\top
$$

$$
+ \frac{1}{N^2 H^2} \sum_{\tilde{c}=1}^N \widetilde{C}_{t+1}^{\tilde{c}} \widetilde{\xi}_t^{\tilde{c}} \widetilde{\xi}_t^{\tilde{c}}{}^\top \sum_{\tilde{c}=1}^N \widetilde{C}_{t+1}^{\tilde{c}}{}^\top - \frac{1}{N H^2} \widetilde{\varepsilon}_{t+1}^c \widetilde{\xi}_t^{\tilde{c}}{}^\top \sum_{\tilde{c}=1}^N \widetilde{C}_{t+1}^{\tilde{c}}{}^\top
$$

$$
+ \frac{1}{\eta H^2} \widetilde{\Gamma}_{t+1}^c \widetilde{\theta}_t \widetilde{\varepsilon}_{t+1}^c{}^\top + \frac{1}{H} \Big( I - \frac{1}{H} C_{t+1}^c \Big) \widetilde{\xi}_t^c \widetilde{\varepsilon}_{t+1}^c{}^\top
$$

$$
+ \frac{1}{N H^2} \sum_{\tilde{c}=1}^N \widetilde{C}_{t+1}^{\tilde{c}} \widetilde{\xi}_t^{\tilde{c}} \widetilde{\varepsilon}_{t+1}^c{}^\top - \frac{1}{H^2} \widetilde{\varepsilon}_{t+1}^c \widetilde{\varepsilon}_{t+1}^c{}^\top .
$$

Taking the expectation, then the norm, and using triangle inequality and Jensen's inequality, we obtain

$$
\|\mathbb{E}[\widetilde{\xi}_{t+1}^c \widetilde{\xi}_{t+1}^c{}^\top]\|
$$

$$
\leq \frac{1}{\eta^2 H^2} \mathbb{E}[\|\widetilde{\Gamma}_{t+1}^c\|^2] \|\mathbb{E}[\widetilde{\theta}_t \widetilde{\theta}_t^\top]\| + \frac{1}{\eta H} \mathbb{E}^{1/2} \left[ \Big\| I - \frac{1}{H} C_{t+1}^c \Big\|^2 \right] \|\mathbb{E}[\widetilde{\xi}_t^c \widetilde{\theta}_t^\top]\|
$$

$$
+ \frac{1}{\eta N H^2} \sum_{\tilde{c}=1}^N \mathbb{E}^{1/2} \left[ \|\widetilde{C}_{t+1}^{\tilde{c}}\|^2 \right] \|\mathbb{E}[\widetilde{\xi}_t^{\tilde{c}} \widetilde{\theta}_t^\top]\|
$$

$$
+ \frac{1}{\eta H} \mathbb{E}^{1/2} \left[ \Big\| I - \frac{1}{H} C_{t+1}^c \Big\|^2 \right] \|\mathbb{E}[\widetilde{\theta}_t \widetilde{\xi}_t^c{}^\top]\| + \mathbb{E} \left[ \Big\| I - \frac{1}{H} C_{t+1}^c \Big\| \Big\| I - \frac{1}{H} C_{t+1}^c \Big\| \right] \|\mathbb{E}[\widetilde{\xi}_t^c \widetilde{\xi}_t^c{}^\top]\|
$$

$$
+ \frac{1}{N H} \sum_{\tilde{c}=1}^N \mathbb{E} \left[ \|\widetilde{C}_{t+1}^{\tilde{c}}\| \Big\| I - \frac{1}{H} C_{t+1}^c \Big\| \right] \|\mathbb{E}[\widetilde{\xi}_t^{\tilde{c}} \widetilde{\xi}_t^c{}^\top]\|
$$

$$
+ \frac{1}{\eta N H^2} \sum_{\tilde{c}=1}^N \mathbb{E}^{1/2} \left[ \|\widetilde{C}_{t+1}^{\tilde{c}}{}^\top\|^2 \right] \|\mathbb{E}[\widetilde{\theta}_t \widetilde{\xi}_t^{\tilde{c}}{}^\top]\|
$$

$$
+ \frac{1}{N H} \sum_{\tilde{c}=1}^N \mathbb{E} \left[ \Big\| I - \frac{1}{H} C_{t+1}^c \Big\| \|\widetilde{C}_{t+1}^{\tilde{c}}{}^\top\| \right] \|\mathbb{E}[\widetilde{\xi}_t^c \widetilde{\xi}_t^{\tilde{c}}{}^\top]\|
$$

$$
+ \frac{1}{N^2 H^2} \sum_{\tilde{c}=1}^N \sum_{\tilde{c}=1}^N \mathbb{E} \left[ \|\widetilde{C}_{t+1}^{\tilde{c}}\| \|\widetilde{C}_{t+1}^{\tilde{c}}{}^\top\| \right] \|\mathbb{E}[\widetilde{\xi}_t^{\tilde{c}} \widetilde{\xi}_t^{\tilde{c}}{}^\top]\| + \|\mathbb{E}[\frac{1}{H^2} \widetilde{\varepsilon}_{t+1}^c \widetilde{\varepsilon}_{t+1}^c{}^\top]\| .
$$

We can now use Lemma C.1, Lemma C.2, and Lemma C.3 to obtain the following upper bound

$$
\|\mathbb{E}[\widetilde{\xi}_{t+1}^c \widetilde{\xi}_{t+1}^c{}^\top]\|
$$

$$
\leq 2 \left\{ C_{\mathbf{A}}^2 + \|\Sigma_{\widetilde{\mathbf{A}}}^c\| \right\} \|\mathbb{E}[\widetilde{\theta}_t \widetilde{\theta}_t^\top]\| + \frac{1}{\eta H} \frac{\eta H}{2} \left\{ C_{\mathbf{A}} + \|\Sigma_{\widetilde{\mathbf{A}}}^c\|^{1/2} \right\} \|\mathbb{E}[\widetilde{\xi}_t^c \widetilde{\theta}_t^\top]\|
$$

$$
+ \frac{1}{\eta N H^2} \sum_{\tilde{c}=1}^N \eta H^2 \left\{ C_{\mathbf{A}} + \|\Sigma_{\widetilde{\mathbf{A}}}^c\|^{1/2} \right\} \|\mathbb{E}[\widetilde{\xi}_t^{\tilde{c}} \widetilde{\theta}_t^\top]\|
$$

$$
+ \frac{1}{\eta H} \frac{\eta H}{2} \left\{ C_{\mathbf{A}} + \|\Sigma_{\widetilde{\mathbf{A}}}^c\|^{1/2} \right\} \|\mathbb{E}[\widetilde{\theta}_t \widetilde{\xi}_t^c{}^\top]\| + \frac{\eta^2 H^2}{4} \left\{ C_{\mathbf{A}}^2 + \|\Sigma_{\widetilde{\mathbf{A}}}^c\| \right\} \|\mathbb{E}[\widetilde{\xi}_t^c \widetilde{\xi}_t^c{}^\top]\|
$$

$$+ \frac{1}{NH} \sum_{\tilde{c}=1}^{N} \eta H^2 \left\{ \mathrm{C_A} + \|\Sigma_{\widetilde{\mathbf{A}}}^c\|^{1/2} \right\} \|\mathbb{E}[\widetilde{\xi}_t^{\tilde{c}} \widetilde{\xi}_t^{\tilde{c}\,\top}]\|$$

$$+ \frac{1}{\eta NH^2} \sum_{\tilde{c}=1}^{N} \eta H^2 \left\{ \mathrm{C_A} + \|\Sigma_{\widetilde{\mathbf{A}}}^c\|^{1/2} \right\} \|\mathbb{E}[\widetilde{\theta}_t \widetilde{\xi}_t^{\tilde{c}\,\top}]\|$$

$$+ \frac{1}{NH} \sum_{\tilde{c}=1}^{N} \eta H^2 \left\{ \mathrm{C_A} + \|\Sigma_{\widetilde{\mathbf{A}}}^c\|^{1/2} \right\} \|\mathbb{E}[\widetilde{\xi}_t^{\tilde{c}} \widetilde{\xi}_t^{\tilde{c}\,\top}]\|$$

$$+ \frac{1}{N^2 H^2} \sum_{\tilde{c}=1}^{N} \sum_{\tilde{c}=1}^{N} \eta^2 H^4 \left\{ \mathrm{C_A^2} + \|\Sigma_{\widetilde{\mathbf{A}}}^c\| \right\} \|\mathbb{E}[\widetilde{\xi}_t^{\tilde{c}} \widetilde{\xi}_t^{\tilde{c}\,\top}]\| + \left\| \frac{1}{H^2} \mathbb{E}[\widetilde{\varepsilon}_{t+1}^c \widetilde{\varepsilon}_{t+1}^{c\,\top}] \right\| .$$

This bound can be simplified as

$$\eta^2 H^2 \|\mathbb{E}[\widetilde{\xi}_{t+1}^c \widetilde{\xi}_{t+1}^{c\,\top}]\|$$

$$\leq 2\eta^2 H^2 \left\{ \mathrm{C_A^2} + \|\Sigma_{\widetilde{\mathbf{A}}}^c\| \right\} b_t^{(\theta,\theta)} + \frac{\eta^2 H^2}{2} \left\{ \mathrm{C_A} + \|\Sigma_{\widetilde{\mathbf{A}}}^c\|^{1/2} \right\} b_t^{(\theta,\xi)}$$

$$+ \eta^2 H^2 \left\{ \mathrm{C_A} + \|\Sigma_{\widetilde{\mathbf{A}}}^c\|^{1/2} \right\} b_t^{(\theta,\xi)}$$

$$+ \frac{\eta^2 H^2}{2} \left\{ \mathrm{C_A} + \|\Sigma_{\widetilde{\mathbf{A}}}^c\|^{1/2} \right\} b_t^{(\theta,\xi)} + \frac{\eta^4 H^4}{4} \left\{ \mathrm{C_A^2} + \|\Sigma_{\widetilde{\mathbf{A}}}^c\| \right\} \|\mathbb{E}[\widetilde{\xi}_t^{\tilde{c}} \widetilde{\xi}_t^{\tilde{c}\,\top}]\|$$

$$+ \eta^3 H^3 \left\{ \mathrm{C_A} + \|\Sigma_{\widetilde{\mathbf{A}}}^c\|^{1/2} \right\} \left\{ \frac{1}{N} b_t^= + \left(1 - \frac{1}{N}\right) b_t^{\neq} \right\}$$

$$+ \eta^2 H^2 \left\{ \mathrm{C_A} + \|\Sigma_{\widetilde{\mathbf{A}}}^c\|^{1/2} \right\} b_t^{(\theta,\xi)} + \eta^3 H^3 \left\{ \mathrm{C_A} + \|\Sigma_{\widetilde{\mathbf{A}}}^c\|^{1/2} \right\} \left\{ \frac{1}{N} b_t^= + \left(1 - \frac{1}{N}\right) b_t^{\neq} \right\}$$

$$+ \eta^4 H^4 \left\{ \mathrm{C_A^2} + \|\Sigma_{\widetilde{\mathbf{A}}}^c\| \right\} \left\{ \frac{1}{N} b_t^= + \left(1 - \frac{1}{N}\right) b_t^{\neq} \right\} + \left\| \frac{1}{H^2} \mathbb{E}[\widetilde{\varepsilon}_{t+1}^c \widetilde{\varepsilon}_{t+1}^{c\,\top}] \right\| ,$$

which can be simplified as

$$\eta^2 H^2 \|\mathbb{E}[\widetilde{\xi}_{t+1}^c \widetilde{\xi}_{t+1}^{c\,\top}]\| \leq 2\eta^2 H^2 \left\{ \mathrm{C_A^2} + \|\Sigma_{\widetilde{\mathbf{A}}}^c\| \right\} b_t^{(\theta,\theta)} + 3\eta H \left\{ \mathrm{C_A} + \|\Sigma_{\widetilde{\mathbf{A}}}^c\|^{1/2} \right\} \eta H b_t^{(\theta,\xi)}$$

$$+ \left( 2\eta H \left\{ \mathrm{C_A} + \|\Sigma_{\widetilde{\mathbf{A}}}^c\|^{1/2} \right\} + \eta^2 H^2 \left\{ \mathrm{C_A^2} + \|\Sigma_{\widetilde{\mathbf{A}}}^c\| \right\} \right) \eta^2 H^2 \left\{ \frac{1}{N} b_t^= + \left(1 - \frac{1}{N}\right) b_t^{\neq} \right\}$$

$$+ \frac{\eta^4 H^4}{4} \left\{ \mathrm{C_A^2} + \|\Sigma_{\widetilde{\mathbf{A}}}^c\| \right\} \|\mathbb{E}[\widetilde{\xi}_t^{\tilde{c}} \widetilde{\xi}_t^{\tilde{c}\,\top}]\| + \left\| \frac{1}{H^2} \mathbb{E}[\widetilde{\varepsilon}_{t+1}^c \widetilde{\varepsilon}_{t+1}^{c\,\top}] \right\| .$$

We now distinguish two cases, when $c = c'$ and when $c \neq c'$. First, let $c = c'$, we obtain

$$\eta^2 H^2 b_{t+1}^= \leq 2\eta^2 H^2 \left\{ \mathrm{C_A^2} + \|\Sigma_{\widetilde{\mathbf{A}}}^c\| \right\} b_t^{(\theta,\theta)} + 3\eta H \left\{ \mathrm{C_A} + \|\Sigma_{\widetilde{\mathbf{A}}}^c\|^{1/2} \right\} \eta H b_t^{(\theta,\xi)}$$

$$+ \left( 2\eta H \left\{ \mathrm{C_A} + \|\Sigma_{\widetilde{\mathbf{A}}}^c\|^{1/2} \right\} + \eta^2 H^2 \left\{ \mathrm{C_A^2} + \|\Sigma_{\widetilde{\mathbf{A}}}^c\| \right\} \right) \eta^2 H^2 \left\{ \frac{1}{N} b_t^= + \left(1 - \frac{1}{N}\right) b_t^{\neq} \right\}$$

$$+ \frac{\eta^2 H^2}{4} \left\{ \mathrm{C_A^2} + \|\Sigma_{\widetilde{\mathbf{A}}}^c\| \right\} \eta^2 H^2 b_t^= + \eta^2 H \|\Sigma_\omega\| ,$$

since when $c = c'$, we have à$\|\mathbb{E}[\widetilde{\xi}_t^c \widetilde{\xi}_t^{c\,\top}]\| \leq b_t^=$ and $\left\| \frac{1}{H} \mathbb{E}[\widetilde{\varepsilon}_{t+1}^c \widetilde{\varepsilon}_{t+1}^{c\,\top}] \right\| \leq b_1^= = \frac{N-1}{NH} \|\Sigma_\omega\|$. Assuming $\eta H \left\{ \mathrm{C_A} + \|\Sigma_{\widetilde{\mathbf{A}}}^c\|^{1/2} \right\} \leq \nu$ and $\eta^2 H^2 \left\{ \mathrm{C_A^2} + \|\Sigma_{\widetilde{\mathbf{A}}}^c\| \right\} \leq \nu$, we obtain

$$\eta^2 H^2 b_{t+1}^= \leq 2\nu b_t^{(\theta,\theta)} + 3\nu \eta H b_t^{(\theta,\xi)} + 4\nu \eta^2 H^2 b_t^= + 3\nu \eta^2 H^2 b_t^{\neq} + \eta^2 H \|\Sigma_\omega\| .$$

We proceed similarly for $c \neq c'$, which gives

$$\eta^2 H^2 b_{t+1}^{\neq} \leq 2\nu b_t^{(\theta,\theta)} + 3\nu \eta H b_t^{(\theta,\xi)} + \frac{3\nu}{N} \eta^2 H^2 b_t^= + 4\nu \eta^2 H^2 b_t^{\neq} + \frac{3\eta^2 H}{N} \|\Sigma_\omega\| ,$$

since, when $c \neq c'$, we have $\|\mathbb{E}[\widetilde{\xi}_t^c \widetilde{\xi}_t^{c\,\top}]\| \leq b_t^{\neq}$ and $\left\| \frac{1}{H} \mathbb{E}[\widetilde{\varepsilon}_{t+1}^c \widetilde{\varepsilon}_{t+1}^{c\,\top}] \right\| \leq b_1^{\neq} = \frac{3}{NH} \|\Sigma_\omega\|$.  $\square$

**Corollary C.7.** *Assume that* $\eta H \left\{ C_{\mathbf{A}} + \|\Sigma_{\widetilde{\mathbf{A}}}^{c}\|^{1/2} \right\} \leq \frac{a}{240(C + \|\Sigma\|)}$ *and* $\eta^2 H^2 \left\{ C_{\mathbf{A}}^2 + \|\Sigma_{\widetilde{\mathbf{A}}}^{c}\| \right\} \leq \frac{a}{240(C + \|\Sigma\|)}$, *set* $\omega = \min\left(1, \frac{a}{12(C + \|\Sigma\|)}\right)$, *then it holds that*

$$b_{t+1}^{(\theta,\theta)} + \omega\eta H b_{t+1}^{(\theta,\xi)} + \frac{\omega\eta^2 H^2}{N} b_{t+1}^{=} + \omega\eta^2 H^2 b_{t+1}^{\neq}$$
$$\leq \left(1 - \frac{\eta a H}{2}\right) b_t^{(\theta,\theta)} + \frac{1}{2}\omega\eta H b_t^{(\theta,\xi)} + \frac{1}{2}\frac{\omega\eta^2 H^2}{N} b_t^{=} + \frac{1}{2}\eta^2 H^2 b_t^{\neq} + \frac{9\eta^2 H}{N}\|\Sigma_\omega\| \ .$$

*Assuming* $\eta a H \leq \frac{1}{2}$, *we have* $1 - \frac{1}{2} \leq 1 - \frac{\eta a H}{2}$. *This in turn ensures that*

$$b_{t+1}^{(\theta,\theta)} + \omega\eta H b_{t+1}^{(\theta,\xi)} + \frac{\omega\eta^2 H^2}{N} b_{t+1}^{=} + \omega\eta^2 H^2 b_{t+1}^{\neq}$$
$$\leq \left(1 - \frac{\eta a H}{2}\right) \left\{ b_t^{(\theta,\theta)} + \omega\eta H b_t^{(\theta,\xi)} + \frac{\omega\eta^2 H^2}{N} b_t^{=} + \eta^2 H^2 b_t^{\neq} \right\} + \frac{9\eta^2 H}{N}\|\Sigma_\omega\| \ ,$$

*which gives, for any* $t \geq 0$,

$$b_t^{(\theta,\theta)} \leq \frac{18\eta}{Na}\|\Sigma_\omega\| \ .$$

*Proof.* From Lemma C.6, we have, for any $0 < \omega < 1$, $\nu > 0$, and assuming that $\eta H \left\{ C_{\mathbf{A}} + \|\Sigma_{\widetilde{\mathbf{A}}}^{c}\|^{1/2} \right\} \leq \nu$ and $\eta^2 H^2 \left\{ C_{\mathbf{A}}^2 + \|\Sigma_{\widetilde{\mathbf{A}}}^{c}\| \right\} \leq \nu$, and since $\omega \leq 1$,

$$b_{t+1}^{(\theta,\theta)} + \omega\eta H b_{t+1}^{(\theta,\xi)} + \frac{\omega\eta^2 H^2}{N} b_{t+1}^{=} + \omega\eta^2 H^2 b_{t+1}^{\neq}$$
$$\leq \left\{ (1 - \eta a)^{2H} + 6\omega\eta H \left\{ C_{\mathbf{A}} + \|\Sigma_{\widetilde{\mathbf{A}}}^{c}\|^{1/2} \right\} \right\} b_t^{(\theta,\theta)}$$
$$+ 10\nu\eta H b_t^{(\theta,\xi)} + 10\nu\frac{\eta^2 H^2}{N} b_t^{=} + 10\nu\eta^2 H^2 b_t^{\neq} + \frac{9\eta^2 H}{N}\|\Sigma_\omega\| \ .$$

Now, we choose $\omega = \min\left(1, \frac{a}{12(C + \|\Sigma\|)}\right)$ and obtain

$$(1 - \eta a)^{2H} + 6\omega\eta H \left\{ C_{\mathbf{A}} + \|\Sigma_{\widetilde{\mathbf{A}}}^{c}\|^{1/2} \right\} \leq 1 - \eta a H + 6\omega\eta H \left\{ C_{\mathbf{A}} + \|\Sigma_{\widetilde{\mathbf{A}}}^{c}\|^{1/2} \right\} \leq 1 - \frac{\eta a H}{2} \ .$$

Additionally, $\omega \leq 1$, thus $3 + 6\omega \leq 9$ and we obtain

$$b_{t+1}^{(\theta,\theta)} + \omega\eta H b_{t+1}^{(\theta,\xi)} + \frac{\omega\eta^2 H^2}{N} b_{t+1}^{=} + \omega\eta^2 H^2 b_{t+1}^{\neq}$$
$$\leq \left(1 - \frac{\eta a H}{2}\right) b_t^{(\theta,\theta)} + 10\nu\eta H b_t^{(\theta,\xi)} + 10\nu\frac{\eta^2 H^2}{N} b_t^{=} + 10\nu\eta^2 H^2 b_t^{\neq} + \frac{9\eta^2 H}{N}\|\Sigma_\omega\| \ .$$

Choosing $\nu \leq \frac{\omega}{20} \leq \frac{a}{240(C + \|\Sigma\|)}$ gives the result. $\qquad\square$

**Complete analysis of SCAFFLSA.** We can now state our main theorem, which gives an upper bound on the expected distance between the iterates of SCAFFLSA and the solution $\theta_\star$.

**Theorem C.8.** *Assume A1 and A3. Let* $\eta, H$ *such that* $\eta a H \leq 1$, *and* $H \leq \frac{a}{240\eta\{C + \|\Sigma\|\}}$, *and set* $\xi_0^c = 0$ *for all* $c \in [N]$. *Then, the sequence* $(\psi_t)_{t \in \mathbb{N}}$ *satisfies, for all* $t \geq 0$,

$$\mathbb{E}[\|\theta_t - \theta_\star\|^2] \leq \left(1 - \frac{\eta a H}{2}\right)^t \left\{ 2\|\theta_0 - \theta_\star\|^2 + 2\eta^2 H^2 \mathbb{E}_c[\|\bar{\mathbf{A}}^c(\theta_\star^c - \theta_\star)\|^2] \right\} + \frac{36 d\eta}{Na}\|\Sigma_\omega\| \ .$$

*Proof.* Recall our decomposition $\theta_t - \theta_\star = \check{\theta}_t - \theta_\star + \widetilde{\theta}$. By Young's inequality, we have

$$\mathbb{E}[\|\theta_t - \theta_\star\|^2] \leq 2\mathbb{E}[\|\check{\theta}_t - \theta_\star\|^2] + 2\mathbb{E}[\|\widetilde{\theta}_t\|^2] \ .$$

By Theorem C.4, we have $\mathbb{E}[\|\check{\theta}_t - \theta_\star\|^2] \leq \left(1 - \frac{\eta a H}{2}\right)^t \psi_0$, and by Corollary C.7, we have
$\mathbb{E}[\|\widetilde{\theta}_t\|^2] \leq d b_t^{(\theta,\theta)} \leq \frac{18\eta d}{Na}\|\Sigma_\omega\|$. Combine the two results, we obtain

$$\mathbb{E}[\|\theta_t - \theta_\star\|^2] \leq \left(1 - \frac{\eta a H}{2}\right)^t 2\psi_0 + \frac{36 d\eta}{Na}\|\Sigma_\omega\| \ ,$$

replacing $\psi_0 = \|\theta_0 - \theta_\star\|^2 + \frac{\eta H}{N}\sum_{c=1}^{N}\|\bar{\mathbf{A}}^c(\theta_\star^c - \theta_\star)\|^2$ gives the result of the theorem. $\qquad\square$

**Corollary C.9.** *Under the Assumptions of Theorem C.8, one may set the parameter of* SCAFFLSA *to*

$$\eta = \min\left(\eta_\infty, \frac{Na\epsilon^2}{72 d\|\Sigma_\varepsilon\|}\right) \ , \quad H = \frac{1}{240\left\{\mathrm{C}_{\mathbf{A}}^2 + \|\Sigma_{\widetilde{\mathbf{A}}}\|\right\}}\max\left(\frac{a}{\eta_\infty}, \frac{72 d\|\Sigma_\omega\|}{N\epsilon^2}\right) \ ,$$

*which guarantees $\mathbb{E}[\|\theta_t - \theta_\star\|^2] \leq \epsilon^2$ after a number of communication rounds*

$$T \geq \frac{240\left\{\mathrm{C}_{\mathbf{A}}^2 + \|\Sigma_{\widetilde{\mathbf{A}}}\|\right\}}{a^2}\log\left(\frac{4\|\theta_0 - \theta_\star\|^2 + \frac{4\eta H}{N}\sum_{c=1}^{N}\|\bar{\mathbf{A}}^c(\theta_\star^c - \theta_\star)\|^2}{\epsilon^2}\right) \ .$$

*The overall sample complexity of the algorithm is then*

$$TH = \max\left(\frac{240}{\eta_\infty a}, \frac{72 d\|\Sigma_\varepsilon\|}{Na^2\epsilon^2}\right)\log\left(\frac{4\|\theta_0 - \theta_\star\|^2 + \frac{4\eta H}{N}\sum_{c=1}^{N}\|\bar{\mathbf{A}}^c(\theta_\star^c - \theta_\star)\|^2}{\epsilon^2}\right) \ .$$

*Proof.* Let $\epsilon > 0$. Starting from Theorem C.8's upper bound, we have $\mathbb{E}[\|\theta_t - \theta_\star\|^2] \leq \epsilon^2$ whenever

$$\left(1 - \frac{\eta a H}{2}\right)^t 2\psi_0 + \frac{36 d\eta}{Na}\|\Sigma_\omega\| \leq \epsilon^2 \ ,$$

where $\psi_0 = \|\theta_0 - \theta_\star\|^2 + \frac{\eta H}{N}\sum_{c=1}^{N}\|\bar{\mathbf{A}}^c(\theta_\star^c - \theta_\star)\|^2$. This gives a first condition $\frac{36 d\eta}{Na}\|\Sigma_\omega\| \leq \epsilon^2$, which requires

$$\eta \leq \frac{Na\epsilon^2}{72 d\|\Sigma_\varepsilon\|} \ .$$

This allows to take any value of $H$ such that $H \leq \frac{a}{240\eta\{\mathrm{C}+\|\Sigma\|\}} = \frac{72}{240 N\epsilon\{\mathrm{C}+\|\Sigma\|\}}$. With such setting, it remains to set the number of communication $T$ to

$$T \geq \frac{1}{\eta a H}\log\left(\frac{2\psi_0}{\epsilon^2}\right) = \frac{240\left\{\mathrm{C}_{\mathbf{A}}^2 + \|\Sigma_\varepsilon^c\|\right\}}{a^2}\log\left(\frac{4\psi_0}{\epsilon^2}\right) \ ,$$

which ensures that $\left(1 - \frac{\eta a H}{2}\right)^t 2\psi_0 \leq \frac{\epsilon}{2}$. $\qquad\square$

# D   Technical proofs

**Lemma D.1.** *For any matrix-valued sequences $(U_n)_{n\in\mathbb{N}}$, $(V_n)_{n\in\mathbb{N}}$ and for any $M \in \mathbb{N}$, it holds that:*

$$\prod_{k=1}^{M} U_k - \prod_{k=1}^{M} V_k = \sum_{k=1}^{M}\{\prod_{j=1}^{k-1} U_j\}(U_k - V_k)\{\prod_{j=k+1}^{M} V_j\} \ .$$

**Lemma D.2** (Stability of the deterministic product). *Assume A3. Then, for any $u \in \mathbb{R}^d$ and $h \in \mathbb{N}$,*

$$\|(\mathrm{I} - \eta\bar{\mathbf{A}}^c)^h u\| \leq (1 - \eta a)^h\|u\| \ .$$

*Proof.* Since $(Z_{t,h}^c)_{1\leq h\leq H}$ are i.i.d, we get

$$\mathbb{E}\big[\Gamma_{t,1:h}^{(c,\eta)} u\big] = \mathbb{E}\big[\prod_{l=1}^{h}(\mathrm{I} - \eta\mathbf{A}(Z_{t,l}^c))u\big] = \prod_{l=1}^{h}\mathbb{E}\big[\mathrm{I} - \eta\mathbf{A}(Z_{t,l}^c)\big]u = (\mathrm{I} - \eta\bar{\mathbf{A}}^c)^h u \ .$$

The proof then follows from the elementary inequality: for any square-integrable random vector $U$, $\|\mathbb{E}[U]\| \leq (\mathbb{E}[\|U\|^2])^{1/2}$. $\qquad\square$

**Lemma D.3.** *Let $(x_i)_{i=1}^N$, and $(y_i)_{i=1}^N$ be $N$ vectors of $\mathbb{R}^d$. Denote $\bar{x}_N = (1/N)\sum_{i=1}^N x_i$ and $\bar{y}_N = (1/N)\sum_{i=1}^N y_i$. Then,*

$$N\|\bar{x}_N - \bar{y}_N\|^2 = \sum_{i=1}^N \|x_i - y_i\|^2 - \sum_{i=1}^N \|x_i - \bar{x}_N - (y_i - \bar{y}_N)\|^2$$

*Proof.* Define $\mathrm{x} = [x_1^\top, \ldots, x_N^\top]^\top$ and $\mathrm{y} = [y_1^\top, \ldots, y_N^\top]^\top \in \mathbb{R}^{Nd}$. Define by P the orthogonal projector on

$$\mathcal{E} = \left\{ \mathrm{x} \in \mathbb{R}^{Nd} : \mathrm{x} = [x^\top, \ldots, x^\top]^\top, x \in \mathbb{R}^d \right\} .$$

We show that $\mathrm{P}\,\mathrm{x} = [\bar{x}_N^\top, \ldots, \bar{x}_N^\top]^\top$. Note indeed that for any $\mathrm{z} = [z^\top, \ldots, z^\top]^\top \in \mathcal{E}$, we get (with a slight abuse of notations, $\langle \cdot, \cdot \rangle$ denotes the scalar product in $\mathbb{R}^{Nd}$ and $\mathbb{R}^d$)

$$\langle \mathrm{x} - \mathrm{P}\,\mathrm{x}, \mathrm{z} \rangle = \sum_{i=1}^N \{\langle x_i, z \rangle - \langle \bar{x}_N, z \rangle\} = 0 .$$

The proof follows from Pythagoras identity which shows that

$$\| \mathrm{P}\,\mathrm{x} - \mathrm{P}\,\mathrm{y} \|^2 = \| \mathrm{x} - \mathrm{y} \|^2 - \|(\mathrm{x} - \mathrm{P}\,\mathrm{x}) - (\mathrm{y} - \mathrm{P}\,\mathrm{y} \|^2)$$

. $\qquad\square$

**Lemma D.4.** *Assume A4. Let $Z$ be a random variable taking values in a state space $(\mathsf{Z}, \mathcal{Z})$ with distribution $\pi_c$. Set $\eta \geq 0$, then for any vector $u \in \mathbb{R}^d$, we have*

$$\mathbb{E}[\|(\mathrm{I} - \eta\mathbf{A}^c(Z))u\|^2] \leq (1 - \eta a)\|u\|^2 - \eta(\tfrac{1}{L} - \eta)\mathbb{E}[\|\mathbf{A}^c(Z)u\|^2] .$$

*Proof.* First, remark that

$$\begin{aligned}
\|(\mathrm{I} - \eta\mathbf{A}^c(Z))u\|^2 &= u^\top (\mathrm{I} - \eta\mathbf{A}^c(Z))^\top (\mathrm{I} - \eta\mathbf{A}^c(Z))u \\
&= u^\top \left(\mathrm{I} - 2\eta(\tfrac{1}{2}(\mathbf{A}^c(Z) + \mathbf{A}^c(Z)^\top)) + \eta^2 \mathbf{A}^c(Z)^\top \mathbf{A}^c(Z)\right)u .
\end{aligned}$$

Since we have $\mathbb{E}[\tfrac{1}{2}(\mathbf{A}^c(Z) + \mathbf{A}^c(Z)^\top)] \succcurlyeq a\mathrm{I}$ and $\mathbb{E}[\tfrac{1}{2}(\mathbf{A}^c(Z) + \mathbf{A}^c(Z)^\top)] \succcurlyeq \tfrac{1}{L}\mathbb{E}[\mathbf{A}^c(Z)^\top \mathbf{A}^c(Z)]$, we obtain

$$\begin{aligned}
\mathbb{E}[\|(\mathrm{I} - \eta\mathbf{A}^c(Z))u\|^2] &= u^\top u - 2\eta u^\top \mathbb{E}[\tfrac{1}{2}(\mathbf{A}^c(Z) + \mathbf{A}^c(Z)^\top)]u + \eta^2 u^\top \mathbb{E}[\mathbf{A}^c(Z)^\top \mathbf{A}^c(Z)]u \\
&\leq \|u\|^2 - \eta a\|u\|^2 - \tfrac{\eta}{L}u^\top \mathbb{E}[\mathbf{A}^c(Z)^\top \mathbf{A}^c(Z)]u + \eta^2 u^\top \mathbb{E}[\mathbf{A}^c(Z)^\top \mathbf{A}^c(Z)]u \\
&= (1 - \eta a)\|u\|^2 - \eta(\tfrac{1}{L} - \eta)u^\top \mathbb{E}[\mathbf{A}^c(Z)^\top \mathbf{A}^c(Z)]u ,
\end{aligned}$$

which gives the result. $\qquad\square$

## E  TD learning as a federated LSA problem

In this section we specify TD(0) as a particular instance of the LSA algorithm. In the setting of linear functional approximation the problem of estimating $V^\pi(s)$ reduces to the problem of estimating $\theta_\star \in \mathbb{R}^d$, which can be done via the LSA procedure. For the agent $c \in [N]$ the $k$-th step randomness is given by the tuple $Z_k^c = (S_k^c, A_k^c, S_{k+1}^c)$. With slight abuse of notation, we write $\mathbf{A}_{t,h}^c$ instead of $\mathbf{A}(Z_{t,h}^c)$, and $\mathbf{b}_{t,h}^c$ instead of $\mathbf{b}(Z_{t,h}^c)$. Then the corresponding LSA update equation with constant step size $\eta$ can be written as

$$\theta_{t,h}^c = \theta_{t,h-1}^c - \eta(\mathbf{A}_{t,h}^c \theta_{t,h-1}^c - \mathbf{b}_{t,h}^c) ,$$

where $\mathbf{A}_{t,h}^c$ and $\mathbf{b}_{t,h}^c$ are given by

$$\begin{aligned}
\mathbf{A}_{t,h}^c &= \phi(S_{t,h}^c)\{\phi(S_{t,h}^c) - \gamma\phi(S_{t,h+1}^c)\}^\top , \\
\mathbf{b}_{t,h}^c &= \phi(S_{t,h}^c)r^c(S_{t,h}^c, A_{t,h}^c) .
\end{aligned} \tag{42}$$

Respective specialisation of FedLSA and SCAFFLSA algorithms to TD learning are stated in Algorithm 4 and Algorithm 5.

---

**Algorithm 4** Federated TD(0): FedLSA applied to TD(0) with linear functional approximation

---

**Input:** $\eta > 0$, $\theta_0 \in \mathbb{R}^d$, $T, N, H > 0$
**for** $t = 0$ to $T-1$ **do**
  Initialize $\theta_{t,0} = \theta_t$
  **for** $c = 1$ to $N$ **do**
    **for** $h = 1$ to $H$ **do**
      Receive tuple $(S_{t,h}^c, A_{t,h}^c, S_{t,h+1}^c)$ following **TD** 1 and perform local update:

$$\theta_{t,h}^c = \theta_{t,h-1}^c - \eta(\mathbf{A}_{t,h}^c \theta_{t,h-1}^c - \mathbf{b}_{t,h}^c) \,,$$

  where $\mathbf{A}_{t,h}^c$ and $\mathbf{b}_{t,h}^c$ are given in (42)

  Average: $\theta_{t+1} = \frac{1}{N} \sum_{c=1}^N \theta_{t,H}^c$ $\hspace{4cm}$ (43)

---

**Algorithm 5** SCAFFTD(0): SCAFFLSA applied to TD(0) with linear functional approximation

---

**Input:** $\eta > 0$, $\theta_0 \in \mathbb{R}^d$, $T, N, H > 0$
**for** $t = 0$ to $T-1$ **do**
  Initialize $\theta_{t,0} = \theta_t$
  **for** $c = 1$ to $N$ **do**
    **for** $h = 1$ to $H$ **do**
      Receive tuple $(S_{t,h}^c, A_{t,h}^c, S_{t,h+1}^c)$ following **TD** 1 and perform local update:

$$\theta_{t,h}^c = \theta_{t,h-1}^c - \eta(\mathbf{A}_{t,h}^c \theta_{t,h-1}^c - \mathbf{b}_{t,h}^c - \xi^c) \,,$$

  where $\mathbf{A}_{t,h}^c$ and $\mathbf{b}_{t,h}^c$ are given in (42)
  Average: $\theta_{t+1} = \frac{1}{N} \sum_{c=1}^N \theta_{t,H}^c$
  Update local control variates: $\xi_{t+1}^c = \xi_t^c + \frac{1}{\eta H}(\theta_{t+1} - \hat{\theta}_{t,H}^c)$.

---

The corresponding local agent's system writes as $\bar{\mathbf{A}}^c \theta_\star^c = \bar{\mathbf{b}}^c$, where we have, respectively,

$$\bar{\mathbf{A}}^c = \mathbb{E}_{s\sim\mu, s\sim P(\cdot|s)}[\phi(s)\{\phi(s) - \gamma\phi(s')\}^\top]$$
$$\bar{\mathbf{b}}^c = \mathbb{E}_{s\sim\mu, a\sim\pi(\cdot|s)}[\phi(s)r^c(s,a)] \,.$$

The authors of [50] study the corresponding virtual MDP dynamics with $\tilde{\mathbb{P}} = N^{-1}\sum_{c=1}^N \mathbb{P}_{\mathrm{MDP}}^c$, $\tilde{r} = N^{-1}\sum_{c=1}^N r^c$. Next, introducing the invariant distribution of the kernel $\tilde{\mu}$ of the averaged state kernel

$$\tilde{\mathbb{P}}_\pi(B|s) = N^{-1}\sum_{c=1}^N \int_{\mathcal{A}} \mathbb{P}_{\mathrm{MDP}}^c(B|s,a)\pi(da|s) \,,$$

we have $\tilde{\theta}$ as an optimal parameter corresponding to the system $\tilde{A}\tilde{\theta} = \tilde{b}$. Here

$$\tilde{A} = \mathbb{E}_{s\sim\tilde{\mu}, s\sim\tilde{\mathbb{P}}(\cdot|s)}[\phi(s)\{\phi(s) - \gamma\phi(s')\}^\top]$$
$$\tilde{b} = \mathbb{E}_{s\sim\tilde{\mu}, a\sim\pi(\cdot|s)}[\phi(s)\tilde{r}(s,a)] \,.$$

### E.1 Proof of Claim 3.1.

We prove the following inequalities

$$C_{\mathbf{A}} = 1 + \gamma \,, \hspace{4cm} (44)$$
$$\|\Sigma_{\tilde{\mathbf{A}}}^c\| \leq 2(1+\gamma)^2 \,, \hspace{4cm} (45)$$
$$\mathrm{Tr}(\Sigma_\varepsilon^c) \leq 2(1+\gamma)^2 \left(\|\theta_\star^c\|^2 + 1\right) \,, \hspace{2cm} (46)$$
$$a = \frac{(1-\gamma)\nu}{2} \,, \hspace{4cm} (47)$$
$$\eta_\infty = \frac{(1-\gamma)}{4} \,. \hspace{4cm} (48)$$

Table 2: Communication and sample complexity for finding a solution with MSE lower than $\epsilon^2$ for FedLSA, Scaffnew, and SCAFFLSA on the federated TD learning problem. Our analysis is the first to show that FedLSA exhibits linear speed-up, as well as its variant that reduces bias using control variates.

| Algorithm | Communication $T$ | Local updates $H$ | Sample complexity $TH$ |
|---|---|---|---|
| FedTD [12] | $\mathcal{O}\left(\frac{N}{(1-\gamma)\nu\epsilon}\log\frac{1}{\epsilon}\right)$ | $1$ | $\mathcal{O}\left(\frac{N}{(1-\gamma)\nu\epsilon}\log\frac{1}{\epsilon}\right)$ |
| FedTD (Cor. 4.4) | $\mathcal{O}\left(\frac{1}{(1-\gamma)\nu\epsilon}\log\frac{1}{\epsilon}\right)$ | $\mathcal{O}\left(\frac{1}{N\epsilon}\right)$ | $\mathcal{O}\left(\frac{1}{N(1-\gamma)\nu\epsilon}\log\frac{1}{\epsilon}\right)$ |
| SCAFFTD (Cor. 5.3) | $\mathcal{O}\left(\frac{1}{(1-\gamma)\nu}\log\frac{1}{\epsilon}\right)$ | $\mathcal{O}\left(\frac{1}{N\epsilon}\right)$ | $\mathcal{O}\left(\frac{1}{N(1-\gamma)\nu\epsilon}\log\frac{1}{\epsilon}\right)$ |

The proof below closely follows [42] (Lemma 7) and [45] (Lemma 1). Everywhere in this subsection we use a generic notation $\mathbf{A}_1^c$ as an alias for the random matrix $\mathbf{A}_{1,1}^c$. Now, using **TD** 3 and (5), we get
$$\|\mathbf{A}_1^c\| \leq (1+\gamma)$$
almost surely, which implies $\|\bar{\mathbf{A}}^c\| \leq 1+\gamma$ for any $c \in [N]$, giving (44). This implies, using the definition of $\Sigma_{\tilde{\mathbf{A}}}^c$, that
$$\|\Sigma_{\tilde{\mathbf{A}}}^c\| = \|\mathbb{E}[\{\mathbf{A}_1^c\}^\top \mathbf{A}_1^c] - \{\bar{\mathbf{A}}^c\}^\top \bar{\mathbf{A}}^c\| \leq 2(1+\gamma)^2 ,$$
and the bound (45) follows. Next we observe that
$$\begin{aligned}
\mathrm{Tr}(\Sigma_\varepsilon^c) &= \mathbb{E}[\|(\mathbf{A}_1^c - \bar{\mathbf{A}}^c)\theta_\star^c - (\mathbf{b}_1^c - \bar{\mathbf{b}}^c)\|^2] \\
&\leq 2\{\theta_\star^c\}^\top \mathbb{E}[\{\mathbf{A}_1^c\}^\top \mathbf{A}_1^c]\theta_\star^c + 2\mathbb{E}[(r^s(S_0^s, A_0^c))^2 \, \mathrm{Tr}(\varphi(S_0^c)\varphi^\top(S_0^c))] \\
&\leq 2(1+\gamma)^2 \{\theta_\star^c\}^\top \Sigma_\varphi[c]\theta_\star^c + 2 \\
&\leq 2(1+\gamma)^2 \left(\|\theta_\star^c\|^2 + 1\right) ,
\end{aligned}$$
where the latter inequality follows from **TD** 3, and thus (46) holds. In order to check the last equation (47), we note first that the bound for $a$ and $\eta_\infty$ readily follows from the ones presented in [42][Lemma 5] and [42][Lemma 7]. To check assumption A4, note first that, with $s \sim \mu^c, s' \sim P^\pi(\cdot|s)$, we have
$$\begin{aligned}
\mathbf{A}^c + \{\mathbf{A}^c\}^\top &= \varphi(s)\{\varphi(s) - \gamma\varphi(s')\}^\top + \{\varphi(s) - \gamma\varphi(s')\}\varphi(s)^\top \\
&= 2\varphi(s)\varphi(s)^\top - \gamma\{\varphi(s)\varphi(s')^\top + \varphi(s')\varphi(s)^\top\} \\
&\preceq (2+\gamma)\varphi(s)\varphi(s)^\top + \gamma\varphi(s')\varphi(s')^\top ,
\end{aligned}$$
where we additionally used that
$$-(uu^\top + vv^\top) \preceq uv^\top + vu^\top \preceq (uu^\top + vv^\top)$$
for any $u, v \in \mathbb{R}^d$. Thus, we get that
$$\mathbb{E}[\mathbf{A}^c + \{\mathbf{A}^c\}^\top] \preceq 2(1+\gamma)\Sigma_\varphi^c .$$
The rest of the proof follows from the fact that
$$\mathbb{E}[\{\mathbf{A}_1^c\}^\top \mathbf{A}_1^c] \succeq \{\bar{\mathbf{A}}^c\}^\top \bar{\mathbf{A}}^c \succeq (1-\gamma)^2\lambda_{\min}\Sigma_\varphi^c ,$$
which holds whenever (48) is satisfied; see e.g. in [30] (Lemma 5) or [45] (Lemma 7).

Based on these results, we instantiate the results summarized in Table 1 to Federated TD learning in Table 2.

## F  Analysis of Scaffnew for Federated LSA

To mitigate the bias caused by local training, we may use control variates. We assume in this section that at each iteration we choose, with probability $p$, whether agents should communicate or not. Consider the following algorithm, where for $k = 1, \ldots, T/p$, we compute
$$\hat{\theta}_k^c = \theta_{k-1}^c - \eta(\mathbf{A}^c(Z_k^c)\theta_{k-1}^c - \mathbf{b}^c(Z_k^c) - \xi_{k-1}^c) ,$$

**Algorithm 6** "Scaffnew": Stochastic Controlled FedLSA with probabilistic communication

---

**Input:** $\eta > 0$, $\theta_0, \xi_0^c \in \mathbb{R}^d$, $T, N, H, p > 0$
Set: $K = T/p$
**for** $k = 1$ to $K$ **do**
  **for** $c = 1$ to $N$ **do**
    Receive $Z_k^c$ and perform local update:

$$\hat{\theta}_k^c = \hat{\theta}_{k-1}^c - \eta(\mathbf{A}^c(Z_k^c)\hat{\theta}_{k-1}^c - \mathbf{b}^c(Z_k^c) - \xi_{k-1}^c)$$

    Draw $B_k \sim \text{Bernoulli}(p)$
    **if** $B_k = 1$ **then**
      Average local iterates: $\theta_k^c = \frac{1}{N}\sum_{c=1}^N \hat{\theta}_k^c$
      Update: $\xi_k^c = \xi_{k-1}^c + \frac{p}{\eta}(\theta_k^c - \hat{\theta}_k^c)$
    **else**
      Set: $\theta_k^c = \hat{\theta}_k^c, \xi_k^c = \xi_{k-1}^c$

---

i.e. we update the local parameters with LSA adjusted with a control variate $\xi_{k-1}^c$. This control variate is initialized to zero, and updated after each communication round. We draw a Bernoulli random variable $B_k$ with success probability $p$ and then update the parameter as follows:

$$\theta_k^c = \begin{cases} \bar{\theta}_k = \frac{1}{N}\sum_{c=1}^N \hat{\theta}_k^c & B_k = 1 \ , \\ \hat{\theta}_k^c & B_k = 0 \ . \end{cases}$$

We then update the control variate

$$\xi_k^c = \xi_{k-1}^c + \frac{p}{\eta}(\theta_k^c - \hat{\theta}_k^c) \ .$$

where we have set $\xi_0^c = 0$. We state this algorithm in Algorithm 6.

Note that, for all $k \in \mathbb{N}$, $\sum_{c=1}^N \xi_t^c = 0$. . We now proceed to the proof, which amounts to constructing a common Lyapunov function for the sequences $\{\theta_k^c\}_{k\in\mathbb{N}}$ and $\{\xi_k^c\}_{k\in\mathbb{N}}$. Define the Lyapunov function,

$$\psi_k = \frac{1}{N}\sum_{c=1}^N \|\theta_k^c - \theta_\star\|^2 + \frac{\eta^2}{p^2}\frac{1}{N}\sum_{c=1}^N \|\xi_k^c - \xi_\star^c\|^2 \ ,$$

where $\theta_\star$ is the solution of $\bar{A}\theta_\star = \bar{\mathbf{b}}$, and $\xi_\star^c = \bar{\mathbf{A}}^c(\theta_\star - \theta_\star^c)$. A natural measure of heterogeneity is then given by

$$\Delta_{\text{heter}} = \frac{1}{N}\sum_{c=1}^N \|\xi_\star^c\|^2 = \frac{1}{N}\sum_{c=1}^N \|\bar{\mathbf{A}}^c(\theta_\star^c - \theta_\star)\|^2 \ .$$

To analyze this algorithm, we'll study the decrease of the expected value of $\psi_k$, where the expectation is over randomness of the communication and the stochastic oracles. This requires a stronger assumption than the Assumption A3 that we used in Section 4.

**A4.** *There exist constants $a, L > 0$, such that for any $\eta \in (0, 1/L)$, $c \in [N]$, it holds for $Z_1^c \sim \pi_c$, that*

$$a\mathrm{I} \preccurlyeq \mathbb{E}[\tfrac{1}{2}(\mathbf{A}^c(Z_1^c) + \mathbf{A}^c(Z_1^c)^\top)] \preccurlyeq \tfrac{1}{L}\mathbb{E}[\mathbf{A}^c(Z_1^c)^\top \mathbf{A}^c(Z_1^c)] \ .$$

This assumption is slightly more restrictive than A3. Indeed, whenever A4 holds, A3 also holds with the same constant $a$ (see 42, 45). In the case of TD, this assumption holds with $L = \frac{1+\gamma}{(1-\gamma)\nu}$.

**Lemma F.1** (One step progress). *Assume A1 and A4. Assume that $\eta \leq \frac{1}{2L}$. The iterates of the algorithm described above satisfy*

$$\mathbb{E}[\psi_k] \leq \left(1 - \min\left(\eta a, p^2\right)\right)\mathbb{E}[\psi_{k-1}] + \frac{2\eta^2}{N}\sum_{c=1}^N \text{Tr}(\Sigma_\varepsilon^c) \ .$$

*Proof.* **Decomposition of the update.** Remark that the update can be reformulated as

$$\hat{\theta}_k^c - \theta_\star = (\mathrm{I} - \eta\mathbf{A}^c(Z_k^c))(\theta_{k-1}^c - \theta_\star) + \eta(\xi_{k-1}^c - \xi_\star^c) - \eta\omega^c(Z_k^c) \ , \tag{49}$$

where $\omega^c(z) = \widetilde{\mathbf{A}}^c(z)\theta_\star - \widetilde{\mathbf{b}}^c(z)$. This comes from the fact that, for all $z$,

$$
\begin{aligned}
\mathbf{b}^c(z) + \xi_{k-1}^c &= \bar{\mathbf{b}}^c + \widetilde{\mathbf{b}}^c(z) + \xi_{k-1}^c \\
&= \bar{\mathbf{A}}^c \theta_\star^c + \widetilde{\mathbf{b}}^c(z) + \xi_{k-1}^c \\
&= \bar{\mathbf{A}}^c \theta_\star + \widetilde{\mathbf{b}}^c(z) + \xi_{k-1}^c - \xi_\star^c \\
&= \mathbf{A}^c(z)\theta_\star - \widetilde{\mathbf{A}}^c(z)\theta_\star + \widetilde{\mathbf{b}}^c(z) + \xi_{k-1}^c - \xi_\star^c \\
&= \mathbf{A}^c(z)\theta_\star - \omega^c(z) + \xi_{k-1}^c - \xi_\star^c \,.
\end{aligned}
$$

**Expression of communication steps.** Using that $\sum_{c=1}^N \xi_{k-1}^c = 0$ and $\sum_{c=1}^N \xi_\star^c = 0$, we get

$$
\frac{1}{N} \sum_{c=1}^N \|\theta_k^c - \theta_\star\|^2 = \mathbf{1}_{\{1\}}(B_k)\|\bar{\theta}_k - \theta_\star\|^2 + \mathbf{1}_{\{0\}}(B_k)\frac{1}{N} \sum_{c=1}^N \|\hat{\theta}_k^c - \theta_\star\|^2
$$

$$
= \mathbf{1}_{\{1\}}(B_k)\|\frac{1}{N} \sum_{c=1}^N (\hat{\theta}_k^c - \frac{\eta}{p}\xi_{k-1}^c) - \frac{1}{N} \sum_{c=1}^N (\theta_\star - \frac{\eta}{p}\xi_\star^c)\|^2 + \mathbf{1}_{\{0\}}(B_k)\frac{1}{N} \sum_{c=1}^N \|\hat{\theta}_k^c - \theta_\star\|^2 \,.
$$

The first term can be upper bounded by using Lemma D.3, which gives

$$
\mathbf{1}_{\{1\}}(B_k)\|\bar{\theta}_k - \theta_\star\|^2
$$

$$
= \mathbf{1}_{\{1\}}(B_k)\left\{ \frac{1}{N} \sum_{c=1}^N \|\hat{\theta}_k^c - \frac{\eta}{p}(\xi_{k-1}^c - \xi_\star^c) - \theta_\star\|^2 - \frac{1}{N} \sum_{c=1}^N \|\bar{\theta}_k - (\hat{\theta}_k^c - \frac{\eta}{p}\xi_{k-1}^c) + \frac{\eta}{p}\xi_\star^c\|^2 \right\}
$$

$$
= \mathbf{1}_{\{1\}}(B_k)\left\{ \frac{1}{N} \sum_{c=1}^N \|\hat{\theta}_k^c - \frac{\eta}{p}(\xi_{k-1}^c - \xi_\star^c) - \theta_\star\|^2 - \frac{\eta^2}{p^2}\frac{1}{N} \sum_{c=1}^N \|\xi_k^c - \xi_\star^c\|^2 \right\} \,.
$$

We now expand the first term in the right-hand side of the previous equation. This gives

$$
\frac{1}{N} \sum_{c=1}^N \|\hat{\theta}_k^c - \frac{\eta}{p}(\xi_{k-1}^c - \xi_\star^c) - \theta_\star\|^2
$$

$$
= \frac{1}{N} \sum_{c=1}^N \left\{ \|\hat{\theta}_k^c - \theta_\star\|^2 - \frac{2\eta}{p}\langle \xi_{k-1}^c - \xi_\star^c, \hat{\theta}_k^c - \theta_\star\rangle + \frac{\eta^2}{p^2}\|\xi_{k-1}^c - \xi_\star^c\|^2 \right\} \,,
$$

which yields

$$
\mathbf{1}_{\{1\}}(B_k)\{\psi_k\} = \mathbf{1}_{\{1\}}(B_k)\left\{ \|\bar{\theta}_k - \theta_\star\|^2 + \frac{\eta^2}{p^2}\frac{1}{N} \sum_{c=1}^N \|\xi_k^c - \xi_\star^c\|^2 \right\}
$$

$$
= \mathbf{1}_{\{1\}}(B_k)\left\{ \frac{1}{N} \sum_{c=1}^N \|\hat{\theta}_k^c - \theta_\star\|^2 - \frac{2\eta}{p}\langle \xi_{k-1}^c - \xi_\star^c, \hat{\theta}_k^c - \theta_\star\rangle + \frac{\eta^2}{p^2}\frac{1}{N} \sum_{c=1}^N \|\xi_{k-1}^c - \xi_\star^c\|^2 \right\} \,. \quad (50)
$$

On the other hand, note that

$$
\mathbf{1}_{\{0\}}(B_k)\{\psi_k\} = \mathbf{1}_{\{0\}}(B_k)\left\{ \frac{1}{N} \sum_{c=1}^N \|\theta_k^c - \theta_\star\|^2 + \frac{\eta^2}{p^2}\frac{1}{N} \sum_{c=1}^N \|\xi_k^c - \xi_\star^c\|^2 \right\}
$$

$$
= \mathbf{1}_{\{0\}}(B_k)\left\{ \frac{1}{N} \sum_{c=1}^N \|\hat{\theta}_k^c - \theta_\star\|^2 + \frac{\eta^2}{p^2}\frac{1}{N} \sum_{c=1}^N \|\xi_{k-1}^c - \xi_\star^c\|^2 \right\} \,. \quad (51)
$$

By combining (51) and (50), we get

$$
\psi_k = \frac{1}{N} \sum_{c=1}^N \|\theta_k^c - \theta_\star\|^2 + \frac{\eta^2}{p^2}\frac{1}{N} \sum_{c=1}^N \|\xi_k^c - \xi_\star^c\|^2
$$

$$
= \frac{1}{N} \sum_{c=1}^N \|\hat{\theta}_k^c - \theta_\star\|^2 - 2\frac{\eta}{p}\mathbf{1}_{\{1\}}(B_k)\langle \xi_{k-1}^c - \xi_\star^c, \hat{\theta}_k^c - \theta_\star\rangle + \frac{\eta^2}{p^2}\frac{1}{N} \sum_{c=1}^N \|\xi_{k-1}^c - \xi_\star^c\|^2 \,. \quad (52)
$$

**Progress in local updates.** We now bound the first term of the sum in (52). For $c \in [N]$, (49) gives

$$
\begin{aligned}
\|\hat{\theta}_k^c - \theta_\star\|^2 &= \|(I - \eta\mathbf{A}^c(Z_k^c))(\theta_{k-1}^c - \theta_\star) + \eta(\xi_{k-1}^c - \xi_\star^c) - \eta\omega^c(Z_k^c)\|^2 \\
&= \|(I - \eta\mathbf{A}^c(Z_k^c))\{\theta_k^c - \theta_\star\} - \eta\omega^c(Z_k^c)\|^2 + \eta^2\|\xi_{k-1}^c - \xi_\star^c\|^2 \\
&\quad + 2\eta\langle \xi_{k-1}^c - \xi_\star^c,\, (I - \eta\mathbf{A}^c(Z_k^c))\{\theta_k^c - \theta_\star\} - \eta\omega^c(Z_k^c)\rangle \\
&= \underbrace{\|(I - \eta\mathbf{A}^c(Z_k^c))\{\theta_k^c - \theta_\star\} - \eta\omega^c(Z_k^c)\|^2}_{T} + 2\eta\langle \xi_{k-1}^c - \xi_\star^c,\, \hat{\theta}_k^c - \theta_\star\rangle - \eta^2\|\xi_{k-1}^c - \xi_\star^c\|^2 .
\end{aligned}
\tag{53}
$$

Define the $\sigma$-algebra $\mathcal{G}_{k-1} = \sigma(B_s, s \le k-1, Z_s^c, s \le k-1, c \in [N])$. We now bound the conditional expectation of $T_1$

$$
\begin{aligned}
&\mathbb{E}^{\mathcal{G}}[T_1] \\
&= \mathbb{E}^{\mathcal{G}}\left[ \|(I - \eta\mathbf{A}^c(Z_k^c))\{\theta_k^c - \theta_\star\}\|^2 - 2\eta\langle (I - \eta\mathbf{A}^c(Z_k^c))\{\theta_k^c - \theta_\star\},\, \omega^c(Z_k^c)\rangle + \eta^2\|\omega^c(Z_k^c)\|^2\right] \\
&= \mathbb{E}^{\mathcal{G}}\left[ \|(I - \eta\mathbf{A}^c(Z_k^c))\{\theta_k^c - \theta_\star\}\|^2 + 2\eta^2\langle \mathbf{A}^c(Z_k^c)\{\theta_k^c - \theta_\star\},\, \omega^c(Z_k^c)\rangle + \eta^2\|\omega^c(Z_k^c)\|^2\right] ,
\end{aligned}
$$

where we used the fact that $\langle I,\, \omega^c(Z_k^c)\rangle = 0$. Using Young's inequality for products, and Lemma D.4 with $\eta \le \frac{1}{2L}$ and $u = \theta_k^c - \theta_\star$, we then obtain

$$
\begin{aligned}
&\mathbb{E}^{\mathcal{G}}[T_1] \\
&\le \mathbb{E}^{\mathcal{G}}\left[ \|(I - \eta\mathbf{A}^c(Z_k^c))\{\theta_k^c - \theta_\star\}\|^2 + \eta^2\|\mathbf{A}^c(Z_k^c)\{\theta_k^c - \theta_\star\}\|^2 + \eta^2\|\omega^c(Z_k^c)\|^2 + \eta^2\|\omega^c(Z_k^c)\|^2\right] \\
&\le (1 - \eta a)\|\theta_k^c - \theta_\star\|^2 - \eta(\tfrac{1}{L} - 2\eta)\mathbb{E}^{\mathcal{G}}\left[\|\mathbf{A}^c(Z_k^c)\{\theta_k^c - \theta_\star\}\|^2\right] + 2\eta^2\mathbb{E}^{\mathcal{G}}\left[\|\omega^c(Z_k^c)\|^2\right] .
\end{aligned}
\tag{54}
$$

Plugging (54) in (53) and using the assumption $\eta \le \frac{1}{2L}$, we obtain

$$
\begin{aligned}
\mathbb{E}^{\mathcal{G}}\left[\|\hat{\theta}_k^c - \theta_\star\|^2 - 2\eta\langle \xi_{k-1}^c - \xi_\star^c,\, \hat{\theta}_k^c - \theta_\star\rangle\right] \\
\le (1 - \eta a)\|\theta_k^c - \theta_\star\|^2 - \eta^2\|\xi_{k-1}^c - \xi_\star^c\|^2 + 2\eta^2\operatorname{Tr}(\Sigma_\varepsilon^c) .
\end{aligned}
\tag{55}
$$

**Bounding the Lyapunov function.** Taking the condtional expectation of (52) and using (55) for $c = 1$ to $N$, we obtain the following bound on the Lyapunov function

$$
\begin{aligned}
\mathbb{E}^{\mathcal{G}}[\psi_k] &= \frac{1}{N}\sum_{c=1}^{N}\mathbb{E}^{\mathcal{G}}\left[\|\hat{\theta}_k^c - \theta_\star\|^2 - 2\eta\langle \xi_{k-1}^c - \xi_\star^c,\, \hat{\theta}_k^c - \theta_\star\rangle\right] + \frac{\eta^2}{p^2}\frac{1}{N}\sum_{c=1}^{N}\|\xi_{k-1}^c - \xi_\star^c\|^2 \\
&\le \frac{1}{N}\sum_{c=1}^{N}\left[(1 - \eta a)\|\theta_k^c - \theta_\star\|^2 - \eta^2\|\xi_{k-1}^c - \xi_\star^c\|^2 + 2\eta^2\operatorname{Tr}(\Sigma_\varepsilon^c)\right] + \frac{\eta^2}{p^2}\frac{1}{N}\sum_{c=1}^{N}\|\xi_{k-1}^c - \xi_\star^c\|^2 \\
&= (1 - \eta a)\frac{1}{N}\sum_{c=1}^{N}\|\theta_k^c - \theta_\star\|^2 + (1 - p^2)\frac{\eta^2}{p^2}\frac{1}{N}\sum_{c=1}^{N}\|\xi_{k-1}^c - \xi_\star^c\|^2 + \frac{2\eta^2}{N}\sum_{c=1}^{N}\operatorname{Tr}(\Sigma_\varepsilon^c) ,
\end{aligned}
$$

and the result of the Lemma follows from the Tower property. $\qquad\square$

**Theorem F.2** (Convergence rate)**.** *Assume A1 and A3(2). Then, for any $\eta \le \frac{1}{2L}$ and $T > 0$, it holds*

$$
\mathbb{E}[\psi_K] \le \left(1 - \zeta\right)^K\left(\|\theta_0 - \theta_\star\|^2 + \frac{\eta^2}{p^2}\Delta_{\mathrm{heter}}\right) + \frac{2\eta^2}{\zeta}\frac{1}{N}\sum_{c=1}^{N}\operatorname{Tr}(\Sigma_\varepsilon^c) ,
$$

*where $\zeta = \min\left(\eta a, p^2\right)$.*

**Corollary F.3** (Iteration complexity)**.** *Let $\epsilon > 0$. Set $\eta = \min\left(\frac{1}{2L}, \frac{\epsilon a}{8\bar{\sigma}}\right)$ and $p = \sqrt{\eta a}$ (so that $\zeta = \eta a$). Then, $\mathbb{E}[\psi_K] \le \epsilon^2$ as long as the number of iterations is*

$$
K \ge \max\left(\frac{2L}{a}, \frac{4\bar{\sigma}_\varepsilon}{\epsilon^2 a^2}\right)\log\left(\frac{\|\theta_0 - \theta_\star\|^2 + \min\left(\frac{1}{2aL}, \frac{\epsilon}{8\bar{\sigma}}\right)\Delta_{\mathrm{heter}}}{2\epsilon^2}\right) ,
$$

*which corresponds to an expected number of communication rounds*

$$
T \ge \max\left(\sqrt{\frac{2L}{a}}, \sqrt{\frac{4\bar{\sigma}_\varepsilon}{\epsilon^2 a^2}}\right)\log\left(\frac{\|\theta_0 - \theta_\star\|^2 + \min\left(\frac{1}{2aL}, \frac{\epsilon}{8\bar{\sigma}}\right)\Delta_{\mathrm{heter}}}{2\epsilon^2}\right) .
$$

**Theorem F.4** (No linear speedup in the probabilistic communication setting with control variates)**.** *The bounds obtained in Theorem F.2 are minimax optimal up to constants that are independent from the problem. Precisely, for every $(p, \eta)$ there exists a FLSA problem such that*

$$\mathbb{E}[\psi_K] = (1 - \zeta)^K \left( \|\theta_0 - \theta_\star\|^2 + \frac{\eta^2}{p^2} \Delta_{\mathrm{heter}} \right) + \frac{2\eta^2}{\zeta} \bar{\sigma}_\varepsilon \ ,$$

*where we have defined $\zeta = \min \left( 2\eta a, p^2 \right)$.*

*Proof.* Define for all $c \in [N]$,

$$\bar{\mathbf{A}}^c = a\mathbf{I} \ , \quad \bar{\mathbf{b}}^c = b_c u \ ,$$

where u is a vector whom all coordinates are equal to 1. We also consider the sequence of i.i.d random variables $(Z_k^c)$ such that that for all $c \in [N]$ and $0 \leq t \leq T$, $Z_k^c$ follows a Rademacher distribution. Moreover, we define

$$\mathbf{A}^c(Z_k^c) = \bar{\mathbf{A}}^c \ , \quad \mathbf{b}^c(Z_k^c) = \bar{\mathbf{b}}^c + Z_k^c u \ .$$

In particular this implies

$$\omega^c(z) = Z_k^c u \ .$$

We follow the same proof of Lemma F.1 until the chain of equalities breaks. Thereby, we start from

$$\mathbb{E}[\psi_k] = \mathbb{E}\left[\sum_{c=1}^N \|\theta_k^c - \theta_\star\|^2 + \frac{\eta^2}{p^2} \sum_{c=1}^N \|\xi_k^c - \xi_\star^c\|^2\right]$$

$$= \mathbb{E}\left[\sum_{c=1}^N \|(\mathbf{I} - \eta\mathbf{A}^c(Z_k^c))\{\theta_{k-1}^c - \theta_\star\} - \eta\omega^c(Z_k^c))\|^2 + (1 - p^2)\frac{\eta^2}{p^2} \sum_{c=1}^N \|\xi_{k-1}^c - \xi_\star^c\|^2\right]$$

$$= \mathbb{E}\left[\sum_{c=1}^N \|(\mathbf{I} - \eta\bar{\mathbf{A}}^c)\{\theta_{k-1}^c - \theta_\star\} - \eta\omega^c(Z_k^c))\|^2 + (1 - p^2)\frac{\eta^2}{p^2} \sum_{c=1}^N \|\xi_{k-1}^c - \xi_\star^c\|^2\right]$$

$$= \mathbb{E}\left[\sum_{c=1}^N (1 - \eta a)^2 \|\theta_{k-1}^c - \theta_\star\|^2 + \eta^2 \|\omega^c(Z_k^c)\|^2 + (1 - p^2)\frac{\eta^2}{p^2} \sum_{c=1}^N \|\xi_{k-1}^c - \xi_\star^c\|^2\right]$$

where we used that $\mathbf{A}^c(Z_k^c) = \bar{\mathbf{A}}^c$. Unrolling the recursion gives the desired result. $\qquad \square$

# G   Experimental Details and Additional Experiments

## G.1   Experimental Details

Here, we give additional details regarding the numerical experiments. The environments used are instances of Garnet, where we use 30 states, embedded via a random projection in a $d = 8$-dimensional space. We use two actions, and consider a branching factor of two, meaning that, from each state, one can transition to two different states with some probability. The rewards are then drawn uniformly randomly from the interval $[0, 1]$.

In the homogeneous setting, we sample one Garnet environment. Each client then receives a perturbation of this instance, where we perturb all non-zeros probabilities of transition from one state to another and all rewards with a random variable $\epsilon \sim \mathcal{U}(0, 0.02)$.

In the heterogeneous setting, we proceed similarly, except that we sample two different Garnet environments, with the same parameters. Half of the agents receive the first environment, and the second half receive the second environment. As in the homogeneous setting, each agent's environment slightly differs from the base environment by a small perturbation $\epsilon \sim \mathcal{U}(0, 0.02)$.

All the experiments presented in this paper can be run on a single laptop in just a few hours.

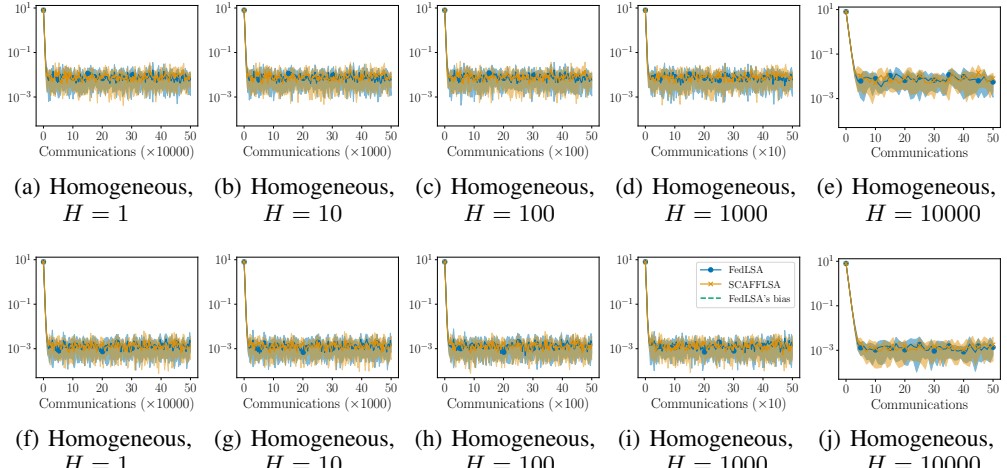

Figure 3: MSE as a function of the number of communication rounds for FedLSA and SCAFFLSA applied to federated TD(0) in homogeneous settings with $\eta = 0.1$, for different number of agents ($N = 10$ on the first line, $N = 100$ on the second line) and different number of local steps. Green dashed line is FedLSA's bias, as predicted by Theorem 4.1. For each algorithm, we report the average MSE and variance over 5 runs.

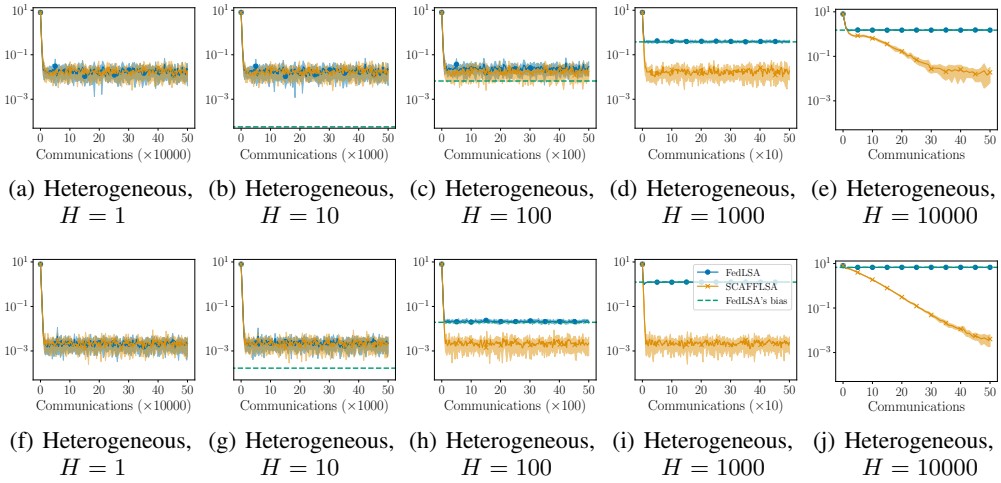

Figure 4: MSE as a function of the number of communication rounds for FedLSA and SCAFFLSA applied to federated TD(0) in heterogeneous settings with $\eta = 0.1$, for different number of agents ($N = 10$ on the first line, $N = 100$ on the second line) and different number of local steps. Green dashed line is FedLSA's bias, as predicted by Theorem 4.1. For each algorithm, we report the average MSE and variance over 5 runs.

### G.2 Additional Experiments: Number of Local Steps and Smaller Step-Size

In this section, we give more experimental results for FedLSA and SCAFFLSA. We use the same setting as in Section 6, but use more settings of local steps.

In Figure 3 and Figure 4, we give report the counterpart of Figure 1 with a wider ranger of number of local updates $H \in \{1, 10, 100, 1000, 10000\}$. The results obtained here match with observations from Section 6: in homogeneous settings, FedLSA and SCAFFLSA exhibit very similar behavior. In both methods, increasing the number of local steps speeds-up the training, until the stochastic noise dominates. At this point, both algorithms reach a stationary regime with similar error. In heterogeneous settings, while FedLSA's bias is smaller than the variance of its iterates, train-

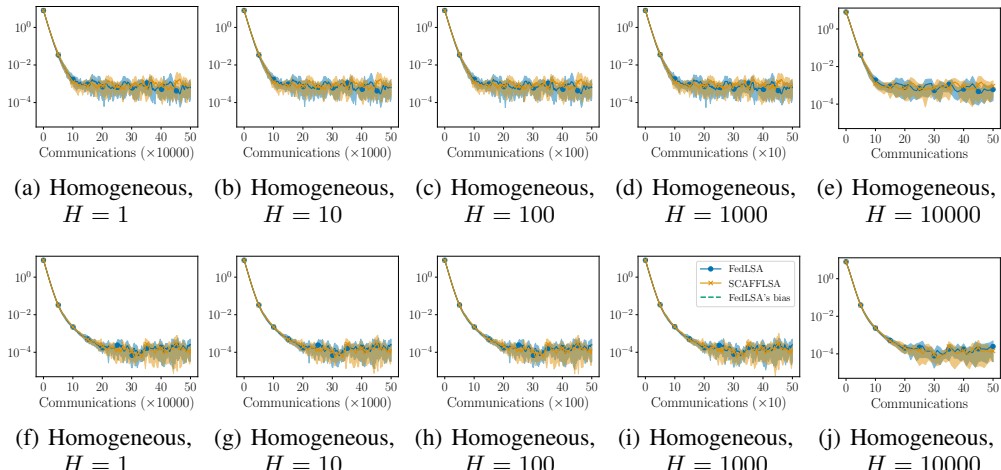

Figure 5: MSE as a function of the number of communication rounds for FedLSA and SCAFFLSA applied to federated TD(0) in homogeneous settings with $\eta = 0.01$, for different number of agents ($N = 10$ on the first line, $N = 100$ on the second line) and different number of local steps. Green dashed line is FedLSA's bias, as predicted by Theorem 4.1. For each algorithm, we report the average MSE and variance over $5$ runs.

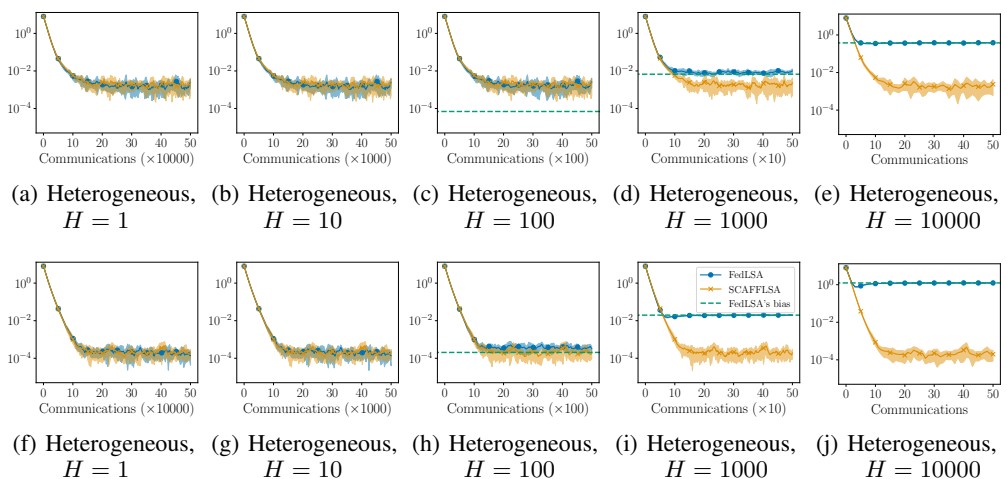

Figure 6: MSE as a function of the number of communication rounds for FedLSA and SCAFFLSA applied to federated TD(0) in heterogeneous settings with $\eta = 0.01$, for different number of agents ($N = 10$ on the first line, $N = 100$ on the second line) and different number of local steps. Green dashed line is FedLSA's bias, as predicted by Theorem 4.1. For each algorithm, we report the average MSE and variance over $5$ runs.

ing speeds up when the number of local steps increases. After that point, bias dominates, while SCAFFLSApreserves the speed-up by eliminating this bias.

Finally, we report in Figure 5 and Figure 6 the results when running the same experiments using a smaller step size $\eta = 0.01$ for different number of agents and local updates. In this setting, all algorithms manage to find better estimators, since the amount of variance depends on the step size (as seen in Theorem 4.1 and Theorem 5.1). Additionally, FedLSA's bias is smaller than in Figure 7, which is also in line with the upper bound $\mathbb{E}^{1/2}[\|\tilde{\theta}_t^{(\mathsf{bi},\mathsf{bi})}\|^2] \lesssim \frac{\eta H \mathbb{E}[\|\theta - \theta\|]}{a}$ from Theorem 4.1.

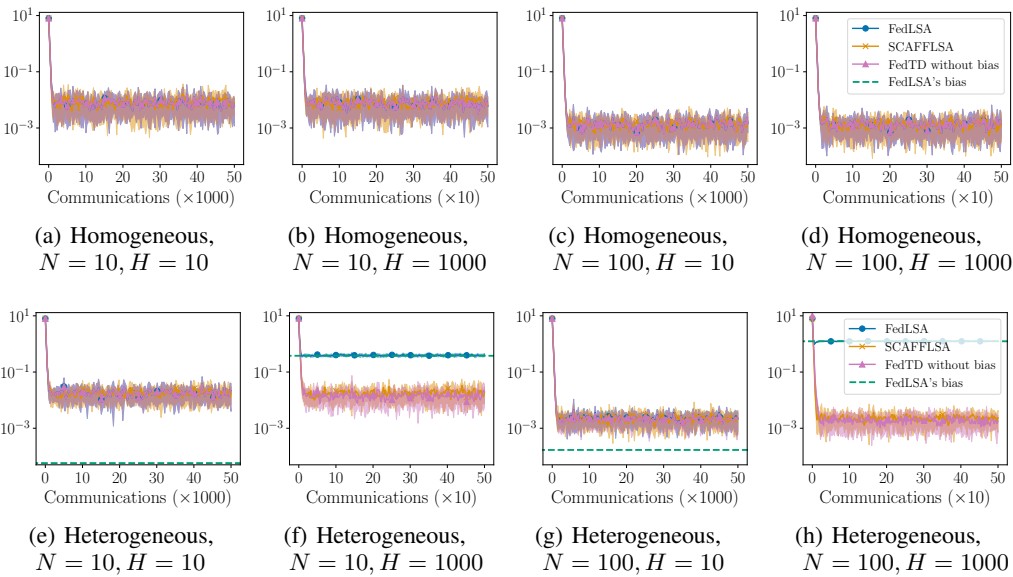

Figure 7: MSE as a function of the number of communication rounds for FedLSA and SCAFFLSA applied to federated TD(0) in homogeneous and heterogeneous settings, for different number of agents and number of local steps. Green dashed line is FedLSA's bias, as predicted by Theorem 4.1. For each algorithm, we report the average MSE and variance over 5 runs.

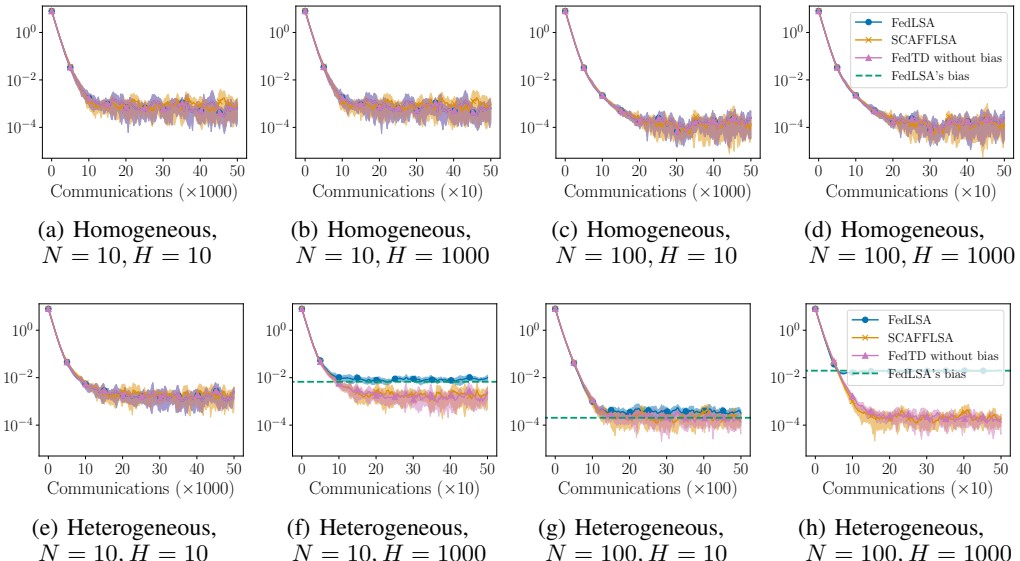

Figure 8: MSE as a function of the number of communication rounds for FedLSA and SCAFFLSA applied to federated TD(0) in homogeneous and heterogeneous settings, for different number of agents and number of local steps, using a smaller step size $\eta = 0.01$. Green dashed line is FedLSA's bias, as predicted by Theorem 4.1. For each algorithm, we report the average MSE and variance over 5 runs.

## G.3 Additional Experiments: Convergence of FedLSA

In Figure 7 and Figure 8, we give the counterpart of Figure 1, where we additionally plot the MSE of the estimator $\theta_t + \tilde{\theta}_\infty^{(\mathrm{bi,bi})}$, for different settings of all parameters. We recall that $\tilde{\theta}_\infty^{(\mathrm{bi,bi})} = (\mathrm{I} - \bar{\Gamma}_H^{(\eta)})^{-1}\bar{\rho}_H$ is the bias of FedLSA, as we proved in Theorem 4.1. Therefore, $\theta_t + \tilde{\theta}_\infty^{(\mathrm{bi,bi})}$ is a

proper estimator of $\theta_\star$, although, of course, it cannot be computed in practice since the bias $\tilde{\theta}_\infty^{(\mathrm{bi},\mathrm{bi})}$ is unknown. and we see in Figure 7 that, in homogeneous settings, we recover the same error as FedLSA and SCAFFLSA. Moreover, in heterogeneous settings, it has an error similar to the one of SCAFFLSA, meaning that FedLSA, once its bias is removed, converges similarly to SCAFFLSA. The latter, however, does not require to remove an unknown bias, and directly estimates the right quantity.

