# OpenReview forum: "SCAFFLSA: Taming Heterogeneity in Federated Linear Stochastic Approximation and TD Learning"
_NeurIPS.cc/2024/Conference — NeurIPS 2024 poster_

### Official Review · Reviewer_czvr · 2024-07-09

**Soundness:** 3
**Presentation:** 2
**Contribution:** 2
**Rating:** 6
**Confidence:** 2

**Summary:**

This paper analyzes the convergence of scaffnew in the quadratic setup and achieves a linear speedup in the number of clients.  It is not attained by the original paper of Scaffnew. Additionaly, the author find an application of the federated quadratic problem -- Federated
Linear Stochastic Approximation and TD Learning.

**Strengths:**

This paper tries to analyze the convergence of scaffnew in the quadratic setup and achieves a linear speedup in the number of clients.  It is not attained by the original paper of Scaffnew.

**Weaknesses:**

The achieved speedup holds only for quadratic setup and the application of the quadratic loss function is quite limited.
The experiments are a little simple.

**Questions:**

According to my understanding, this work saves more communication complexity compared with scaffnew, while it is not claimed by the authors. This is observed in Table 1. Is it another advantage of this analysis?

Can the theoretical analysis be extended to a general case?

The comparison in Figure 1 is hard to catch. Can you plot them in one figure for comparison, at least using the same legend? And considering more value of H and N.

**Limitations:**

The achieved speedup holds only for quadratic setup and the application of the quadratic loss function is quite limited.

---

> ### Author Rebuttal · Authors · 2024-08-02
>
> **The achieved speedup holds only for quadratic setup and the application of the quadratic loss function is quite limited.**
> We emphasize that our analysis holds for the general setting of linear stochastic approximation, which encompasses minimization of quadratic functions, but also works for other settings where the system matrix is not symmetric. This is crucial, as the matrices involved in TD learning are not symmetric. We stress that our analysis is the first of its kind for federated LSA.
>
> **The experiments are a little simple.**
> The Garnet problems used in our experiments are common for federated TD, and the only problem that we are aware of in the federated TD literature. We stress that our paper is a theoretical paper, and the purpose of our experiment is thus only to illustrate our theory numerically.
> Nonetheless, we are happy to add more experiments on other problems if the reviewer has some specific problems to suggest.
>
> **According to my understanding, this work saves more communication complexity compared with scaffnew, while it is not claimed by the authors. This is observed in Table 1. Is it another advantage of this analysis?**
> This is indeed the case, thank you for pointing it out. We will make sure to add a sentence about this in the discussion after Corollaries 5.2 and 5.3.
>
> **Can the theoretical analysis be extended to a general case?**
> Extending our analysis to other settings, e.g. for strongly-convex and smooth problems is much more technical, and is a very interesting direction for future research.
>
> **The comparison in Figure 1 is hard to catch. Can you plot them in one figure for comparison, at least using the same legend? And considering more value of H and N.**
> Thank you for pointing this issue, we will update the plots in the camera ready version, making sure that the y-axis is the same each time. We will also provide addditional plots for $H \in \\{1, 10, 100, 1000, 10000\\}$ and $N \in \\{10, 100\\}$ in the appendix; we add these additional plots to the pdf attached to the global rebuttal. If you have any additional suggestions to improve the experiments in the camera ready, we will be happy to integrate them.

---

> > ### Comment · Reviewer_czvr · 2024-08-12
> >
> > Thanks for your reply. Personally, I am positive about this work. However, we must recognize that there are some limitations in this paper, such as only applying to quadratic problems, the algorithm is directly the same as Scaffnew. Based on these, I will maintain my score!

---

### Official Review · Reviewer_bpaa · 2024-07-13

**Soundness:** 3
**Presentation:** 2
**Contribution:** 3
**Rating:** 7
**Confidence:** 2

**Summary:**

This paper provides a non-asymptotic analysis of Federated Linear Stochastic Approximation. The authors provide (biased and unbiased) finite-time MSE bounds for general LSA and TD learning under the assumption that the noise is i.i.d.. For Markovian noise, only an unbiased MSE bound is provided. Most importantly is that these bounds express this error as a function of the step-size used, number of agents and number of local updates. Finally, a new algorithmic variant, namely, SCAFFLSA is introduced, reducing communication between agents while maintaining the linear speed up in the algorithm. The contributions of the paper are illustrated through numerical studies.

**Strengths:**

The reviewer is not very familiar with the federated learning literature, but as far as they are aware, most of the ideas of the paper are new.

The reviewer was unable to look at all proofs in detail, but have not identified any issues with respect to correctness.

**Weaknesses:**

The organization of the paper is not great. First, it is very hard to follow the main text given the amount of symbols and equations. Secondly, contributions related to i.i.d and Markovian noise and TD learning seemed to be scrambled together. I particularly think that splitting the results into different sections (one for i.i.d. noise, one for Markovian noise and one tailored to TD learning) would improved the readability of the paper a lot.

Also, there is a missing "c" superscript right above equation (1) in Algorithm 1.

The paper should be revised for clarity and typos.

**Questions:**

- Is it possible for the authors provide a table similar to Table 1 in order to illustrate and compare the different outcomes between the three different scenarios analyzed by the paper (i.r. the i.i.d. , Markov and TD setting)?

-Do the authors believe that their analysis could be extended to federated learning algorithms with vanishing step-size?

- It is possible that I missed this in the main text, but are any drawbacks to using SCAFFLSA instead of Federated LSA?

**Limitations:**

The authors identify and state which assumptions are necessary to hold for each of the results to be true, so the limitations of this work are objectively identified.

---

> ### Author Rebuttal · Authors · 2024-08-02
>
> **Is it possible for the authors provide a table similar to Table 1 in order to illustrate and compare the different outcomes between the three different scenarios analyzed by the paper (i.r. the i.i.d. , Markov and TD setting)?**
> Thank you for the suggestion, which will greatly help to improve the readability of our paper. We will add the following table, instantiating our results for TD learning, in the camera ready version. In this table, we highlight the dependence on the discount factor $\gamma$ and on the smallest eigenvalue of the $\Sigma_\varphi^c$ (see Assumption TD3).
> Regarding Markovian sampling, complexity is the same as i.i.d. setting up to a multiplicative factor $\max_c \tau_{mix}(c)$ (as defined in Assumption A2) in the number of local samples $H$. We will add a sentence in both tables' captions mentioning this.
>
> | Algorithm                   | Communication complexity $T$                                                                                         | Local updates $H$                               | Sample complexity $TH$                                                                                           |
> |-----------------------------|------------------------------------------------------------------------------------------------------------------------|--------------------------------------------------|---------------------------------------------------------------------------------------------------------------------|
> | FedTD (Doan 2020)          | $\mathcal{O}\left(\tfrac{N^2}{(1-\gamma)^{2} \nu^{2} \epsilon^2} \log \tfrac{1}{\epsilon}\right)$                    | $1$                                            | $\mathcal{O}\left(\tfrac{N^2}{(1-\gamma)^{2} \nu^{2} \epsilon^2} \log \tfrac{1}{\epsilon}\right)$                 |
> | FedTD (Cor 4.4)            | $\mathcal{O} \left({\tfrac{1}{ (1-\gamma)^{2} \nu^{2} \epsilon}} \log\tfrac{1}{\epsilon} \right)$                     | $\mathcal{O}\bigl(\tfrac{1}{ N \epsilon } \bigr)$ | $\mathcal{O}\left(\tfrac{1}{N (1-\gamma)^2\nu^2\epsilon^2} \log\tfrac{1}{\epsilon}\right)$                        |
> | SCAFFTD (Cor. 5.3)      | $\mathcal{O}\left(\tfrac{1}{(1-\gamma)^2\nu^2} \log\tfrac{1}{\epsilon}\right)$                                        | $\mathcal{O}\bigl(\tfrac{1}{N\epsilon^2} \bigr)$ | $\mathcal{O}\left(\tfrac{1}{N (1-\gamma)^2\nu^2\epsilon^2} \log\tfrac{1}{\epsilon}\right)$                         |
>
> **The organization of the paper is not great. First, it is very hard to follow the main text given the amount of symbols and equations. Secondly, contributions related to i.i.d and Markovian noise and TD learning seemed to be scrambled together. I particularly think that splitting the results into different sections (one for i.i.d. noise, one for Markovian noise and one tailored to TD learning) would improved the readability of the paper a lot.**
> We will make the separation between results on general federated LSA and federated TD more clear, by stating first the general results in i.i.d. and markovian settings (in two different paragraph), then moving to federated TD. We will also use the additional page in the camera ready to give more details regarding equations and mathematical symbols. If you have any additional suggestion regarding presentation, we are happy to hear it to try to make the paper as clear as possible.
>
> **Do the authors believe that their analysis could be extended to federated learning algorithms with vanishing step-size?**
> The analysis could indeed easily be extended to federated linear stochastic approximation with vanishing step sizes, using the same technique. We refrained from doing so due to space limitation.
>
> Regarding extension to more general "federated learning", we want to emphasize that our analysis already works for the special case of quadratic problems $\min_\theta \| X \theta - y \|^2$, that can be cast as $X^\top X \theta = X^\top y$, and fit our assumptions as long as $X^\top X$ is invertible. Still, we stress that our analysis is more general since we do not assume $\bar{A}^c$ to be symmetric.
> Extending our analysis to other settings, e.g. for strongly-convex and smooth problems is much more technical, and is a very interesting direction for future research.
>
> **It is possible that I missed this in the main text, but are any drawbacks to using SCAFFLSA instead of Federated LSA?**
> This is the main claim of our paper: in terms of sample and communication complexity, SCAFFLSA is guaranteed to perform as good as FedLSA in all settings, and can significantly reduce communication cost in heterogeneous settings. We show this theoretically, and illustrate it numerically; we refer the reviewer to the pdf attached to the general rebuttal for additional numerical evidence of this.
>
> The only drawback of SCAFFLSA lies in the fact that each agent is required to store an additional vector (the local control variate), which increases the cost in terms of memory for each agent. This drawback is shared with all control variates methods (such as Scaffold, Scaffnew...), and is generally not a problem provided agents have access to sufficiently large memory.

---

> > ### Comment · Reviewer_bpaa · 2024-08-11
> >
> > I would like to thank the authors for their responses and for incorporating my suggestions in order to improve the readability of the paper. I have no further questions.

---

### Official Review · Reviewer_WekN · 2024-07-15

**Soundness:** 3
**Presentation:** 3
**Contribution:** 3
**Rating:** 6
**Confidence:** 3

**Summary:**

This paper first analyzed the performance of federated linear stochastic approximation or FedLSA algorithm. Second, it proposed a new algorithm called stochastic controlled averaging for Federated LAS or SCAFFLSA and analyzed its performance. The key idea of SCAFFLSA is to use a control variable to mitigate the client drift. The performance of the proposed algorithm was verified using experiments.

**Strengths:**

1. The paper proposed a new analytical framework to analyze the sample and communication complexity of FedLSA. Using this approach, the paper analyzed the performance of SCAFFLSA. The paper also extended the analytical framework to Federated TD learning.

2. The paper proposed a new algorithm, SCAFFLSA, which mitigate the client drift using a control variable. The sample and communication complexity of SCAFFLSA is significantly better than that of FedLSA and Scaffnew. The performance analysis of SCAFFLSA was also extended to Federated TD.

**Weaknesses:**

The experiments are relatively weak. It will be more beneficial if the authors could provide more applications of the proposed algorithm to other problems, especially for federated TD learning.

**Questions:**

What are the functions $\boldsymbol{A}^c()$ and $\boldsymbol{b}^c()$ like in practice? Could you please provide some examples?

**Limitations:**

There was no discussion on the limitations of the proposed algorithm.

---

> ### Author Rebuttal · Authors · 2024-08-02
>
> **The experiments are relatively weak. It will be more beneficial if the authors could provide more applications of the proposed algorithm to other problems, especially for federated TD learning.**
> The Garnet problems used in our experiments are common in the TD literature and serve the purpose of illustrating numerically our theoretical findings.
> Nonetheless, we are happy to add more experiments on other problems if the reviewer has some specific problems to suggest.
>
> **What are the functions $A^c()$ and $b^c()$ like in practice? Could you please provide some examples?**
> In TD learning, the system is given by $\bar{A}^c = \mathbb{E} [\phi(s)\{\phi(s)-\gamma \phi(s')\}^{\top}]$ and $\bar{b}^c = \mathbb{E}[\phi(s) r^c(s,a)]$, where $s \sim \mu^c, s' \sim P^{\pi,c}(\cdot|s)$.
> Thus, taking the sampling set $\mathsf{Z} = \mathcal{S} \times \mathcal{A} \times \mathcal{S}$, we can define $A^c(s, a, s') = \phi(s)\{\phi(s)-\gamma \phi(s')\}^{\top}$ and $b^c(s, a, s') = \phi(s) r^c(s,a)$.
> The points $s, a, s'$ are then either i.i.d. samples (following Assumption TD1) or a Markov chain (following Assumption TD2).

---

### Author Rebuttal · Authors · 2024-08-02

We thank the reviewers for their thorough feedback. We are pleased that reviewers deemed our contributions as new ("new analytical framework", "new algorithm", reviewer WekN; and "the ideas of the paper are new", reviewer bpaa), and that our analysis technique is the first to show that Scaffnew has linear speed-up ("and achieves a linear speedup in the number of clients. It is not attained by the original paper of Scaffnew", reviewer czvr).

Reviewers WekN and czvr found our experiments a little bit too simple. Although we stress that this paper is a theoretical paper, we understand the concern and provide some more experiments, with different settings of number of agents $N$ and number of local steps $H$ on a federated Garnet problem. The results can be found in the file attached: when agents are homogeneous, both algorithms perform very similarly; when agents are heterogeneous, FedLSA gets more and more biased as the number of iterations grow, while SCAFFLSA still reaches the same precision even when number of communication is divided by 1000.

We remark that the experimental setup that we used is very common in federated temporal difference learning, and we are not aware of other problems used in the literature. If reviewers have suggestions for other problems, we will be happy to add them to the manuscript.

We address the other more specific concerns directly in the rebuttal to each reviews.

---

### Decision · Program_Chairs · 2024-09-25

**Decision:**

Accept (poster)

**Comment:**

This paper employs control variates to correct for client drift, thereby improving the sample and communication complexity of the federated linear stochastic approximation (FedLSA) algorithm. All reviewers concur that it is a strong paper with significant contributions. We therefore recommend its acceptance.